# Cryo-EM structure of a type IV secretion system

Kévin Macé[1,8 ✉], Abhinav K. Vadakkepat[1,8], Adam Redzej[1], Natalya Lukoyanova[1], Clasien Oomen[1], Nathalie Braun[1,5], Marta Ukleja[1,6], Fang Lu[1], Tiago R. D. Costa[1,7], Elena V. Orlova[1], David Baker[2], Qian Cong[2,3 ✉] & Gabriel Waksman[1,4 ✉]

Bacterial conjugation is the fundamental process of unidirectional transfer of DNAs, often plasmid DNAs, from a donor cell to a recipient cell[1]. It is the primary means by which antibiotic resistance genes spread among bacterial populations[2,3]. In Gram-negative bacteria, conjugation is mediated by a large transport apparatus—the conjugative type IV secretion system (T4SS)—produced by the donor cell and embedded in both its outer and inner membranes. The T4SS also elaborates a long extracellular filament—the conjugative pilus—that is essential for DNA transfer[4,5]. Here we present a high-resolution cryo-electron microscopy (cryo-EM) structure of a 2.8 megadalton T4SS complex composed of 92 polypeptides representing 8 of the 10 essential T4SS components involved in pilus biogenesis. We added the two remaining components to the structural model using co-evolution analysis of protein interfaces, to enable the reconstitution of the entire system including the pilus. This structure describes the exceptionally large protein–protein interaction network required to assemble the many components that constitute a T4SS and provides insights on the unique mechanism by which they elaborate pili.

Conjugative T4SSs generally contain 12 proteins, VirB1–VirB11 and VirD4 (also known as TrwB in the R388 T4SS under investigation here), one of which (VirB1 (also known as TrwN)) is non-essential[6]. Three ATPases, VirB4 (also known as TrwK), VirB11 (also known as TrwD) and VirD4, power the system[5]. Three proteins, VirB7 (also known as TrwH), VirB9 (also known as TrwF) and VirB10 (also known as TrwE), form the outer membrane core complex (OMCC), which contains an O-layer embedded in the outer membrane and an I-layer underneath[7] (Extended Data Fig. 1a). The other proteins (except VirB2 (also known as TrwL), which forms the conjugative pilus and VirB5 (also known as TrwJ), which locates at the tip of the pilus) assemble to form three additional sub-complexes, which were revealed by two different low-resolution structural approaches, negative stain electron microscopy[8,9] (NSEM) and in cellulo cryo-electron tomography[10,11] (cryo-ET) (Extended Data Fig. 1a). These sub-complexes consist of an inner membrane complex (IMC) embedded in the inner membrane, a structure bridging the OMCC and the IMC (the stalk (also called the cylinder), and a ring complex surrounding the stalk (the arches) (Extended Data Fig. 1a). However, the two approaches reveal very different IMC architectures. NSEM provides a view of a double-barrelled IMC made of two side-by-side trimers of dimers (the barrels) of the AAA+ VirB4 ATPase, whereas cryo-ET shows a central hexamer of dimers of the same protein. Conjugative T4SSs must first produce a conjugative pilus, which makes contact with a recipient cell[12] and may serve as a conduit for DNA[13]. In this pilus biogenesis mode, only the VirB2–VirB11 proteins are required[14,15]. After contact between cells is made, the T4SS switches to a DNA-transfer mode involving VirB2–VirB11 and VirD4[16].

Here we present a single-particle cryo-EM structure of a T4SS complex from the R388 plasmid that comprises all four sub-complexes: OMCC, stalk, arches and IMC (Fig. 1a–c and Extended Data Figs. 1 and 2). Near-atomic resolution was achieved for all except for the arches sub-complex (Extended Data Fig. 3 and Supplementary Table 1). However, VirD4, VirB11, and VirB2 were absent from the structure (see Extended Data Fig. 1b for the naming convention; see Methods and Extended Data Fig. 1c for details). It became apparent early on during data processing that the various sub-complexes displayed very different symmetry and oligomerization states (Extended Data Fig. 4a–f). The IMC is composed of six protomers, each including one VirB3 (also known as TrwM), two VirB4, and the three N-terminal tails of VirB8 (hereafter referred to VirB8$_{tails}$; VirB8 is also known as TrwG). Three of these protomers are occupied to a significant degree, whereas the occupancy of the three others is weaker (Extended Data Fig. 4d). All protomers were related by an angle of 60° (Extended Data Fig. 4c,d). The IMC is thus a hexameric structure with compositional heterogeneity (that is, variable occupancy of its constituent protomers) as defined by Huiskonen[17], thereby limiting our ability to observe interactions and transitions in situ. The arches, comprising the VirB8 periplasmic domains (VirB8$_{peri}$), also form a hexameric assembly with variable occupancy (Extended Data Fig. 4c,d). The stalk is composed of pentamers of each VirB5 and VirB6 (also known as TrwI) (Extended Data Fig. 4e,f). The O-layer, which comprises full-length VirB7 and the C-terminal domains of VirB10 (VirB10$_{CTD}$) and VirB9 (VirB9$_{CTD}$), and the I-layer,

[1]Institute of Structural and Molecular Biology, Department of Biological Sciences, Birkbeck College, London, UK. [2]University of Washington, Molecular Engineering and Sciences, Seattle, WA, USA. [3]Eugene McDermott Center for Human Growth and Development University of Texas Southwestern Medical Center, Houston, TX, USA. [4]Institute of Structural and Molecular Biology, Division of Biosciences, University College London, London, UK. [5]Present address: strucTEM Microscopic Services, Gammelsdorf, Germany. [6]Present address: National Center For Biotechnology CNB-CSIC, Madrid, Spain. [7]Present address: MRC Center for Molecular Bacteriology and Infection, Department of Life Sciences, Imperial College London, London, UK. [8]These authors contributed equally: Kévin Macé, Abhinav K. Vadakkepat. ✉e-mail: k.mace@mail.cryst.bbk.ac.uk; congqian1986@gmail.com; g.waksman@ucl.ac.uk

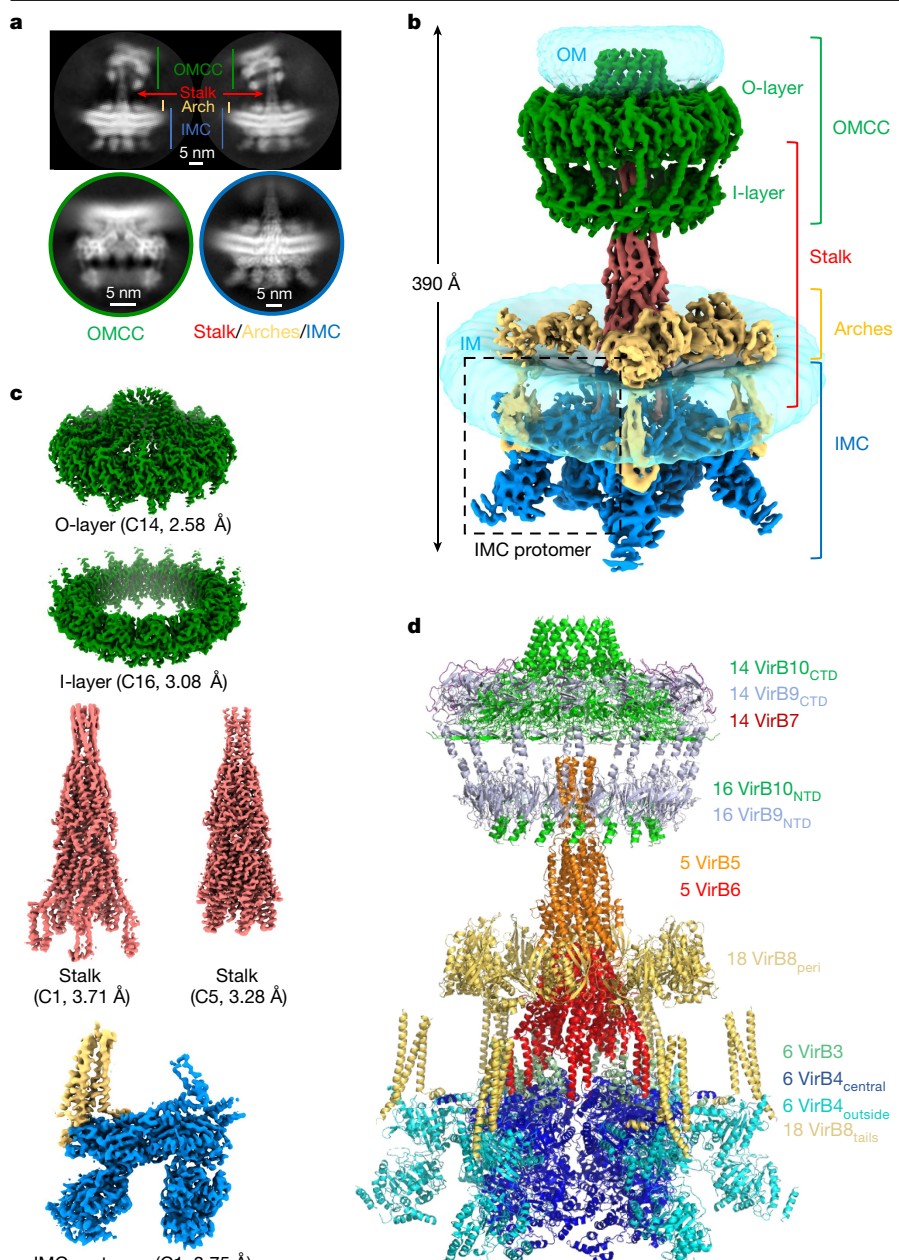

**Fig. 1 | Overall structure of the R388 conjugative T4SS. a**, Representative 2D class averages of the T4SS obtained using cryoSPARC. Top, two 2D class averages of the entire T4SS demonstrate substantial flexibility of the OMCC relative to the stalk and the IMC. As a result, particles were subsequently centred on the OMCC (bottom left) or on the IMC–stalk (bottom right) and processed separately. **b**, Composite electron density map of the R388 T4SS. This map results from the assembly of two C1 maps, that of the OMCC (OMCC C1 3.28 Å map) and that of the IMC, arches and stalk (IMC–arches–stalk C1 6.18 Å map). The OMCC, stalk, arches and IMC are shown in green, red, yellow and dark blue, respectively, except for the VirB8$_{tails}$ that are part of the IMC, which are shown in yellow. The various regions are labelled accordingly. $\sigma$ levels for these maps are as in Extended Data Fig. 3b,f. For the detergent and/or lipid densities (in transparent light blue) at the membrane and outer membranes, the maps are shown at increased contour levels of 0.03 and 0.15, respectively, and smoothed using a Gaussian filter. **c**, Near-atomic resolution maps used in this study. Each map is labelled and contoured as in Extended Data Fig. 3. The resolution of the map is indicated. **d**, Overall composite model of the R388 T4SS. Each protein is in ribbon representation.

composed of the N-terminal domains of these proteins (termed VirB10$_{NTD}$ and VirB9$_{NTD}$, respectively) form tetradecameric and hexadecameric assemblies, respectively (Extended Data Fig. 4a,b). Using these models and associated maps (Extended Data Fig. 5 and Methods), we constructed a composite model of the entire T4SS (Fig. 1d).

## The inner membrane complex

The IMC is 1.32 MDa in size and 295 Å in diameter (Fig. 2a). The main component of the IMC is the AAA+ VirB4 ATPase, for which 12 subunits are present. We first solved the structure of VirB4 in its unbound form (VirB4$_{unbound}$) and fitted and rebuilt the structure within the IMC density (Methods and Extended Data Fig. 6a–c). In the IMC, six dimers of VirB4 come together to form a hexamer of dimers (Fig. 2a). One subunit (VirB-4$_{central}$) in each of the 6 dimers form a central hexamer with a diameter of 130 Å (Fig. 2). The second subunit of the dimers (VirB4$_{outside}$; structurally very similar to VirB4$_{central}$ (Extended Data Fig. 6d)) protrudes out (in cyan blue in Fig. 2). This organization is similar to the architecture observed by low-resolution cryo-ET[10,11,18] (Extended Data Figs. 1a and 4d).

The dimer interface between VirB4$_{central}$ and VirB4$_{outside}$ is mediated entirely by the N-terminal domains of each subunit (Fig. 2b,c and Supplementary Table 2). By contrast, the interface between two adjacent VirB4$_{central}$ subunits in the central hexamer is spread out over both the N-terminal and C-terminal domains (Fig. 2a, Extended Data Fig. 6e and Supplementary Table 2).

VirB4$_{unbound}$ is also composed of dimers that have essentially the same structure as the VirB4$_{central}$–VirB4$_{outside}$ dimer in the T4SS (Cα root mean squared deviation (r.m.s.d.) 1.2 Å; Extended Data Fig. 6a,f). However, in this unbound structure, part of the dimer interface is used to form trimers of dimers (Extended Data Fig. 6b,g), which when mapped onto our T4SS structure forms two barrels of trimers of dimers, as in the previously reported double-barrelled architecture[8] (Extended Data Fig. 6h,i). Of note, a minority of the 2D classes in our cryo-EM data set displayed the typical side-views of the double-barrelled architecture observed in the NSEM structure (Extended Data Fig. 1f), indicating the presence of a small number of double-barrelled particles. Thus, the IMC protomer appears to provide a building block for the formation of various T4SS IMC assemblies.

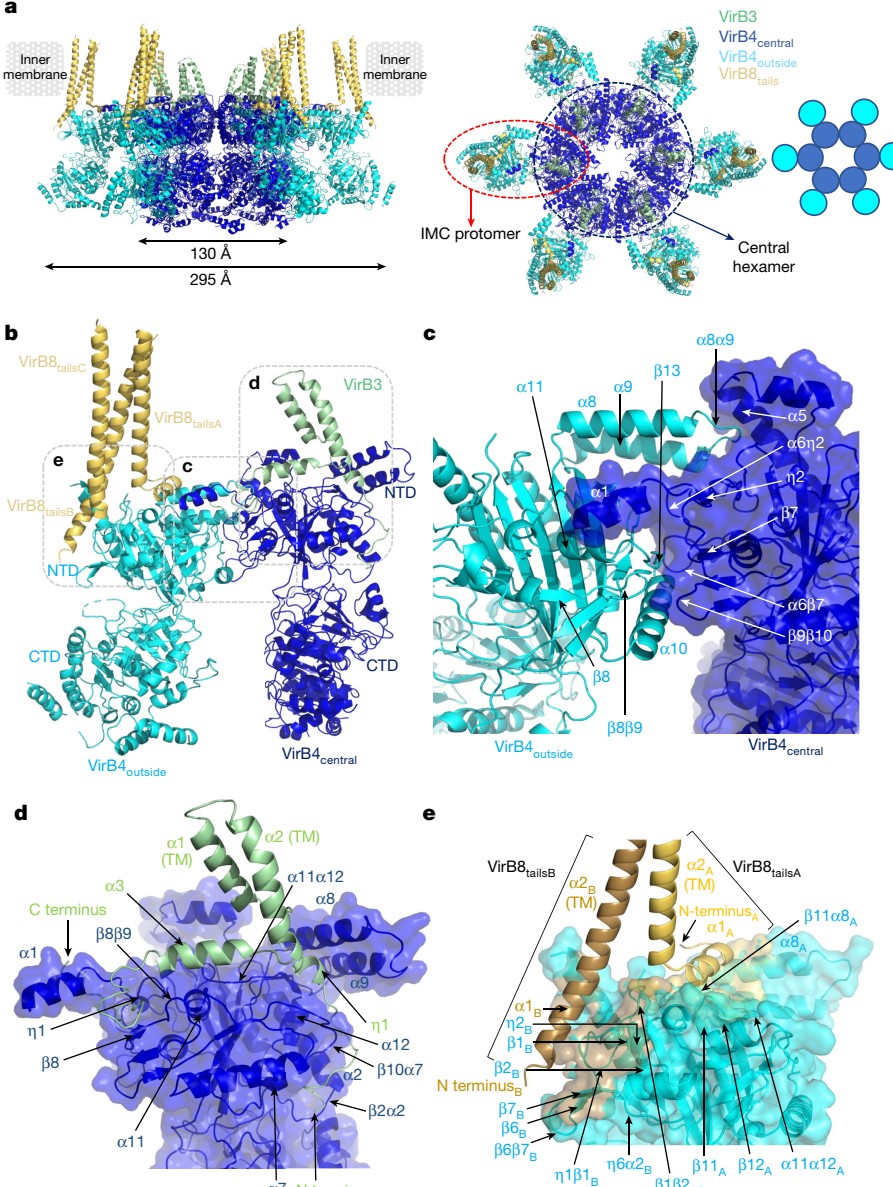

**Fig. 2 | Molecular details of IMC protein structures and interactions.**
a, Overall structure of the IMC. The IMC is shown in ribbon representation, with subunits coloured dark blue (VirB4$_{central}$), cyan (VirB4$_{outside}$), pale green (VirB3) and yellow (VirB8$_{tails}$). Left, side view of the IMC. The external dimensions of the central VirB4 hexamer and of the IMC are indicated, as well as the position of the inner membrane derived from the density. Right, top view of the IMC. The IMC protomer and the central hexamer are shown in a dashed red oval and dark blue circle, respectively. A schematic diagram of the hexamer of VirB4 dimers is shown on the right. b, Overall structure of the IMC protomer. Proteins are shown and colour-coded as in a. The boxes locate the regions detailed in c–e. c, Details of the interactions between subunits within the VirB4 dimer. VirB4$_{central}$ is shown in ribbon and semi-transparent surface in dark blue

and VirB4$_{outside}$ is shown in cyan ribbon. All secondary structures involved in the interactions are shown. d, Details of the interactions between VirB4 and VirB3. VirB4$_{central}$ is shown in dark blue ribbon and surface representation and VirB3 is shown in pale green ribbon. All secondary structures containing residues involved in the interaction are labelled. e, Details of the interactions between VirB4$_{outside}$ and two of the VirB8$_{tails}$. Only two are shown because although three VirB8$_{tails}$ form a three-helix bundle, one of the helices makes very few interactions with VirB4$_{outside}$. The two VirB8$_{tails}$ (VirB8$_{tailsA}$ and VirB8$_{tailsB}$) are shown in yellow and wheat ribbons, respectively. VirB4$_{outside}$ is shown in cyan ribbon and its semi-transparent surface is coloured yellow or wheat according to the VirB8$_{tail}$ that it interacts with, or cyan for non-interacting surfaces.

The IMC includes two other components: VirB3 and VirB8$_{tails}$ (Fig. 2b and Extended Data Fig. 6c). VirB3 makes interactions only with the VirB4$_{central}$ subunit of the dimer, and consequently there are six VirB3 subunits in the entire T4SS (Fig. 2a,b). VirB3 wraps around the N-terminal domain of VirB4$_{central}$ (Fig. 2d and Supplementary Table 2). Three helices are present (α1–α3), two of which, α2 and α3, form TM segments through the inner membrane (Fig. 2d and Extended Data Fig. 6c). By contrast, VirB8$_{tails}$ interact only with the VirB4$_{outside}$ subunit of the dimer and three copies of each are observed per VirB4$_{outside}$ subunit

(Fig. 2b); the IMC model thus includes 18 VirB8$_{tails}$. Of the three VirB8$_{tails}$ bound to each VirB4$_{outside}$ subunit, two form a unique extended helix (VirB8$_{tailsB}$ and VirB8$_{tailsC}$ in Fig. 2b), whereas in VirB8$_{tailsA}$, the N-terminal part adopts a very different conformation in which the helix is split into two helices with the N-terminal one being redirected to interact with other parts of VirB4$_{outside}$ (Fig. 2e). Only VirB8$_{tailsA}$ and VirB8$_{tailsB}$ interact substantially with the VirB4$_{outside}$ subunit, with VirB8$_{tailsC}$ making only sparse contact (Fig. 2b,e and Supplementary Table 2).

## The stalk, arches and OMCC

The stalk, a previously unknown structure, is a central, cone-shaped assembly 0.29 MDa in size, with a diameter of 92 Å and a length of 216 Å. It arises from the inner membrane and is composed of a pentamer of VirB6 inserted into the inner membrane and a pentamer of VirB5 mounted onto the VirB6 stalk base (Fig. 3a,b). The VirB6 consists of seven long α-helices, five of which are mostly hydrophobic (α1, α2 and α5–α7) (Extended Data Fig. 7a,b). Two of these helices, α1 and α2, insert into the inner membrane. VirB6 subunits interact extensively with each other (Extended Data Fig. 7c and Supplementary Table 2). The VirB5 subunits also interact with each other (Extended Data Fig. 7d and Supplementary Table 2). Structures of VirB5 homologues crystallized as single proteins are available[19,20] (Extended Data Fig. 7e). However, in its pentameric form, VirB5 appears to have undergone a conformational change compared with the protein on its own, with its N-terminal part projected out in a manner reminiscent of pore-forming proteins (Extended Data Fig. 7e). This function may be required to interact with the membrane of the recipient cell. Finally, one VirB5 binds between two VirB6 subunits (Extended Data Fig. 7f and Supplementary Table 2).

The arches (the composition and architecture of which were previously unknown) are composed of hexamers of homotrimeric units of VirB8$_{peri}$, forming a 177 Å diameter ring around the stalk (Fig. 3a,c). A feature of the homotrimeric unit is how the three subunits (labelled MolA–MolC in Extended Data Fig. 7g) come together: the MolA–MolB interface is very different from that formed between MolB and MolC; the MolA–MolB interface is similar to the interface in the periplasmic domain of the *Helicobacter pylori* VirB8 paralogue, CagV[21] (Cα r.m.s.d. 2.9 Å; Extended Data Fig. 7g, middle), whereas the the MolB–MolC interface is similar to the interface in *Brucella suis* VirB8[22] (Cα r.m.s.d. 2.2 Å; Extended Data Fig. 7g, right). Six VirB8$_{peri}$ homotrimeric units come together using the β1–β4 sheet on both sides (Extended Data Fig. 7h and Supplementary Table 2).

The OMCC is composed of the O-layer embedded in the outer membrane and the I-layer located underneath within the periplasm (Extended Data Fig. 8a). The structure of this sub-complex is very similar to that of the pKM101 plasmid O-layer[23] and that of the *Xanthomonas citri* I-layer[24] (details in Extended Data Fig. 8b–e), except that in *X. citri*, 14 VirB10$_{NTD}$ α1s were observed bound to 14 VirB9$_{NTDs}$, whereas here we observe 16 such complexes. Thus, two heterodimeric VirB10$_{NTD}$–VirB9$_{NTD}$ complexes insert into the I-layer (diametrically opposite, as shown in Extended Data Fig. 8f), whereas the C-terminal domains of these two complexes are not inserted in the O-layer. Similar symmetry mismatches have been reported between layers of the OMCC of other T4SSs[25,26].

A surprising feature of the T4SS structure presented here is the paucity of interactions between sub-complexes (Fig. 3d). Contacts are observed between the stalk and the IMC through interactions between the TM segments of VirB6 and VirB3 (Extended Data Fig. 8g; Supplementary Table 2). However, some interactions are yet unaccounted for, which may contribute to interactions between sub-complexes (Extended Data Fig. 8h, i). Finally, a cut-away view of the entire complex surface reveals that there is no continuous central channel, suggesting that the architecture revealed here is not that executing DNA transfer (Fig. 3d). We hypothesize below that, instead, this structure captures the state of the T4SS responsible for pilus biogenesis.

## Validation of the T4SS structure

Structural models are usually tested using site-directed mutagenesis of residues deemed to have important structural roles. However, more efficient and reliable methods for experimental structure validation are available, taking advantage of the fact that residues within interfaces are subjected to considerable evolutionary pressure[27–29]. Thus, we used

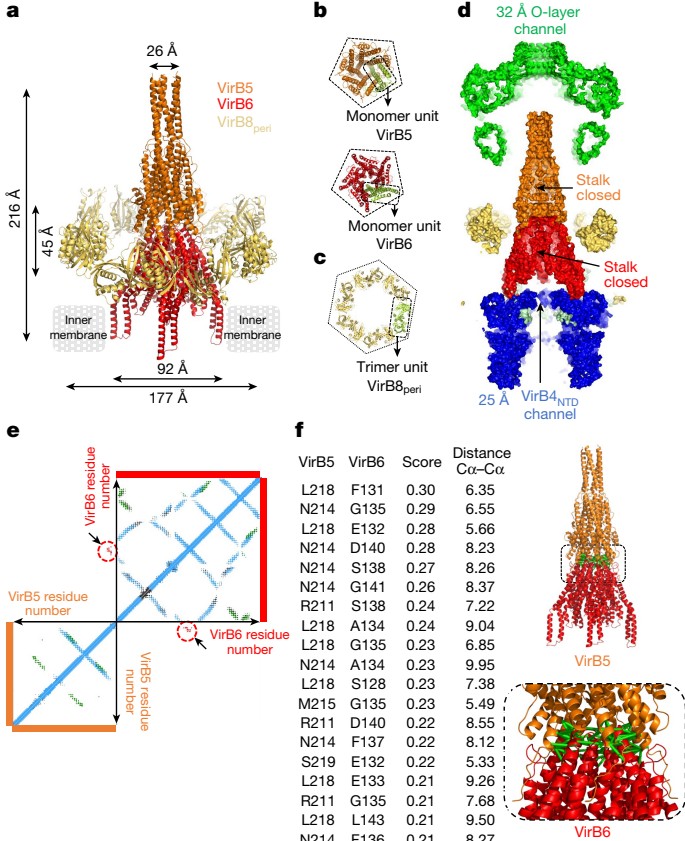

| VirB5 | VirB6 | Score | Distance Cα–Cα |
|---|---|---|---|
| L218 | F131 | 0.30 | 6.35 |
| N214 | G135 | 0.29 | 6.55 |
| L218 | E132 | 0.28 | 5.66 |
| N214 | D140 | 0.28 | 8.23 |
| N214 | S138 | 0.27 | 8.26 |
| N214 | G141 | 0.26 | 8.37 |
| R211 | S138 | 0.24 | 7.22 |
| L218 | A134 | 0.24 | 9.04 |
| L218 | G135 | 0.23 | 6.85 |
| N214 | A134 | 0.23 | 9.95 |
| L218 | S128 | 0.23 | 7.38 |
| M215 | G135 | 0.23 | 5.49 |
| R211 | D140 | 0.22 | 8.55 |
| N214 | F137 | 0.22 | 8.12 |
| S219 | E132 | 0.22 | 5.33 |
| L218 | E133 | 0.21 | 9.26 |
| R211 | G135 | 0.21 | 7.68 |
| L218 | L143 | 0.21 | 9.50 |
| N214 | F136 | 0.21 | 8.27 |

**Fig. 3 | Molecular details of stalk and arches protein structures and interactions, and structure validation by co-evolution analysis. a**, Overall structure of the stalk and arches. Stalk and arches proteins are shown in ribbon coloured orange (VirB5), red (VirB6) and yellow (VirB8$_{peri}$). Proteins constituting the complexes and the dimensions of the two sub-complexes–stalk and arches–are indicated. **b**, Symmetry arrangements of VirB5 and VirB6. All proteins are shown in ribbon representation, colour-coded as in **a**, except one monomer in each box is shown in green. Top, bottom view of the VirB5 pentamer. Bottom, bottom view of the VirB6 pentamer. The dashed line in both illustrates the pentameric nature of each structure. **c**, Top view of the arches, showing the symmetry arrangement of VirB8$_{peri}$. All proteins are shown in ribbon representation. The arches are made of six trimeric units of VirB8$_{peri}$, one of which is shown in pale green and outlined; the rest are colour-coded as in **a**. The hexagon surrounding the hexamer of trimer highlights the six-fold symmetrical arrangement of this part of the structure. **d**, Cross-section of the T4SS surface. The channels are shown with dimensions of interest. The VirB4$_{outside}$ subunits are not shown. **e**, Co-evolution at the interface of VirB5 and VirB6. Results of computational analysis. Each dot represents a pair of co-evolving residues with TrRosetta score ≥0.21. Dots are coloured blue (intra-protein co-evolution pairs), green (homo-oligomeric co-evolution pairs) or red surrounded by red circles and located by arrows (hetero-oligomeric co-evolution pairs). **f**, Co-evolution at the interface of VirB5 and VirB6. Left, list of hetero-oligomeric co-evolution pairs with TrRosetta scores above the threshold of 70% (Methods and main text) and Cα–Cα distances in angstrom in the structure reported here. Numbering is that of the R388 proteins. Full list in Supplementary Table 4. Right, mapping of co-evolution pairs listed in the table onto the VirB5–VirB6 stalk sub-complex structure. Pairs of residues across the interface are linked by green bars.

TrRosetta[30,31] to analyse the co-evolution of T4SS components (Methods and Supplementary Table 4). An example is shown for the VirB5–VirB6 interface (Fig. 3e). Residue pairs between proteins were ranked by TrRosetta scores and all pairs above a threshold of 70% of the score for the top-scoring pair were mapped and displayed on the structure of the stalk (Methods). All top-scoring pairs displayed in this way are

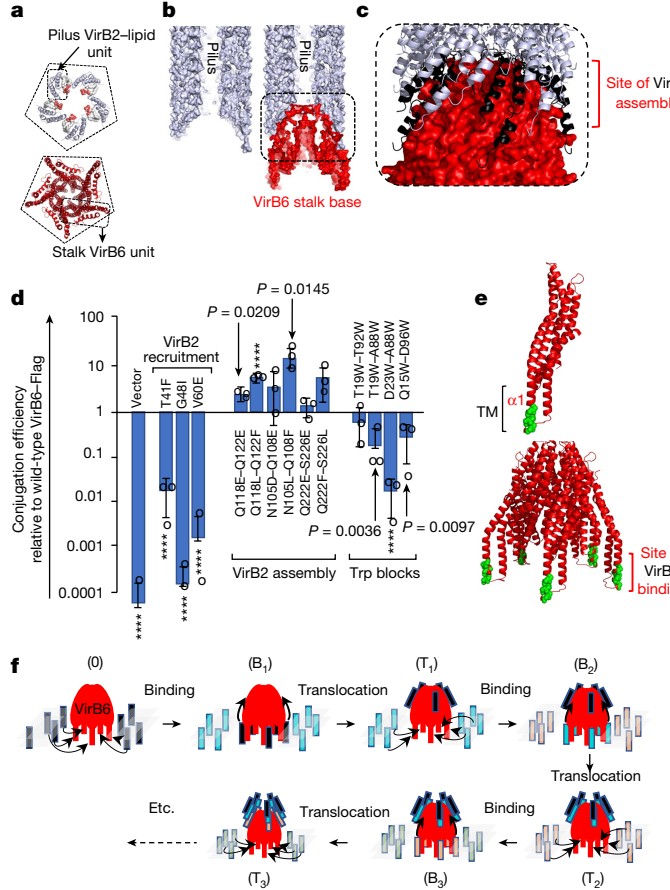

located within the interface between the proteins (Fig. 3f), thereby validating our model. Regions of structures that do not interact score poorly compared with regions that do interact, providing internal validation of the co-evolution results. Similar results were obtained for the VirB3–VirB4$_{central}$, VirB10$_{CTD}$–VirB9$_{CTD}$ and VirB4–VirB8$_{tails}$ interactions (Extended Data Fig. 9a–c and Supplementary Table 4), suggesting that our structural models are accurate.

## Mechanism of pilus biogenesis by T4SS

The conjugative T4SS functions as a pilus biogenesis machinery, elaborating a long polymer of VirB2–phospholipid units[1,32]. Prior to being assembled into a pilus, VirB2 subunits are located in the inner membrane[32]. Pilus assembly requires the VirB2 subunits to be extracted from the membrane through the concerted action of the VirB4 and VirB11 ATPases[33,34].

VirB4 and VirB11 have been previously shown to interact using cryo-ET[18,35]. Using AlphaFold[36], a deep learning method to predict 3D structures, we generated a model for the VirB4–VirB11 interaction for both R388 VirB4–VirB11 and the paralogue from the related pKM101 plasmid (TraB–TraG) (Extended Data Fig. 10a,b). The models were highly similar to our VirB4 structure and previous VirB11 structures[37–39] (Extended Data Fig. 10a). Next, using TrRosetta, we obtained a list of co-evolving pairs of residues between VirB4 and VirB11 (Supplementary Table 4). All pairs with a TrRosetta score above the threshold of 70% of the score of the top-scoring pair (Supplementary Table 4) were mapped onto the AlphaFold-generated VirB4–VirB11 models (Extended Data Fig. 10b, middle). The paired residues all mapped onto the interface, providing a validation of the model. To provide further independent validation of the VirB4–VirB11 model, we co-purified the TraB–TraG complex (as R388 VirB4–VirB11 could not be purified as a stable complex) and assessed biochemically the effect of eight single interface residue mutations—four on TraB and four on TraG—on the stability of the interaction (Extended Data Fig. 10b,c). All 8 mutations significantly weakened the TraB–TraG interaction, thus providing further validation of the proposed structural model of VirB4–VirB11. As noted above, the IMC protomers are not equally occupied, and therefore VirB4 is unlikely to function as an ATPase before a full hexamer is formed. By contrast, VirB11 is a constitutively hexameric protein[37,38]. By binding to VirB4, it may therefore stabilize a full set of six IMC protomers, giving rise to a fully functional VirB4$_{central}$ ATPase capable of orchestrating pilus biogenesis. In that context, the role of VirB4$_{outside}$ remains unclear.

Conjugative pili are five-start helical assemblies of VirB2–phospholipid units[32]. The base of a pilus is made of five symmetrical VirB2–phospholipid complexes, a symmetry matching that of the T4SS stalk (Fig. 4a). Moreover, when the shape complementarity of the F TraA (VirB2) pentamer (there is no known structure of the R388 pilus) is assessed against the shape of the VirB6 pentamer using the shape-complementarity software Patchdock[40], the ten top-scoring structures show the concave side of the VirB2 pentamer docking on top of VirB6 (Extended Data Fig. 10d), allowing the docking of the entire F pilus accordingly (Fig. 4b). Thus, we hypothesize that the pilus may locate between VirB5 and VirB6 (Extended Data Fig. 10e), consistent with reports describing VirB5 at the distal end of the pilus[12]. So placed, the pilus would grow from the concave end using VirB6 as a base. We propose that the surfaces of each of the five VirB6 subunits that make contact with the first VirB2 pentamer layer of the pilus may form the site of VirB2 assembly (Fig. 4c). To validate this site, we made three pairs of double mutants (Extended Data Fig. 10f–h), each to either acidic or hydrophobic residues. Since pilus biogenesis is essential for conjugation between bacteria to occur, these mutated T4SSs were tested for their ability to mediate conjugation (Fig. 4d). Notably, these mutants all displayed increasing conjugation rates (Fig. 4d), with substitution to hydrophobic residues having a more pronounced effect.

Before pilus biogenesis, VirB2 pilus subunits are embedded in the inner membrane. Given the C5 symmetry of the pilus and the fact that

**Fig. 4 | Mechanism of pilus biogenesis by conjugative T4SSs. a**, Conjugative pili and the stalk have the same C5 symmetry. Top, the F pilus pentamer unit with TraA (VirB2) shown in white ribbon and the phospholipid shown in ball-and-stick representation colour-coded by atoms. Bottom, the VirB6 pentamer shown in red ribbon. A pentagon is shown to highlight the C5 symmetry. The outlines indicate the monomeric unit. **b**, Cut-away surface of the conjugative pilus (left) and of the pilus–VirB6 interaction (right). The pilus and VirB6 are shown in white and red surfaces, respectively. The outlined region is magnified further in **c**. **c**, Magnified view of the pilus–VirB6 interface. VirB6 is shown in red surface representation. The pilus VirB2 subunits are shown in white ribbon, except for the VirB2 pentamer at the bottom of the pilus, which is shown in black. The VirB2-contacting region on VirB6 defines this region as the site of VirB2 assembly (labelled). **d**, Mutational analysis of the surfaces of VirB6 hypothesized to form the binding/recruitment site (VirB2 recruitment), the assembly site (VirB2 assembly) and the effect of Trp mutations (Trp blocks) between the two sites. Locations of mutations in the structure are shown in Extended Data Fig. 10f,g. Conjugation results (data points indicated by open circles) are reported from three independent experiments ($n = 3$) and expressed as mean ± s.d. Unpaired two-tailed Student's $t$-test with 95% confidence level was used to compare wild-type and mutant constructs. Significant $P$-values (less than 0.1) are shown, except where $P ≤ 0.0001$ (indicated by ****). **e**, Identification of the VirB2 binding/recruitment site on VirB6. The residues of VirB6 in the 50 top-scoring co-evolving residues obtained by TrRosetta between VirB2 and VirB6 were mapped onto the VirB6 structure (list in Supplementary Table 4). VirB6 is shown in red ribbon, except for the mapped residues, for which only the Cα atom is shown, coloured green. Top, the VirB6 monomer. Bottom, the VirB6 pentamer. **f**, Model of pilus biogenesis by conjugative T4SSs. Three cycles of VirB2 subunit incorporation are shown. VirB6 is shown as a dome-like red diagram with five legs (its transmembrane helices). The inner membrane is shown as semi-transparent lozenges. VirB2 subunits are shown as vertical rectangles colour-coded differently for each cycle. State 0 represents the structure described here. $B_x$, VirB2-bound state at the VirB6 transmembrane regions in cycle $x$. $T_x$, translocated state in cycle $x$ in which the VirB2 subunits have reached the assembly site.

the only membrane protein that assembles into a C5 assembly is VirB6, we hypothesized that VirB6 may contain a binding or recruitment (binding/recruitment) site for VirB2 subunits. We therefore investigated the co-evolution between VirB2 and VirB6 using TrRosetta. VirB6 residues in all 50 top scoring pairs were located in the transmembrane α1 helix, making it a strong candidate to form a VirB2-binding/recruitment site (Fig. 4e and Extended Data Fig. 10i; full list in Supplementary Table 4). To test this hypothesis, we introduced three mutations in VirB6 α1 and its immediate vicinity (Extended Data Fig. 10f–h): these mutants exhibited decreased conjugation (Fig. 4d), consistent with the hypothesis that VirB6 α1 may contain the VirB2-binding/recruitment site. On the basis of TrRosetta analysis, another strong candidate to form a VirB2 binding site is VirB3 α1 (Extended Data Fig. 10j and Supplementary Table 4). However, VirB3 is hexameric, a symmetry that does not match the symmetry of the pilus. We therefore propose that VirB2 binding to VirB3 α1 may represent an intermediate binding station for VirB2 subunits.

The data presented here suggest a model for pilus biogenesis by T4SS whereby five VirB2 subunits bound to five VirB6 subunits (Fig. 4f) are levered up to the assembly site, while five more are recruited to the vacated binding sites. To further test this model, we introduced Trp residues between the binding/recruitment site and the assembly site, which may form steric obstacles (Trp blocks) affecting the translocation path of VirB2 subunits from their binding/recruitment site to their assembly site (Fig. 4d and Extended Data Fig. 10f–h): all mutants exhibited decreased conjugation, consistent with expectation. The previously described VirB2 dislocation function of VirB4[33] could comprise levering up VirB2 subunits from the recruitment site to the assembly site. The identities of the regions of VirB4 that act as a lever remain unclear. However, potential triggers may include binding of VirB11[35] as well as ATP binding and/or hydrolysis. As layers of pentameric VirB2 are added, the pilus grows from the bottom, pushing the VirB5 pentamer out, passing through the arches, the I-layer (no conformational changes are needed (Extended Data Fig. 10k,l)), and finally through the O-layer channel, which is known to be flexible enough to open up[7,23] (Extended Data Fig. 10k,l).

Thus, the near-atomic structure of a conjugative T4SSs presented here provides the structural basis for a plausible model for conjugative pilus biogenesis by T4SSs. Conjugative pili are crucial appendages, without which DNA transfer among bacterial populations would not occur and thus the structure also provides the means to develop anti-conjugation strategies (including the design of pilus assembly inhibitors) that could limit the spread of antibiotic resistance genes among pathogens.

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

## Methods

### Bacterial strains and constructs

Strains, plasmids, constructs and oligonucleotides used in this study are shown in Supplementary Table 3.

### Expression and purification of T4SS

Plasmid pBADM11_trwN/virB1-trwE/virB10Strep_rbstrwD/virB11_rbsHistrwB/virD4 (a clone shown to mediate conjugation[9]) was used. Expression, detergent extraction and strep column purification was performed as described[9]. Next, the T4SS complex was concentrated by ultracentrifugation at 195,500 $g$ for 1 h. The pellet was resuspended by overnight incubation at 4 °C with 400 μl of a buffer containing 50 mM HEPES pH 7.6, 200 mM sodium acetate, 0.1% digitonin, 0.05 mM tetradecyldimethylamineoxide (TDAO). The resuspended pellet was then loaded onto a 15-45% sucrose density gradient made in the same buffer and centrifuged at 99,223 $g$ (using a SW40Ti rotor) at 4 °C for 18 h. Samples from the gradient were fractionated and analysed by SDS–PAGE. Fractions containing the T4SS were used for cryo-EM data collection after sucrose removal using a NAP10 column (Amersham).

### Cloning, expression and purification of VirB4~unbound~

Coding regions of trwM/virB3 and trwK/virB4 were amplified using PCR from pMAK3 plasmid (R388) and cloned using BsaI restriction sites at the 5′ end on the primer to generate IBA3C:trwM/virB3-trwK/virB4_C-Strep. Hcp1 amplified from the pET29b:hcp1_C-His plasmid and L6 linker (GSGSGS) were subsequently cloned into IBA3C:trwM/virB3-trwK/VirB4_C-Strep using In-Fusion cloning (Takara Bio) to yield IBA3C:trwM/virB3-trwK/virB4-L6-hcp1_C-Strep, from which the VirB4-L6-HCP1 (termed VirB4~unbound~) protein was expressed and purified (VirB3 did not co-purify).

Cells were cultured in chloramphenicol (35 μg ml⁻¹) containing LB medium at 37 °C until they reached absorbance at 600 nm ($A_{600\,nm}$) of 0.7 and induced overnight using 0.2 mg ml⁻¹ anhydrotetracycline (Abcam) at 18 °C. Cells were then pelleted by centrifugation at 5,000 $g$ for 30 min and resuspended in lysis buffer (50 mM Tris pH 8.0, 800 mM NaCl, 1 mM EDTA and 1 mM DTT) with 4 protein inhibitor cocktail tablets (Roche), 1 mg ml⁻¹ hen egg lysozyme (Sigma) and 20 μl Benzonase-Nuclease (Sigma) followed by lysis in an EmulsiFlex-C3 high pressure homogenizer (Avestin). The lysate was clarified by centrifugation at 12,000 $g$ for 30 min and the supernatant was loaded onto a GE Healthcare StrepTrap HP 5 ml column pre-equilibrated with buffer B (50 mM Tris pH 8.0, 400 mM NaCl, 2 mM EDTA and 2 mM DTT). For elution, buffer B was supplemented with 2.5 mM of desthiobiotin, peak fractions were pooled and purified further using the HiTrap Q-Sepharose HP anion exchange column (GE Healthcare) using a linear gradient of buffer C (50 mM Tris pH 8, 100 mM NaCl, 1 mM EDTA and 2 mM DTT) with 1 M NaCl. This was followed up by size exclusion chromatography using Superose-6 Increase 100/300 GL column (GE Healthcare) equilibrated with buffer D containing 50 mM Tris pH 8, 400 mM NaCl, 10 mM magnesium acetate and 2 mM DTT, peak fractions were analysed using SDS–PAGE, pooled, quantified, flash frozen and stored at −80 °C.

### Assessment of the stability of the TraB–TraG complex in the presence of detergents

pCDFDuet-1 in which traB/virB4_C-His and traG/virB11_C-Strep were cloned was transformed into Escherichia coli BL21(DE3)*. The cells were cultured in Terrific Broth (Formedium) with spectinomycin (100 μg ml⁻¹) (Sigma) at 37 °C until $A_{600nm}$ = 0.7 and induced overnight using 1 mM Isopropyl β-ᴅ-1-thiogalacto-pyranoside at 18 °C. The cells were then pelleted by centrifugation at 5,000 $g$ for 30 min and resuspended in lysis buffer (50 mM Tris pH 8.0, 150 mM NaCl, 2 mM EDTA, 1 mM DTT) with 4 protein inhibitor cocktail tablets (Roche), 1 mg ml⁻¹ hen egg lysozyme (Sigma) and 20 μl Benzonase–Nuclease (Sigma). After 30 min vortexing at 4 °C, cells were lysed by passing through EmulsiFlex-C3 high pressure homogeniser (Avestin). The lysate was clarified by centrifugation at 18,000 $g$ for 30 min at 4 °C and the supernatant was loaded onto two GE Healthcare StrepTrap HP 5 ml columns, each pre-incubated with buffer D (50 mM Tris pH 8, 150 mM NaCl, 1 mM DTT). For elution, buffer D was supplemented with 2.5 mM of desthiobiotin, peak fractions were pooled after SDS–PAGE analysis and loaded onto a GE Healthcare HisTrap HP 5 ml column pre-equilibrated with buffer D. After extensive wash with buffer D followed by a wash with buffer D supplemented with 20 mM imidazole, the recombinant protein was eluted in high imidazole buffer (50 mM Tris pH 8, 150 mM NaCl, 1 mM DTT and 300 mM imidazole) in one step and analysed on SDS–PAGE. Eluted sample was divided in two with one buffer-exchanged in the membrane-extraction buffer described above for the T4SS complex and the other in the same buffer but without detergents, and subsequently analysed using SDS–PAGE.

### Testing the effect of single site mutations on the stability of the TraB/VirB4-TraG/VirB11 complex

pCDFDuet1 containing traB/virB4_C-His and traG/virB11_C-Strep was used as a template to design primers and introduce point mutations (Q8D, R54E, N55E and K58E) in TraG and (E591R, E594R, A598E and Y619R) in TraB using PCR and In-Fusion cloning (Takara Bio). After the lysate-clarification step described above, supernatants were loaded onto a GE Healthcare HisTrap HP 5 ml column pre-equilibrated with buffer D (50 mM Tris pH 8, 150 mM NaCl, 1 mM DTT). After extensive wash with buffer D followed by a wash with buffer D supplemented with 20 mM imidazole, the recombinant protein was eluted in a gradient of high imidazole buffer (50 mM Tris pH 8, 150 mM NaCl, 1 mM DTT and 300 mM imidazole) and analysed on SDS–PAGE, adjusting load volumes so as to have equal amounts TraB on the gel. Western blot was performed to confirm the identities of TraB and TraG using a Bio-Rad mini-blot module and the two proteins forming the complex were probed using horseradish peroxidase (HRP) conjugated anti-His (Sigma Aldrich; 1:2,000 dilution)) and anti-Strep (EMD Merck; 1:4,000 dilution) antibodies and visualized by incubation with SigmaFast DAB with metal enhancer (Sigma Aldrich). Expression of TraG wild-type and mutants was assessed by comparing the corresponding TraG band in induced and non-induced cells. Non-edited pictures of gels and westerns are shown in the supplementary information.

### Cloning, expression and conjugation assay of VirB6–Flag mutants

Coding region of trwI/virB6 was amplified using PCR from pMAK3 plasmid (R388) and cloned into pBADM11 vector using In-Fusion cloning (Takara Bio). This clone was modified by addition of flexible linker composed of Gly-Ser-Gly and a Flag tag at the 3′ end of the trwI/virB6 gene by PCR amplification followed by In-Fusion cloning (Takara Bio). Mutations were introduced into the trwI/virB6-Flag by In-Fusion cloning (Takara Bio). R388Δtrwl/virB6 was generated by incorporation of chloramphenicol cassette inside of trwI/virB6 gene by homologous recombination according to a protocol described previously[41,42] using the SW102 strain[43].

The mating assay was performed as previously described[9,44]. E. coli TOP10 strain containing R388Δtrwl/virB6 plasmid and complementation plasmids expressing VirB6-Flag or mutants were used for mating assay as donor strains and E. coli DH5α as recipient strain. The conjugation frequencies were calculated as transconjugants per recipients. All experiments were performed three times. The data are expressed as mean ± s.d. For comparison of two groups, an unpaired $t$-test was employed as implemented at https://www.socscistatistics.com/tests/studenttest/default2.aspx. Unprocessed numbers are reported in the supplementary information.

### Detection of expression levels of VirB6–Flag and mutants in membranes

E. coli TOP10 strain containing R388Δtrwl/virB6 plasmid and complementation plasmids expressing VirB6-Flag or mutants were cultured

in LB medium containing trimethoprim (10 μg ml⁻¹) and carbenicillin (100 μg ml⁻¹) at 37 °C until the cells reached $A_{600nm}$ = 0.5–0.6. The expression of VirB6–Flag or mutants was induced by addition of 0.05% arabinose for 1 h at 37 °C. Cell were pelleted by centrifugation at 5,000 g for 15 min and resuspended in resuspension buffer (50 mM Tris pH 7.5, 200 mM NaCl, 1 mM EDTA) with protease inhibitor cocktail tablets (Roche) followed by lysis by sonication. The lysate was clarified by centrifugation at 25,750 g for 30 min, and the membrane pellet was collected by ultracentrifugation at 95,834 g for 45 min. The pellet was washed with resuspension buffer and membrane was pelleted by another ultracentrifugation at 95,834 g for 45 min. The membrane pellets were resuspended in NuPAGE LDS Sample Buffer (Thermo Fischer), boiled at 95 °C for 5 min and cooled down on ice before loading to the SDS–PAGE gel. The antibodies used to detect the amount of expressed VirB6–Flag from different constructs were anti-Flag antibody produced in rabbit (Abcam; 1:4,000 dilution) followed by incubation with anti-rabbit antibody conjugated with horseradish peroxidase (Abcam; 1:5,000 dilution). The bands were visualized by incubating the membrane with SigmaFast DAB with metal enhancer (Sigma-Aldrich).

## Cryo-EM grid preparation and data acquisition

C-flat grids (Protochips; 2/2 400 mesh and 1.2/1.3 300 mesh) were used for T4SS and UltrAuFoil grids (Quantifoil; 1.2/1.3 300 mesh) for VirB4$_{unbound}$ protein (preincubated with 0.1 mM LDAO (N,N-dimethyl-1-dodecanamine-N-oxide; Anatrace)). Grids were negatively glow discharged using PELCO Easiglow (Ted Pella) and coated with graphene oxide[45]. After application of 3 μl of sample, grids were incubated for 20–30 s in the chamber of a Vitrobot Mark IV (Thermo Fisher Scientific, USA) at 4 °C and 94% humidity and vitrified in liquid ethane.

The T4SS data were collected at the ISMB Birkbeck EM facility using a Titan Krios microscope (Thermo Fisher Scientific) operated at 300 keV and equipped with a BioQuantum energy filter (Gatan) with a slit width of 20 eV. The images were collected with a post-GIF K3 direct electron detector (Gatan) operating in super-resolution mode, at a magnification of 81,000 corresponding to a pixel size of 1.067 Å. The dose rate was set to 19.2 e⁻ per pixel per second and a total dose of 57.5 e⁻ per Å² was fractionated over 50 frames. Data were collected using the EPU (version 2.7) software with a defocus range −1.5 μm to −3.3 μm. A total of 104,711 movies were collected.

The VirB4$_{unbound}$ data were collected using the same setup as for the T4SS. The images were collected with a post-GIF K2 Summit direct electron detector (Gatan) operating in counting mode, at a magnification of 130,000 corresponding to a pixel size of 1.048 Å. The dose rate was set to 5.38 e⁻ per pixel per second, and a total dose of 49 e⁻ per Å² was fractionated over 50 frames. Data were collected using EPU software (Thermo Fisher) with a defocus range −1.2 μm to −2.8 μm. A total of 6,184 micrographs were collected in one session.

## Image processing of T4SS

MOTIONCOR2[46] was used for motion correction and dose weighting, followed by contrast transfer function (CTF) estimation using CTFFIND v4.1[47]. Workflows for image processing are reported in Extended Data Fig. 2.

## Image processing of the T4SS OMCC

Reprojections of a low pass filtered (20 Å) map generated using PDB 3JQO[23] were used to pick particles centred on the OMCC using GAUTOMATCH v0.56[48]. A total of 1,729,311 particles were selected after multiple rounds of 2D classification using cryoSPARC v2.15[49] (Extended Data Fig. 2a,b).

**Symmetry analysis.** One-hundred thousand of these particles (chosen automatically by cryoSPARC) were submitted to ab initio reconstruction with no symmetry applied and the resulting map was used as initial model to a 3D refinement with the same particles, yielding a map at 3.52 Å (referred to in Supplementary Table 1a as the ab initio model for OMCC C1 map; Extended Data Fig. 3a). Symmetry of the O-layer was assessed visually by displaying sections of the corresponding region of the ab initio map (Extended Data Fig. 4a, top left), clearly indicating C14 symmetry for this region. Then, a 3D homogeneous refinement was carried out using this new map as initial model using all 1,729,311 particles, yielding a C1 map of the OMCC with a resolution of 3.28 Å (referred to in Supplementary Table 1a as the OMCC C1 3.28 Å map; Extended Data Fig. 3b). To assess the symmetry in various regions of this map (O- and I-layers), sets of map sections were selected and extracted as separate images using the FIJI 1.53[50] software. Images were imported into IMAGIC-5[51] where the function rotational-auto-correlation was used and results were plotted (Extended Data Fig. 4a,b).

**Structure determination.** Using RELION 3.1[52], the 1,729,311 particles were re-extracted, re-centred and subjected to 3D refinement using the low pass filtered map mentioned above as initial model, with C14 symmetry applied. The outputs of this job were used for two different 3D classifications using RELION, one with a mask focused on the O-layer with C14 symmetry applied, and the other comprising the I-layer with C16 symmetry applied. Both classifications were performed without image alignment and using Tau = 100. The best resulting classes (based on the presence high resolution features) corresponded to 1,280,606 particles for the O-layer and 709,769 particles for the I-layer. These particles were selected to perform homogeneous refinement with cryoSPARC using corresponding symmetries, on-the-fly CTF and defocus refinement. The resulting electron density maps have an average resolution of 2.58 Å for the O-layer (O-layer C14 2.58 Å map) and 3.08 Å for the I-layer (I-layer C16 3.08 Å map) as estimated using the gold standard Fourier shell correlation (FSC) with a 0.143 threshold (Extended Data Fig. 3c,d and Supplementary Table 1a). For the I-layer, we further validated the C16 symmetry by applying C14 or C15 symmetry which yielded maps of much inferior quality compared to C16 (see Methods and Extended Data Fig. 4b).

## Image processing of the T4SS IMC, stalk and arches

Reprojections of the negative-strain EM map of the IMC, stalk and arches[9] (EMDB 3585) were used to pick particles centred on the IMC, stalk and arches using GAUTOMATCH. Particles were extracted, binned and subjected to multiple rounds of 2D classification using cryoSPARC, resulting in the selection of 1,292,734 particles which were re-extracted without binning and re-centred using RELION.

cryoSPARC was then used for all subsequent image processing described below.

To generate our first 3D map, we first made a strict selection of 60,722 particles (see Extended Data Fig. 2c for selection) which were obtained from several rounds of 2D classification, which then were used for the ab initio 3D classification with no symmetry imposed. The resulting map was used as the initial model for all subsequent processing (Extended Data Fig. 3e and Supplementary Table 1a).

Using the 1,292,734 particles mentioned above and this new reference map, heterogeneous 3D classification was carried out and resulted in the selection of a subset of 566,815 particles which were subjected to 3D homogeneous refinement with no symmetry imposed (Extended Data Fig. 2d). This yielded a map with an average resolution of 6.18 Å (IMC-arches-stalk C1 6.18 Å map; Extended Data Fig. 3f, Supplementary Table 1a). In this map, we observed three regions: a cone shape structure (the stalk) surrounded by a ring (the arches), both located above an assembly of three large bulks of density (the IMC). However, when applying lower contour levels, two additional large bulks of density were observed in the IMC (Extended Data Fig. 4d) and a third additional one was observed when contouring the map at an even lower level. All 5 readily visible bulks of density were related by ~60° (Extended Data Fig. 4c,d). The ~60° angles between IMC density bulks were confirmed using the map symmetry analysis methods described above for the OMCC (Extended Data Fig. 4c).

This observation led us to focus on one of the three better-defined density bulks which we define as the IMC protomer (Extended Data Fig. 2d, left). Thus, the IMC–arches–stalk C1 6.18 Å map and corresponding subset of particles were used to perform particle subtraction using a mask excluding the IMC protomer so defined. Following local refinement with no symmetry applied, a map of the IMC protomer was thus obtained with an average resolution of 3.75 Å as estimated by the gold-standard FSC at a 0.143 threshold (IMC protomer C1 3.75 Å map; Extended Data Fig. 3g and Supplementary Table 1a) into which 2 VirB4 subunits (the $VirB4_{central}$–$VirB4_{outside}$ dimer), 1 VirB3 and 3 $VirB8_{tails}$ were built and refined.

For the stalk, the IMC-arches-stalk C1 6.18 Å map was first subjected to the same symmetry analysis described above for the OMCC and was found to be five-fold symmetrical (Extended Data Fig. 4e). Next, the corresponding subset of particles were used to perform particle subtraction using a mask excluding the cone shape structure (Extended Data Fig. 2d, right). Following local refinement with no symmetry applied, a map of the stalk was obtained with an average resolution of 3.71 Å as estimated by the gold-standard FSC at a 0.143 threshold (stalk C1 3.71 Å map; Extended Data Fig. 3h and Supplementary Table 1a) into which 5 VirB6 and 5 VirB5 subunits were built and refined. In this map, the transmembrane regions of 2 VirB6 subunits were poorly resolved, suggesting flexibility. After confirming C5 symmetry (Extended Data Fig. 4f), symmetry was applied using a mask that excluded the VirB6 transmembrane regions, yielding much improved density for the included regions (stalk C5 3.28 Å map; Extended Data Fig. 3i and Supplementary Table 1a).

For the arches, we also observed three bulks of density (as for the IMC) in the IMC–arches–stalk C1 6.18 Å map (Extended Data Fig. 4d, right). Symmetry analysis as described above showed these bulks (made of 3 $VirB8_{peri}$ domains) to be related by ~60° angles, suggesting that, like for the IMC, the arches protomers locate along a hexagon and therefore form a hexamer (Extended Data Fig. 4c,d). The strategy used for the IMC (particle subtraction/local refinement) was therefore used, but did not produce high-resolution features for this region. Nevertheless, in the IMC–arches–stalk C1 6.18 Å map, secondary structural features were clearly recognizable and this map was used to dock homology models of $VirB8_{peri}$ without side chains (see details below).

All maps used for model building were subjected to sharpening using AutoSharpen in Phenix v1.18.2[53] and local resolution estimated using cryoSPARC.

## Image processing of $VirB4_{unbound}$

MOTIONCOR2 was used for motion correction and dose weighting, followed by CTF estimation using CTFFIND v4.1. After removing micrographs with non-vitreous ice, poor particle distribution or poor CTF fit, a total of 3,931 micrographs were selected for subsequent processing. RELION auto-picking with high threshold was first used to pick 54,956 particles which were then used in multiple rounds of 2D classification using cryoSPARC. The best 2D averages were then used as a template for particle picking using GAUTOMATCH using a low threshold (0.1), yielding 1,622,003 particles. After multiple rounds of 2D classification and selection focusing on removing excess bottom and top views, followed by ab initio 3D classification using cryoSPARC, 209,217 particles were selected. Homogenous refinement using this set of particles together with the ab initio map as reference yielded a 4.14 Å resolution map as estimated by gold-standard FSC at a 0.143 threshold ($VirB4_{unbound}$ trimer of dimers C1 4.14 Å map; Extended Data Fig. 6b and Supplementary Table 1a). This map clearly shows a trimer of dimers of VirB4, with one dimer better defined in the electron density. This led us to focus the refinement on the dimer. Local and non-uniform refinement[49] using a mask encompassing the so-defined dimer was performed, yielding a map with an average resolution of 3.49 Å as estimated by gold-standard FSC at a 0.143 threshold ($VirB4_{unbound}$ dimer C1 3.49 Å; Extended Data Fig. 6a and Supplementary Table 1a). This map was sharpened in Phenix v1.18.

## Model building and refinement of T4SS structure

An O-layer homology model (generated using Robetta[54] consisting of the hetero-trimeric unit of VirB7, $VirB9_{CTD}$ and $VirB10_{CTD}$ was initially fitted as a rigid-body into the asymmetric unit of the 2.58 Å resolution C14 map of the corresponding region (Extended Data Fig. 3c) using Chimera v1.4[55]. Individual residues were rebuilt into density using Coot v0.9.3[56], and the resulting structure refined using Phenix. 14 copies of this model were then manually fitted into the O-layer map using Chimera to generate a model of the entire O-layer, which was refined using Phenix with secondary structures and Ramachandran restraints applied (Supplementary Table 1b). The same procedure was used to produce the I-layer structure except that an I-layer homology model consisting of a hetero-dimeric unit of $VirB9_{NTD}$ and α1 of $VirB10_{NTD}$ was first obtained and the 3.08 Å resolution C16 map of the region (Extended Data Fig. 3d) was used. A model of $VirB9_{NTD}$ bound to α1 of $VirB10_{NTD}$ was obtained after several rounds of rebuilding and refinement. 16 copies of this model were fitted in the map to generate the entire model for the I-layer, which was rebuilt using Coot and refined using Phenix with secondary structural elements and Ramachandran restraints applied (Supplementary Table 1b).

For the stalk, the stalk C1 3.71 Å and stalk C5 3.28 Å (Extended Data Fig. 3h, i) maps were used to build de novo one VirB5 (residues 32 to 229) and one VirB6 (residues 1 to 272) using Coot, aided by secondary structure prediction (Psipred 4.0[57]). Five copies of each subunits were then generated to obtain the entire stalk structure. For 2 VriB6 subunits, the C1 map did not display clear density for the transmembrane region and therefore this region was omitted. Each residue of each molecule was then rebuilt/adjusted independently into the stalk C1 3.71 Å map density using Coot and the resulting structure refined using Phenix with secondary structures and Ramachandran restraints applied (Supplementary Table 1b).

For the IMC, 2 $VirB4_{unbound}$ structures (generated as described in next Methods section) were used and fitted into the IMC protomer C1 3.75 Å map (Extended Data Fig. 3g), rebuilt in density (Coot) and refined (Phenix). 1 VirB3 (residues 1 to 104) and 3 $VirB8_{tails}$ (residues 12 to 62) were build de novo using Coot guided by secondary structure prediction (Psipred) and transmembrane prediction (TMpred[58]). A final model of the IMC protomer structure was obtained after several rounds of rebuilding (Coot) and refinement with secondary structures and Ramachandran restraints applied (Phenix) (Supplementary Table 1b).

For the arches, the IMC-arches-stalk C1 6.18 Å map (Extended Data Fig. 3f) was used, the best map for this region. The resolution was however high enough to clearly show secondary structural features into which 9 homology models of the VirB8 periplasmic domain ($VirB8_{peri}$; residues 95-231; obtained using Robetta) were docked as rigid bodies using Chimera. The high cross-correlation (0.83) indicated the good correspondence between the map and the model. Confidence in the correctness of this model was increased considerably when it was realized that the $VirB8_{peri}$ domains come together in a manner that has been observed before (see main text). Note that the side chains are removed from our final model of the arches as they are not defined in the density.

Next, the structural composite model of the entire IMC and arches hexamer was obtained using the six-fold symmetry operators derived from computing a new map as described in Extended Data Fig. 5. In brief, the same workflow that was used to generate the ab initio model for the IMC-arches-stalk (see Extended Data Fig. 2c) was used except that C6 symmetry was applied during ab initio classification. This resulted in a C6 map (IMC-arches C6 8.33 Å map; Extended Data Fig. 5a and Supplementary Table 1a) from which the symmetry operators could be inferred using Phenix. This map also displayed density for the bottom of the I-layer and was therefore used to position the OMCC relative to the IMC-arches-stalk (Extended Data Fig. 5b) and generate a composite model for the entire T4SS. The positioning was checked using the 2D

classes (circa 2.4% of particles) where the OMCC and the IMC–arches–stalk are aligned (Extended Data Fig. 5c).

For all models, regions with poor C-α backbone density were subsequently deleted while side chains were removed for areas with poor side chain densities. MolProbity v4.5.1[59] was used to evaluate the quality of all structures. All data and model statistics are reported in Supplementary Table 1b.

## Model building and refinement of VirB4$_{unbound}$ structures

A homology model of the C-terminal domain of VirB4 (residues 400–760) was first generated with Phyre[60] using the structure of the C-terminal domain of the VirB4 protein from *Thermoanaerobacter pseudethanolicus* (PDB 4AG5[39]) as the template. This homology model was fitted as a rigid body into the corresponding region of the 3.49 Å map of the dimer using Chimera v1.13.1 and adjusted/rebuilt into the density using Coot. Next, the remaining C-terminus (residues 760–823) as well as the entire N-terminal domain (residues 15–400) for which the structure was unknown were built de novo, aided by secondary structure elements predicted by Psi-Pred and models generated by Robetta. The resulting models of the completed C- and N-terminal domains were combined and RosettaCM[61] was used to generate a full-length model docked into the density. The monomer model was improved further using iterative rounds of RosettaCM and manual readjustment in Coot against the map and refined using real space refinement with simulated annealing and secondary structure restraints in Phenix v1.18. A second copy of this model was rigid body fitted into the density corresponding to the adjacent subunit to generate the dimer model which was then adjusted/rebuilt in Coot and refined using Phenix.

The two other VirB4$_{unbound}$ dimers and HCP (PDB 1Y12[62]) were independently fitted as rigid bodies into the 4.11 Å map and adjusted using RosettaCM. The GSGSGS-linker connecting VirB4$_{unbound}$ subunits to HCP was built. The resulting trimer of dimers model was refined using Phenix.

Regions with poor Cα backbone density were subsequently deleted while areas with poor side chain densities were mutated to polyalanine. MOLPROBITY v4.4 was used to evaluate the quality of the structures. All data and model statistics are reported in Supplementary Table 1b.

## Model of the T4SS with pilus and VirB11 bound

Docking of the pilus base layer of the F TraA/VirB2 pentamer (PDB entry code 5LER)[32] was carried out using the shape-complementarity software Patchdock[40] using default parameters. The top-scoring structure was then used to position the entire F pilus on top of VirB6. The VirB5 structure was then fitted on top of the pilus using the shape-complementarity software HDOCK[63]. The model for VirB4–VirB11 was generated using AlphaFold1[36]. Finally, to generate an open O-layer structure, the outer membrane helices of VirB10$_{CTD}$ were manually moved using Chimera and the residues in the two linkers between these helices and the central barrel structure were rebuilt using Coot and energy minimized using YASARA yielding a *z*-score of −1.63[64].

## Interaction analysis and representation of the T4SS structure

Interaction analysis was conducted using the PISA server[65], and structure representations were generated using ChimeraX v1.1[55] and PyMOL v2.3.2[66]. Details of alignment and interactions are shown in the supplementary information.

## Identifying and aligning T4SS components for co-evolution studies

Starting from the T4SS components of R388 plasmid, the homologues of each protein encoded by nucleotide sequences in the European Nucleotide Archive database and the Integrated Microbial Genomes and Microbiomes database of the Joint Genome Institute were identified. Six rounds of iterative HMMER[67] search with e-value cut-offs of $10^{-12}, 10^{-12}, 10^{-12}, 10^{-12}, 10^{-6}$ and $10^{-3}$, respectively, were used. Homologues

found in each round of sequence search were used to construct the sequence profile for each T4SS protein using Hmmer hmmbuild, which was used to identify more homologues in the next round. We filtered the homologues found in the last round of database search by their coverage (>60%) over the query sequence and recorded their loci on the nucleotide sequences.

For each pair of the T4SS proteins, their homologues that are encoded on the same nucleotide sequences and separated by less than 20 coding genes were extracted. This requirement ensures that we include protein pairs that are encoded by genes close to each other in the bacterial genome, that is, probably on the same T4SS operon and thus function together. The sequences of these protein pairs were concatenated and the multiple sequence alignment (MSA) was derived from the pairwise sequence alignments made by HMMER. The MSA was then filtered for each T4SS protein pair by sequence identity (maximal identity for remaining sequences ≤90%) and gap ratio in each sequence (gap ratio ≤25%), and the resulting non-redundant MSA was used for co-evolution analysis. The number of sequences in the MSA ranges from 213 to 8571, with an average of 2,809. Most protein pairs have more than 1,000 sequences in the MSA, which is sufficient for accurate co-evolution analysis according to our previous study[28].

## Co-evolution analysis of T4SS components and validation of the T4SS complex structure

The MSA for each protein pair was analysed by TrRosetta.v1 to infer interacting residues from coevolutionary patterns in the MSA. TrRosetta is a deep learning network trained on tens of thousands of proteins in the PDB to convert the coevolution patterns detected in the MSA of a protein and its homologues to residue-residue distances. TrRosetta predicts the probability distribution for residue-residue distances in a set of distance bins. We summed the probability for bins corresponding to distance ≤12 Å to obtain the contact probability score between residues.

We ranked these predicted inter-protein contacts according to the TrRosetta contact probability scores. Residue pairs with TrRosetta score ≥ 0.05 for each protein pair mentioned below are shown in the supplementary information (a cutoff of 0.05 returns between 315 (for VirB5-VirB6) and 896 (for VirB3-VirB4) pairs). We mapped the top-ranking residues onto the T4SS complex structure for protein pairs that directly interact in the experimental structure: VirB3–VirB4, VirB5–VirB6, VirB4–VirB8 and VirB9–VirB10. For VirB5–VirB6, VirB4–VirB8 and VirB9–VirB10, we mapped residue pairs with contact probability scores above a threshold of 70% of the score of the top scoring pair (Fig. 3e,f, Extended Data Fig. 9b,c and Supplementary Table 4). This arbitrary threshold is used for illustration purposes since it results in about 15–25 residue pairs (a manageable number for the reader) being displayed in the figures. For VirB3–VirB4 (Extended Data Fig. 9a and Supplementary Table 4), such a threshold would have resulted in too many residue pairs (102 in total) to show, so we mapped the top 30 co-evolving residue pairs for this interface. For T4SS components that are not present in the cryo-EM structure, that is, VirB11 or VirB2, their co-evolution with VirB4 and VirB3/VirB6, respectively, were analysed. For the VirB4–VirB11 interaction, we obtained a model of the complex for both VirB4–VirB11 and TraB–TraG using AlphaFold1[36] (see above and Extended Data Fig. 10a,b) and this model was used to map the top TrRosetta co-evolving pairs (70% threshold as above; Extended Data Fig. 10b and Supplementary Table 4). For the VirB2–VirB6 or VirB2–VirB3 interactions, we mapped the VirB6 or VirB3 residues listed in the 50 top TrRosetta corresponding residue pairs onto the structure of VirB6 (Extended Data Fig. 10i and Supplementary Table 4) or VirB3 (Extended Data Fig. 4j and Supplementary Table 4), respectively.

## Reporting summary

Further information on research design is available in the Nature Research Reporting Summary linked to this paper.

## Data availability

EM maps and atomic models were deposited to the Electron Microscopy Data Bank (EMDB) and Protein Data Bank (PDB) databases. Accession codes can be found in Supplementary Table 1 of the manuscript. PDB codes for the various structures reported in this manuscript are 7O3J, 7O3T, 7O3V, 7O41, 7O42, 7O43, 7OIU, 7Q1V and the EMDB accession codes are EMD-12707, EMD-12708, EMD-12709, EMD-12715, EMD-12716, EMD-12717, EMD-13765, EMD-13766, EMD-13767, EMD-13768 and EMD-12933. All constructs (wild type and mutants) used in this study can be obtained on request to G.W. A Chimera session highlighting the unaccounted densities observed in the IMC–arches–stalk C1 6.18 Å unsharpened map is provided in the supplementary information. Source data are provided with this paper.

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

**Acknowledgements** This work was supported by Wellcome grants 098302 and 217089 to G.W., a fellowship from Washington Research Foundation and start-up fund from Southwestern Medical Foundation fellowships to Q.C., and Yeast Program grant 5 P41 GM 103533-24 to D.B. Cryo-EM data for this investigation were collected at the ISMB EM facility at Birkbeck College, University of London with financial support from Wellcome Trust (202679/Z/16/Z and 206166/Z/17/Z). We thank D. Houldershaw for IT support.

**Author contributions** K.M. and A.K.V. purified the T4SS complex and VirB4_unbound, respectively. N.L. and A.K.V. prepared grids of the T4SS complex and VirB4_unbound, respectively. N.L. collected the data. EM processing work on the T4SS was carried out by K.M. and K.M. built and refined the structure. A.K.V. processed the VirB4_unbound data and built and refined the structure. T.R.D.C., M.U., N.B. and A.R. were involved in grid making and data collection in the initial phase of the project. E.V.O. with T.R.D.C., M.U., N.B. and A.R. were involved in data processing in the initial phase of the project. F.L. was involved in cloning and purification of VirB4_unbound in the initial phase of the project. D.B. and Q.C. generated and interpreted the co-evolution data. A.R. and A.K.V. made mutations in VirB6 and TraB–TraG and tested these mutants, respectively. A.K.V. and C.O. were involved in the purification of the TraB–TraG. G.W. supervised the work and wrote the manuscript. All authors read the manuscript and discussed the materials.

**Competing interests** The authors declare no competing interests.

**Additional information**
**Correspondence and requests for materials** should be addressed to Kévin Macé, Qian Cong or Gabriel Waksman.

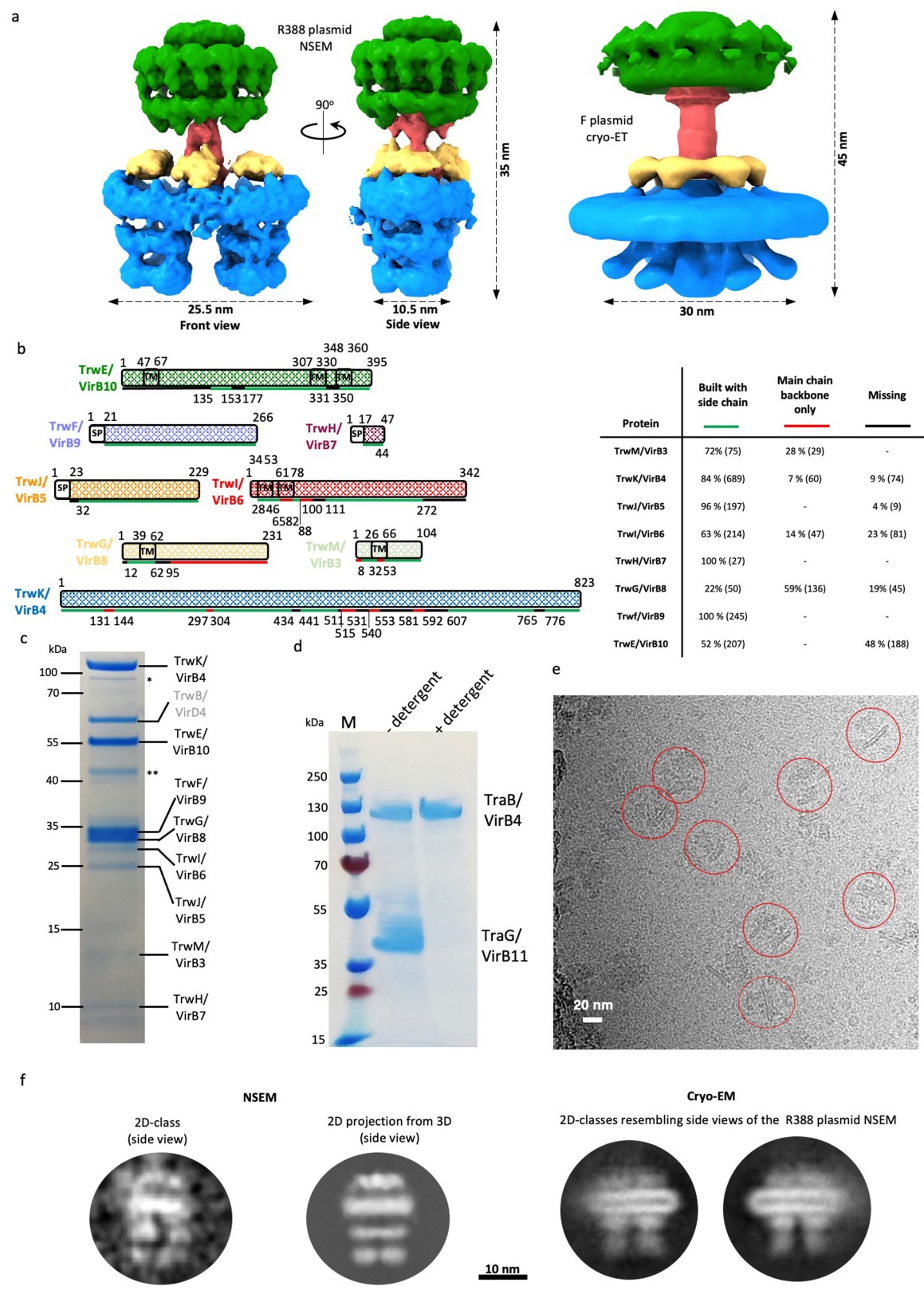

**Extended Data Fig. 1** | See next page for caption.

**Extended Data Fig. 1 | Prior knowledge of conjugative T4SS architectures and proteins, purification and cryo-EM analysis of the R388 T4SS.** a, Known 3D architectures of conjugative T4SSs. Left and middle: front and side views of the low resolution negative-stained EM structure of the R388 T4SS (EMD-2567)[8]. Right: low resolution cryo-electron tomography structure of the F T4SS (EMD-9344 and EMD-9347)[10]. The IMC, Stalk, Arches, and OMCC are colour-coded as in Fig. 1b. b, Primary structure of the Trw/VirB proteins observed in this study. Each protein is named TrwX/VirBX according to convention where the first name is that of the Trw protein in the *trw* R388 plasmid gene cluster, and the second name is the name of its homologue in the *Agrobacterium* system. Each protein is shown as a hashed rectangle, the length of which is proportional to the length of its sequence. Colour-coding for each protein is as in Fig. 1d. Transmembrane segments as observed in the structure are shown in boxes labelled "TM". Signal sequences are shown in empty boxes labelled SP. Lines under the rectangles indicate the parts of the sequence for which the electron density was of high enough quality to build a complete atomic model including side chains (in green), or where the secondary structures definition was good enough to build main chain secondary structures but not side chains (in red), or was so poor that no model could be built (in black). Boundary residue numbering for each box and line are indicated. Table at right recapitulates, for each protein, the proportion of the sequence either built with side chains, or with built main chain only, or missing in our structure. c, SDS-PAGE analysis of the purified R388 T4SS (For gel source data, see Supplementary Fig. 1a and corresponding legend in SI guide; n = 9 independent experiments). Molecular weight markers are indicated on the left. The proteins bands (identified by mass spectrometry) are labelled on the right. * and ** indicate minor contaminants (OmpA and OmpC). We purified a T4SS complex as in Redzej et al. (2017)[9] i.e. a complex composed of 9 of the essential Trw/VirB proteins, TrwM/VirB3-TrwE/VirB10 and TrwB/VirD4 (Extended Data Fig. 1c), except that 1- the His-tag column purification step to enrich the preparation with VirD4-bound complexes was not carried out, 2- the complex was not crosslinked, a step shown by Redzej et al. (2017)[9] to be required to keep TrwB/VirD4 bound, 3- concentration of the complex was achieved using ultracentrifugation, and 4- potential aggregates resulting from concentration were separated using sucrose density gradient centrifugation (see Methods). TrwB/VirD4 was therefore absent from the structure as it needs crosslinking to remain bound[9]. Association of VirD4 with the T4SS may need stabilising through its interaction with the DNA substrate or its dissociation might be triggered by the cryogenic conditions. TrwD/VirB11 was also not part of the complex since it dissociates in the presence of detergents (Extended Data Fig. 1d). As a result, the structure contains TrwM/VirB3-TrwE/VirB10, a complex which was previously examined by negative stain EM (NSEM) by Low et al. (2014)[8] and found to adopt the double-barrelled architecture. However, the purification protocol used to purify the complex here was greatly modified compared to that used in Low et al. (2014)[8]. Crucial to the improved conditions used here was the use of TDAO, a zwitterionic detergent. Moreover, concentration and separation from aggregates were also changed: in Low et al. (2014)[8], the complex could not be concentrated without heavily aggregating, while here, due to a change in detergents mix, concentration using ultra-centrifugation was achieved as well as removing of minority aggregates by sucrose density gradient centrifugation. As a result of these significant modifications in the purification protocol, the yields were greatly improved (assessed to being between 20-30-fold), the complex being also much less prone to aggregation. Improved yields and higher quality sample combined to make the determination of this complex structure by cryo-EM possible. As explained in main text, a minority of particles in the cryo-EM data set presented here display the typical side views of the double-barrelled structure, indicating that the majority hexamer of dimers architecture we observe here must have been unstable in the buffer and NSEM conditions used by Low et al. (2014)[8] since they did not observe it. In contrast, yields and stability of the double-barrelled complex obtained by Low et al. (2014)[8] were low, making it impossible to solve its cryo-EM structure. d, SDS-PAGE analysis of the purified TraB/VirB4-TraG/VirB11 complex in the absence (No detergent) or presence of the detergents used to extract the T4SS (+detergent). For gel source data, see Supplementary Fig. 1b and corresponding legend in SI guide. n=3 independent experiments. e, Cryo-EM micrograph of the R388 T4SS. Red circles indicate examples of particles. 104,711 such micrographs over 7 datasets were collected. f, 2D classes found in the cryo-EM data set that show a view similar to that of the side views of the double-barrelled architecture observed by Low et al. (2014)[8]. The double-barrelled structure is characterised by a unique side view shown in Extended Data Fig. 1a, middle panel. Therefore, we asked whether such side views could be found in the cryo-EM data set described here. Left: an example of side view 2D classes (labelled "2D-class") typically found in the NSEM double-barrelled architecture data and a corresponding 2D projection from the NSEM double-barrelled map (labelled "2D projection from 3D"). Right: 2 examples of similar side views but in the cryo-EM data set presented here. These 2D classes were generated using 2D classification of the set of 1,292,734 particles mentioned in Extended Data Fig. 1i, and selected for their resemblance to the side-view projections shown at left, resulting in the final selection of about 4,838 particles, i.e. circa 0.3% of the data set.

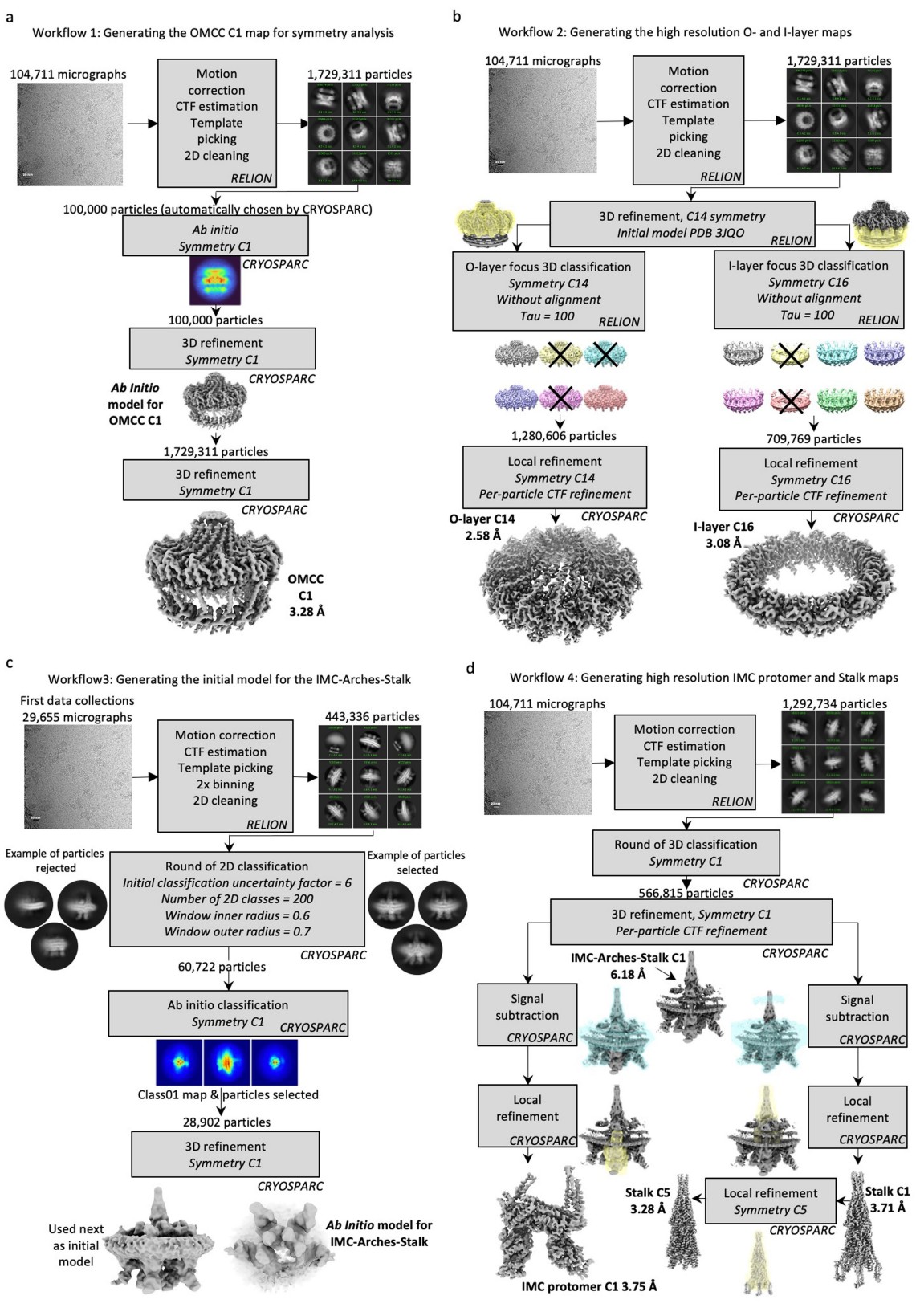

**Extended Data Fig. 2 | Workflow used to determine the structure of the various T4SS sub-complexes.** a, Workflow used to generate the OMCC C1 *Ab Initio* and 3.28 Å map. These maps were used for symmetry analysis of the OMCC O-layer and I-layer. b, workflow used to generate the O-layer C14 2.48 Å map and the I-layer C16 3.08 Å map. These maps were used to build and refine the atomic O- and I-layer models, respectively. c, workflow used to generate the initial *Ab Initio* model for the IMC-Arches-Stalk map. d, workflow used to generate the IMC-Arches-Stalk C1 6.18 Å map which was used 1- to build the Arches model, 2- for symmetry analysis of the IMC, the Arches, and the Stalk, 3- to derive the IMC protomer C1 3.75 Å map (used to build the IMC protomer model), 4- to derive the Stalk C1 3.71 Å and the Stalk C5 3.28 Å maps (used to build the Stalk model).

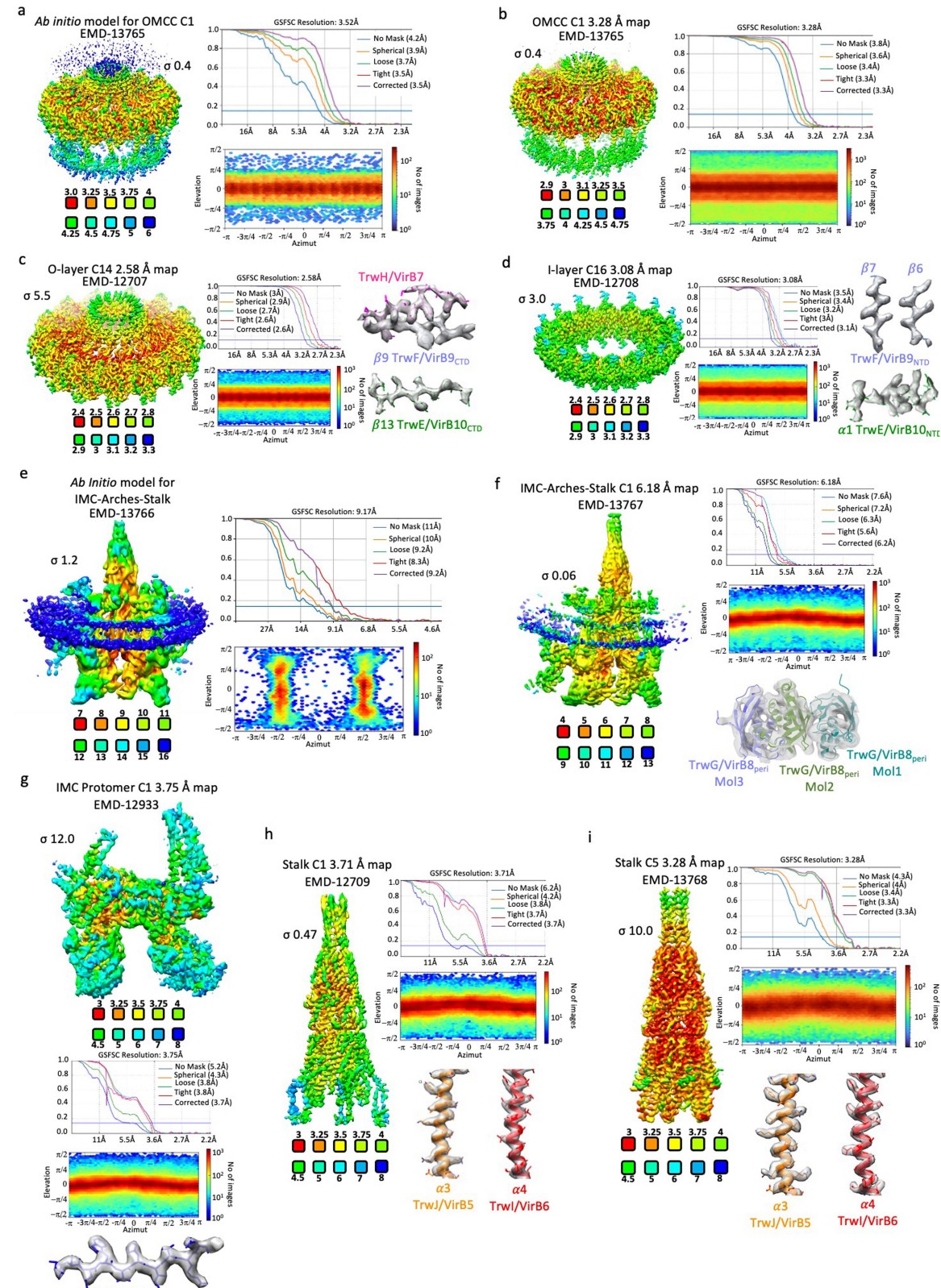

**Extended Data Fig. 3 | Cryo-EM maps used in this study.** For each map, the density coloured by local resolution, the average resolution derived from Fourier Shell Correlation (FSC) and, for panels c,d,f-i, a representative region of the electron density map with the final model of the T4SS built in it (in stick representation colour-coded as in Fig. 1d) are shown. Local resolution was calculated using CRYOSPARC (FSC cut-off 0.5) and coloured as indicated in the scale below the map. For each map, FSC plots show curves for correlation between 2 independently refined half-maps with no mask (blue), spherical mask (green), loose mask (red), tight mask (cyan) and corrected (purple). Cut-off 0.143 (blue line) was used for resolution estimation. a, the *Ab Initio* model for the OMCC; b, the OMCC C1 3.28 Å map; c, the O-layer C14 2.58 Å map; d, the I-layer C16 3.08 Å map; e, the *Ab Initio* model for the IMC-Arches-Stalk; f, The IMC-Arches-Stalk C1 6.18 Å map; g, the IMC protomer C1 3.75 Å map; h, the Stalk C1 3.71 Å map and i, the Stalk C5 3.28 Å map. All maps are sharpened. Contour levels and EMD codes are indicated.

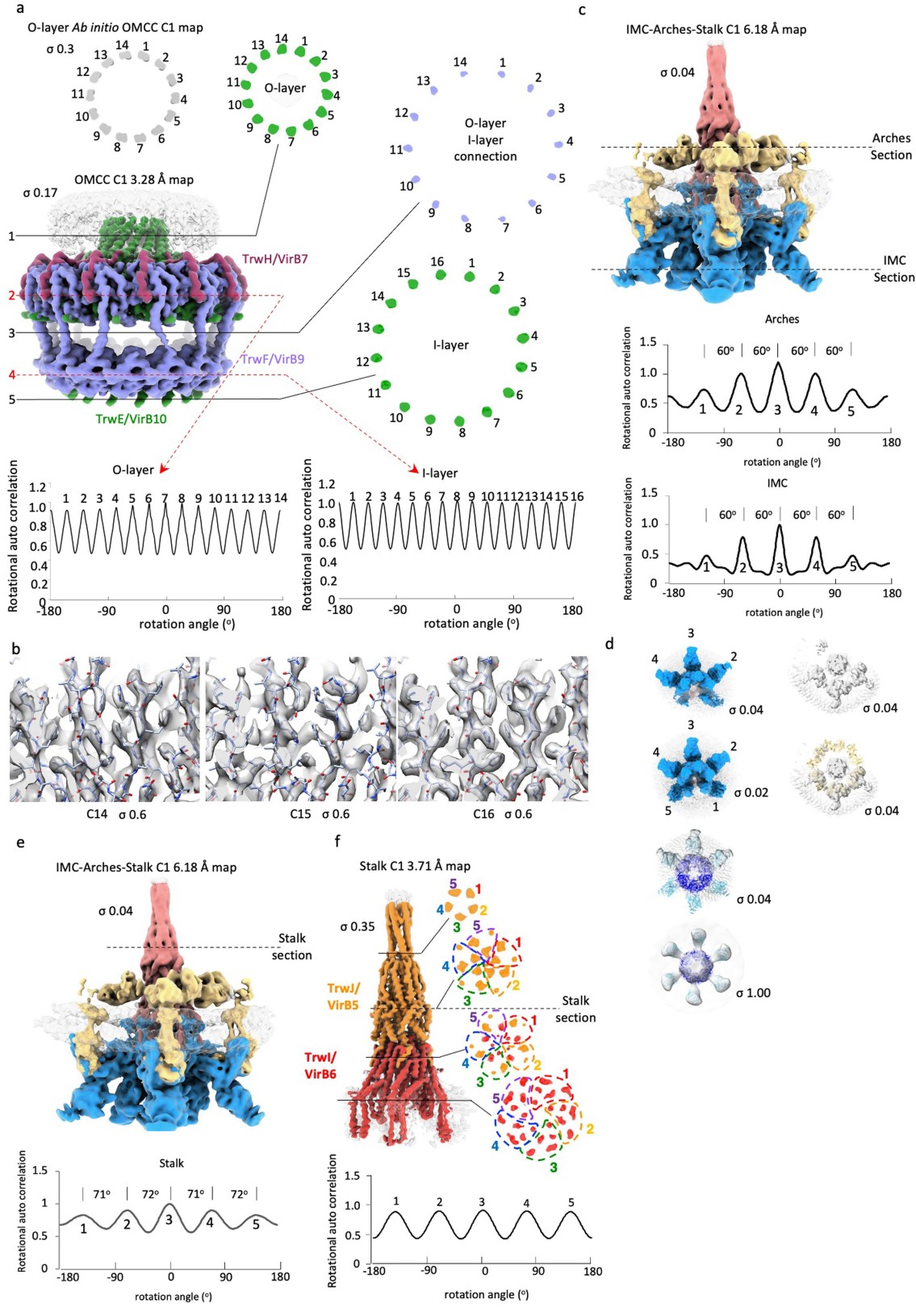

**Extended Data Fig. 4 |** See next page for caption.

**Extended Data Fig. 4 | Symmetry analysis of the various T4SS sub-complexes.** a, Symmetry analysis of the OMCC C1 *Ab Initio* and 3.28 Å maps. Upper left: section of the *Ab Initio* OMCC C1 map. The section is taken through the helical trans-membrane region of the O-layer. Middle Left: the non-averaged and unsharpened C1 map of the OMCC determined at a resolution of 3.28 Å colour-coded as in Fig. 1d. The contour level is indicated. The three filled lines and two dashed arrows labelled 1 to 5 indicate where the map sections have been taken for analysis shown at right (filled lines) and at bottom (dashed lines in red). Right: three sections of the map shown at left, one for the O-layer, one for the O-layer/I-layer connection, and one for the I-layer. Bottom: IMAGIC rotational auto correlation analysis of sections 2 and 4, independently corroborating C14 and C16 symmetry for the O- and I-layer, respectively. b, I-layer symmetry test. Using the OMCC C1 3.28 Å map (Extended Data Fig. 3b) as reference, three maps were generated using local refinement with a mask encompassing the I-layer and applying either C14, C15, or C16 symmetry. Highest interpretability is observed when C16 is used. The contour levels are indicated. The final model is shown in stick representation color-coded by atoms (red, blue, and white for oxygen, nitrogen, and carbon, respectively). c, Symmetry analysis of the IMC and Arches. Left: IMC-Arches-Stalk C1 6.18 Å unsharpened map colour coded as in Fig. 1b. The contour level is indicated. Two dashed lines through the Arches and the IMC indicate where the map sections have been taken for further symmetry analysis using IMAGIC shown underneath. However, for the Arches section, the section contains both the Arches and the Stalk and therefore, the Stalk region of the section was excluded from the analysis. The symmetry analysis of the IMC shows 5 peaks separated by a ~60° angle indicating 5 IMC protomers organised along a hexagon. A 6th protomer has very weak occupancy: correspondingly, a weak but visible 6th peak is observed in the IMAGIC symmetry analysis. Similar conclusions can be drawn from the analysis of Arches symmetry. d, Further symmetry analysis of the IMC and Arches. Left panels from top to bottom: bottom view of the IMC-Arches-Stalk C1 6.18 Å map coloured in blue and contoured at 0.04 σ level; same map contoured at 0.02 σ level; IMC hexameric model fitted into the same map (now coloured in semi-transparent grey); IMC hexameric model fitted into the cryo-ET map of the F plasmid IMC (EMD-9347 coloured in semi-transparent grey and contoured at 1 σ level; fitting correlation = 0.78). Right panels from top to bottom: section of the Arches and stalk density (0.04 σ level) in the IMC-Arches-Stalk C1 6.18 Å map coloured in semi-transparent grey; fit of the hexameric Arches into the external ring density. e, Symmetry analysis of the Stalk region in the unsharpened IMC-Arches-Stalk C1 6.18 Å map. The dashed line through the Stalk (in red) indicates where the map section has been taken for the symmetry analysis using IMAGIC shown underneath. Regularly spaced peaks spaced by a circa 72° are observed indicating C5 symmetry. f, Symmetry analysis of the non-averaged and unsharpened Stalk C1 3.71 Å map (colour coded orange for TrwJ/VirB5, red for TrwI/VirB6). The four lines indicate where the map slabs/slices shown at right have been taken. At right: four sections of the map shown at left, two for the TrwJ/VirB5 Stalk tip, one for the TrwI/VirB6 Stalk base and one in the middle. For one section, the symmetry analysis using IMAGIC is shown in the lower panel.

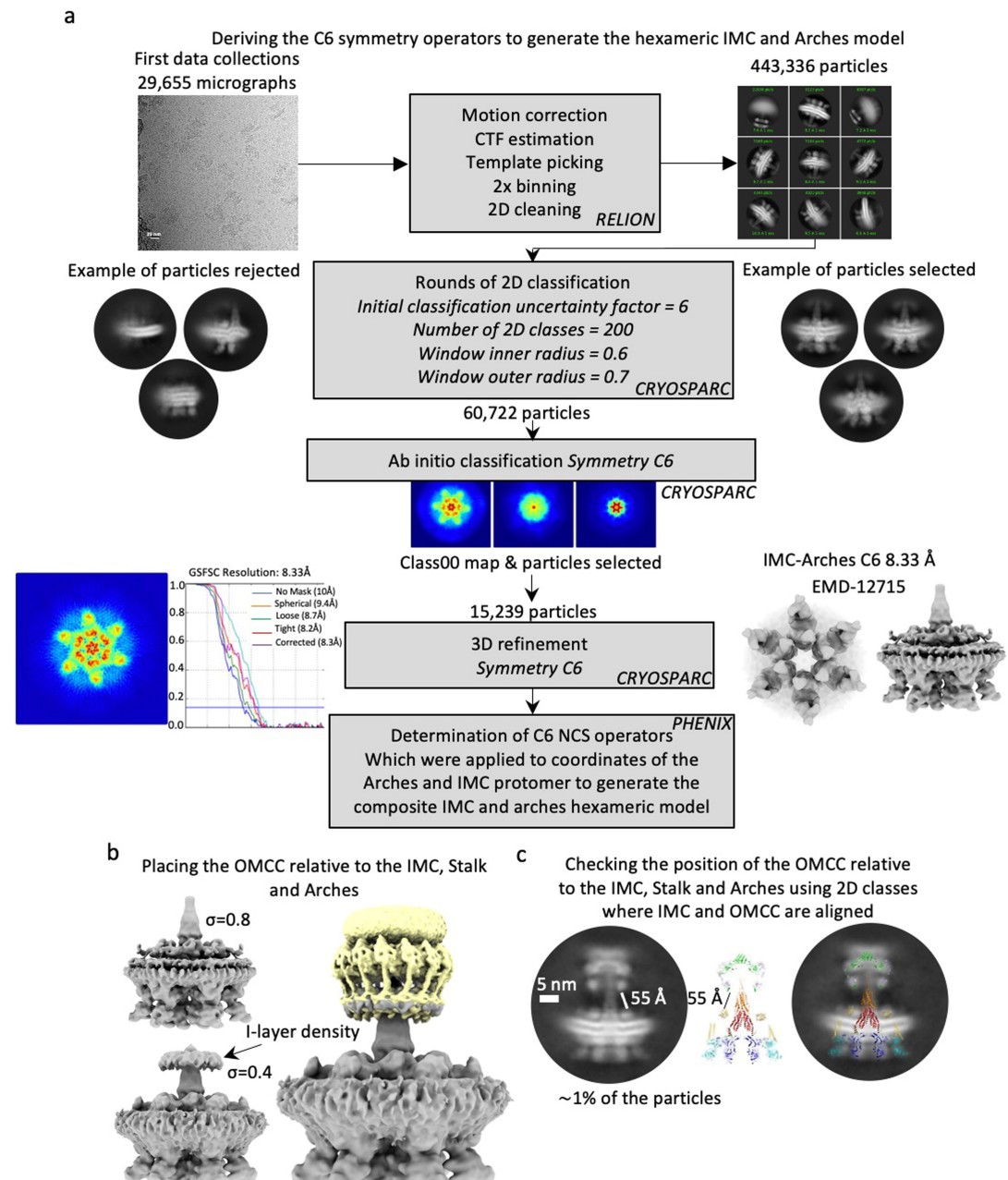

**Extended Data Fig. 5 | Generating the composite T4SS model shown in Fig. 1d.** a, Derivation of the C6 symmetry operators. The same particles used to generate the *Ab Initio* model for the IMC, Arches, and Stalk (Extended Data Fig. 2c) were used to generate an *Ab Initio* model with C6 symmetry imposed. The resulting map was used to generate the C6 symmetry operators using PHENIX. b, Positioning the OMCC relative to the IMC, Arches, and Stalk. As explained in Methods, the same C6 map was used to position the I-layer part of the OMCC. c, Verifying the positioning of the OMCC relative to the IMC, Arches and Stalk. While in most 2D classes, the OMCC is not aligned with the IMC-Arches-Stalk, in about 2.4 % of the particles, it is. Using these 2D classes, the distance (reported here) between the Arches and the I-layer was used to check that the positioning of the OMCC relative to the rest of the structure is correct.

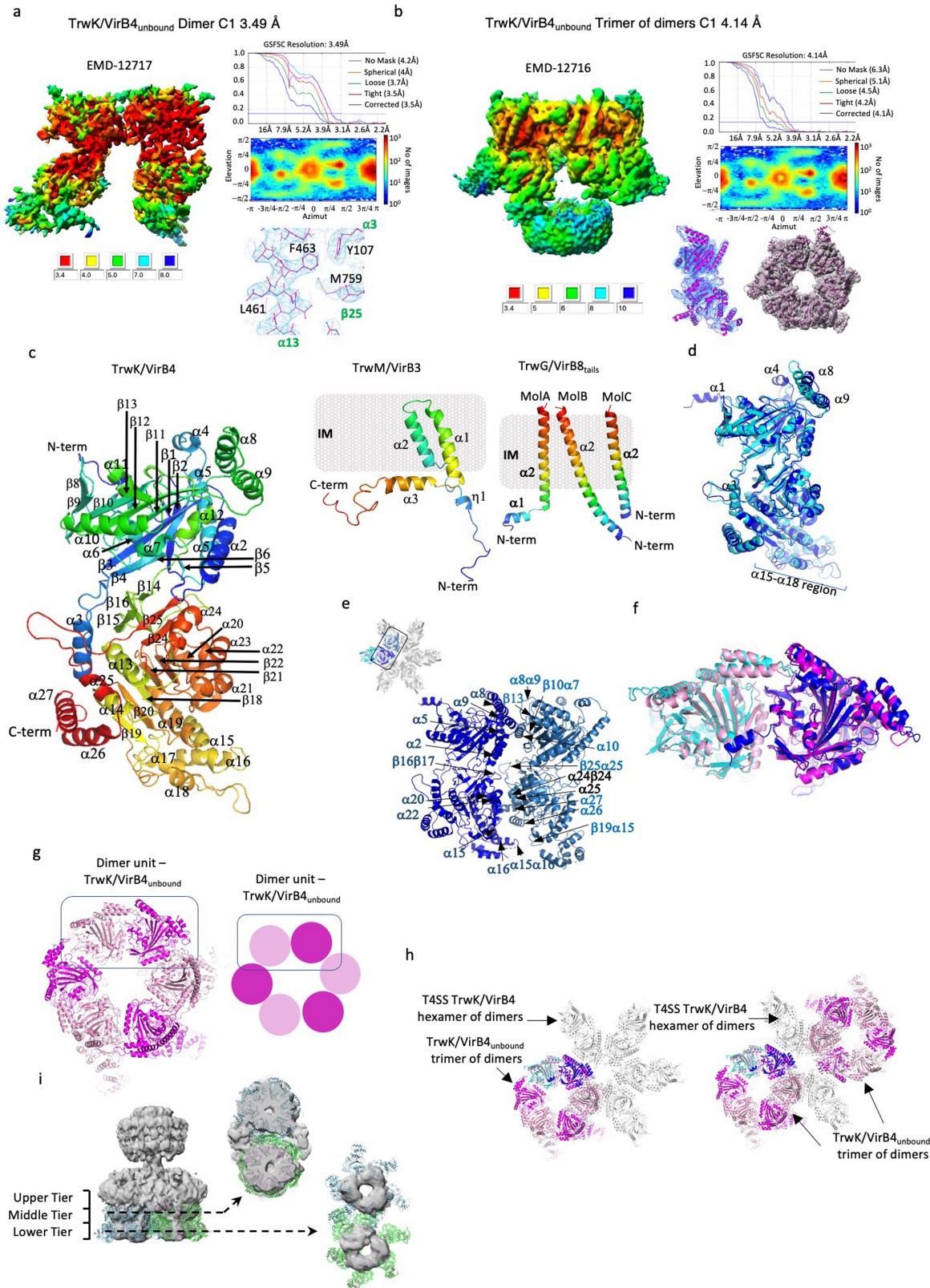

**Extended Data Fig. 6** | See next page for caption.

**Extended Data Fig. 6 | Structural details of TrwK/VirB4$_{unbound}$ and T4SS TrwK/VirB4 and IMC protomer.** a and b, Assessment of map resolution and quality for the TrwK/VirB4$_{unbound}$ dimer (a) and trimer of dimers (b) structures. The local resolution variations of the corresponding map (left), the overall resolution derived from Fourier Shell Correlation (FSC) (upper right), the angular distribution (middle right) and a representative region of the electron density map with the final model of the TrwK/VirB4$_{unbound}$ model built in it (lower right) are shown. Local resolution was calculated using CRYOSPARC and coloured as indicated in the scale below the map. The FSC plot shows curves for correlation between 2 independently refined half-maps with no mask (blue), spherical mask (green), loose mask (red), tight mask (cyan) and corrected (purple). Cut-off 0.143 (blue line) was used for average resolution estimation. The final model in representative regions of the maps is shown in magenta stick and ribbon representation for the dimer and trimer of dimers, respectively. c, Secondary structure definition of TrwK/VirB4 (Left), TrwM/VirB3 (Middle) and TrwG/VirB8$_{tails}$ (right). The ribbon for each protein is coloured in rainbow colours from dark blue for the N-terminus to red for the C-terminus. All secondary structures are labelled. The IM is shown as a grey rectangle. d, Superposition of TrwK/VirB4$_{central}$ and TrwK/VirB4$_{outside}$ subunits of the IMC protomer. The two structures are very similar (RMSD in Cα position of 1.6 Å). Regions of differences between the two structures are indicated. e, Details of the interactions between two adjacent TrwK/VirB4$_{central}$ subunits within the central hexamer. The two subunits are both shown in ribbon but coloured dark and sky blue, respectively. All secondary structures involved in the interaction are shown. Without the structure of ATP-bound TrwK/VirB4, it is unclear whether the TrwK/VirB4$_{central}$ hexamer is in an active form or conformational changes are required to transition into one. f, Superposition of the TrwK/VirB4 dimeric unit of the T4SS (TrwK/VirB4$_{central}$ and TrwK/VirB4$_{outside}$ in blue and cyan, respectively, as in Fig. 2) onto the dimeric unit of TrwK/VirB4$_{unbound}$ (in

magenta and pink). These two structures superimpose very well with an RMSD in Cα position of 1.2 Å. g, Assembly of TrwK/VirB4$_{unbound}$. In TrwK/VirB4$_{unbound}$, three dimer units (shown here in magenta and pink, one of which is surrounded by a rectangle) come together in a roughly head to tail manner to form a trimer of dimers. Left: top view of the trimer of dimers structure. Right: schematic diagram showing the trimer of dimers configuration of TrwK/VirB4$_{unbound}$. h, Superposition of the TrwK/VirB4$_{unbound}$ trimer of dimers (in magenta and pink) and the T4SS TrwK/VirB4 hexamer of dimers (in grey except for the TrwK/VirB4 dimer used for superposition which is shown in cyan and blue for TrwK/VirB4$_{outside}$ and TrwK/VirB4$_{central}$, respectively). Left: the two types of assembly are superposed using the superposed dimeric units shown in panel f as a guide. In this superposition, the TrwK/VirB4$_{unbound}$ trimer of dimers can be observed in an off-centered position relative to the T4SS TrwK/VirB4 hexamer of dimers. Right: two of the TrwK/VirB4$_{unbound}$ trimers of dimers superposed on two diametrically opposite TrwK/VirB4$_{central}$-TrwK/VirB4$_{outside}$ dimers results in a double-barrelled architecture reminiscent of that observed in the NSEM double-barrelled structure by Low et al. (2014)[8] where two off-centered barrels (also trimers of dimers) were observed side by side as shown in Extended Data Fig. 1a, left panel. i, Docking of two TrwK/VirB4$_{unbound}$ trimers of dimers (green and blue ribbons) into the NSEM double-barrelled structure. Left: NSEM map of the T4SS double-barrelled architecture[8] contoured at σ 10 (EMD-2567). The region corresponding to TrwK/VirB4 is within the middle and lower tiers densities of each barrel (see Low et al. (2014) for details)[8]. The two dashed arrows indicate where the sections shown at right have been taken. The TrwK/VirB4$_{unbound}$ NTDs fit well into the middle tier density (fitting correlation 0.59; upper right panel) while the CTDs, which are very flexible because of being unconstrained, protrude out of the lower tier density (fitting correlation 0.29; lower right panel).

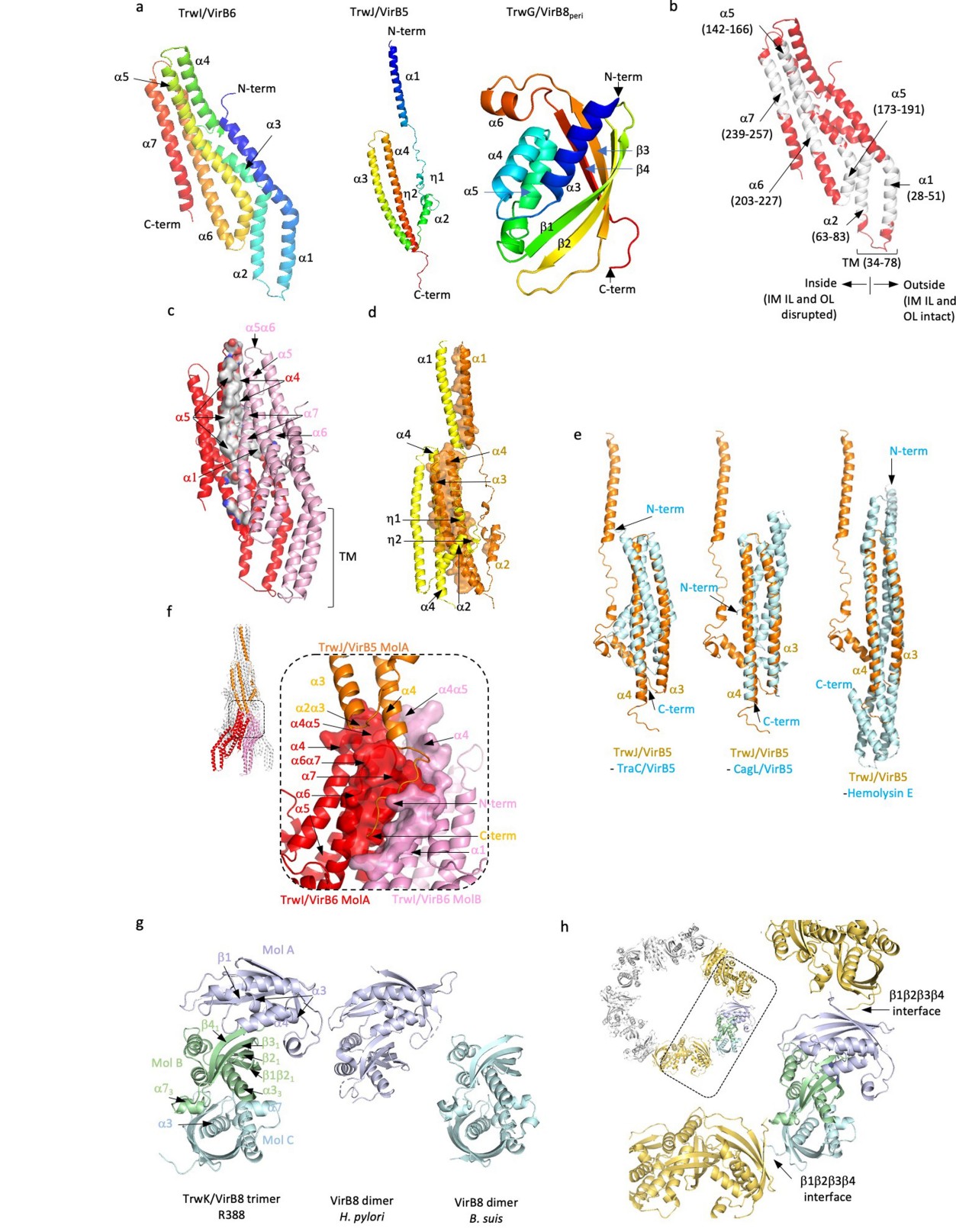

**Extended Data Fig. 7** | See next page for caption.

**Extended Data Fig. 7 | Details of Stalk and Arches proteins.** a, Secondary structure definition of TrwI/VirB6 (left), TrwJ/VirB5 (middle) and TrwG/VirB8$_{peri}$ (right). The ribbon for each protein is coloured in rainbow colours from dark blue for the N-terminus to red for the C-terminus. All secondary structures are labelled. b, Locations of the predicted hydrophobic TMs in TrwI/VirB6. The TMpred server[58] suggests a number of potential TMs in TrwI/VirB6: these predicted TMs are here mapped in white onto the TrwI/VirB6 structure. Of these, only two are observed inserting in the inner membrane, $\alpha1$ and $\alpha2$. All $\alpha$-helices that contain a hydrophobic region are labelled. Boundary residues for these regions and location of the TM region are indicated. Also, we observe that the region of the IM within the TrwI/VirB6 pentamer is disrupted while it remains intact outside it (indicated in the figure). c, Interactions between subunits within the TrwI/VirB6 pentamer. Two adjacent subunits are shown in pink and red ribbon, respectively. Interface residues in the subunit in red are shown as surface coloured in grey. All secondary structures where residues are involved in interactions are labelled. The positions of the TMs as defined in the electron density map are shown. d, Interactions between subunits within the TrwJ/VirB5 pentamer. Two adjacent subunits are shown in orange and yellow ribbon, respectively. Interface residues in the subunit in orange are shown in surface representation coloured in orange. All secondary structures where residues are involved in interactions are labelled. e, Superposition of TrwJ/VirB5 with two known VirB5 homologues and one pore-forming protein Hemolysin E. Left: superposition with TraC (PDB entry code: 1R8I[19]), the VirB5 homologue of the pKM101 plasmid-encoded T4SS. Middle: superposition with CagL (PDB entry code: 3ZCI[20]), the VirB5 homologue in the Cag pathogenicity island of *Helicobacter pylori*. Right: superposition with Hemolysin E (PDB entry code: 6MRU[68]), a bacterial pore-forming protein, one of top hits in DALI[69].

The superposition is particularly good with the C-terminal half of CagL and Hemolysin E (RMSD of 3.1 and 3.2 Å in C$\alpha$ atoms, respectively). f, Interactions between TrwJ/VirB5 and TrwI/VirB6. One TrwJ/VirB5 subunit (in orange ribbon) interacts with two TrwI/VirB6 subunits (shown in red and pink). For TrwI/VirB6, the regions of two subunits involved in interactions with TrwJ/VirB5 are shown as a surface while the rest of the molecules are shown in ribbon. Secondary structures contributing residues to the interfaces are labelled. g, Interactions between subunits within the TrwG/VirB8$_{peri}$ homo-trimeric unit. Left: the homo-trimeric TrwG/VirB8$_{peri}$ unit. Each subunit is shown in a different colour, pale cyan (MolC), pale green (MolB) and pale blue (MolA), respectively. Centre: the VirB8$_{peri}$ dimer from *H. pylori*[21] (PDB entry code: 6IQT). The orientation shown results from a superposition of this dimer on MolA/MolB of TrwG/VirB8$_{peri}$. As can be seen, the interface between subunits within this dimer is similar to that of the MolA/MolB interface between TrwG/VirB8$_{peri}$ subunits. Right: the VirB8$_{peri}$ dimer from *Brucella suis*[22] (PDB entry code: 2BHM). The orientation shown results from a superposition of this dimer on MolB/MolC of TrwG/VirB8$_{peri}$. As can be seen, the interface between subunits within this dimer is similar to that of the MolB/MolC interface between TrwG/VirB8$_{peri}$ subunits. RMSDs are reported in main text. Secondary structures contributing residues to the interfaces are labelled. h, Interface between TrwG/VirB8$_{peri}$ trimeric units in the Arches hexamer. In the T4SS structure presented here, six trimeric units come together to form the Arches. Inset: top view of the TrwG/VirB8$_{peri}$ trimeric units forming the Arches hexamer. The dashed lined box locates the region zoomed-in at right. One trimeric unit is colour-coded as in panel g, while the adjacent trimeric units are coloured in yellow orange. The secondary structures involved in interactions between trimeric units are labelled.

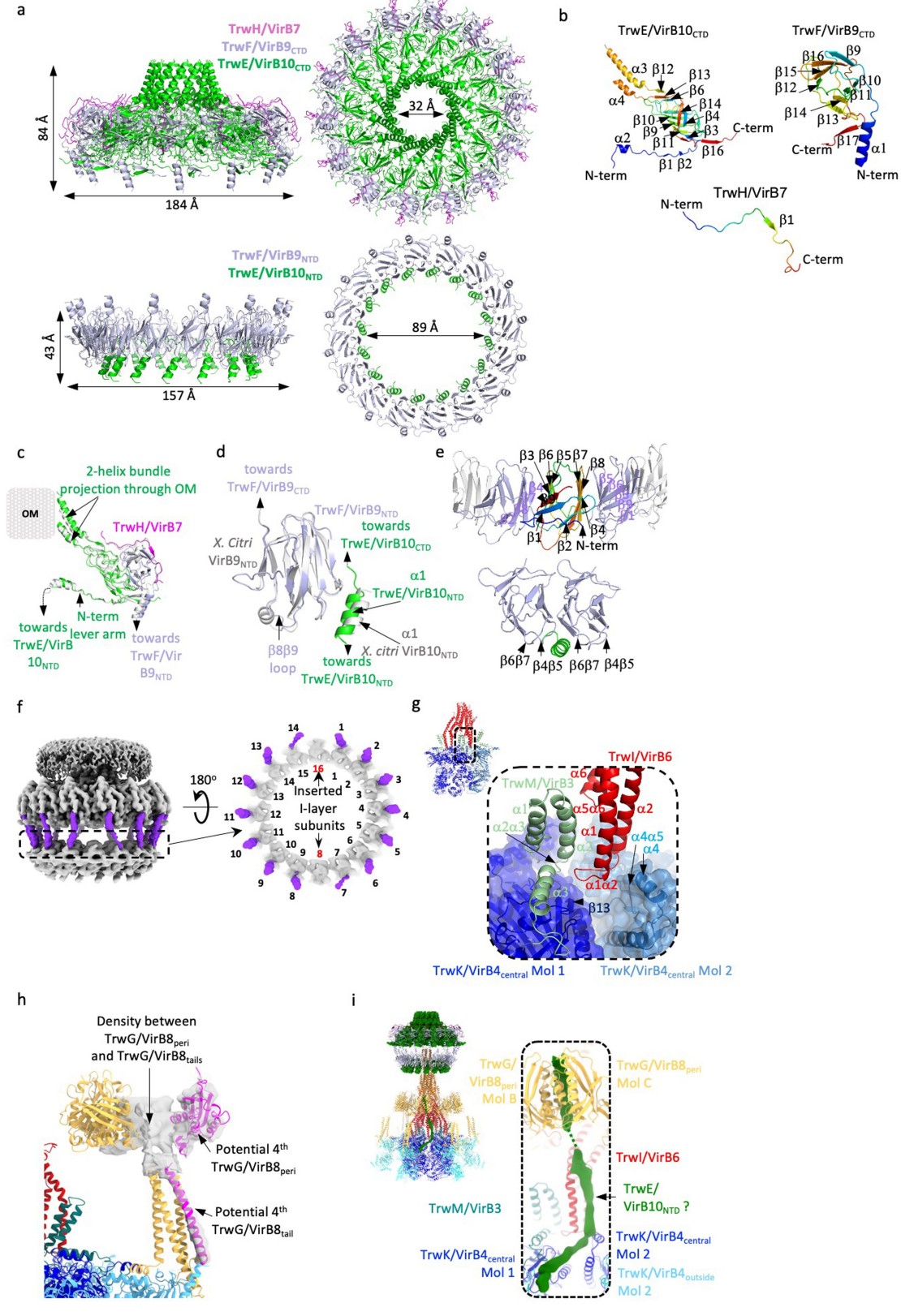

**Extended Data Fig. 8 |** See next page for caption.

**Extended Data Fig. 8 | Details of the OMCC proteins and description of extra-densities that could not be ascribed.** a, Structure of the O-layer and I-layer. All proteins are in ribbon, with TrwH/VirB7, TrwF/VirB9 and TrwE/VirB10 shown in magenta, light blue and green, respectively. Dimensions of interest are reported. Proteins constituting the shown complexes are indicated. Upper left: side view of the O-layer. Upper right: top view of the O-layer. Lower left: side view of the I-layer. Lower right: top view of the I-layer. b, Secondary structure definition of TrwE/VirB10$_{CTD}$ (left), TrwF/VirB9$_{CTD}$ (middle) and TrwH/VirB7 (right). The ribbon for each protein is coloured in rainbow colours from dark blue for the N-terminus to red for the C-terminus. All secondary structures are labelled. Note that loop between α3 and α4 of TrwE/VirB10$_{CTD}$ is disordered and, as a result, these helices appear to insert half-way through the membrane. However, in Chandran et al. (2009)[23], we showed that the loop connecting the two helices is accessible from the surface of the bacterium and therefore completes the TM region and emerges out to the bacterial cell surface. c, Superposition of the structures of the hetero-trimeric unit of the O-layer from pKM101 (grey) and R388 (light blue, green and magenta for TrwE/VirB10$_{CTD}$, TrwF/VirB9$_{CTD}$, and TrwH/VirB7, respectively). The two heterotrimers superimpose with an RMSD in Cα of 0.8 Å. The various parts of the heterotrimeric complex are shown and labelled. d, Superposition of the structure of the hetero-dimeric unit of the I-layer from R388 (TrwF/VirB9$_{NTD}$ (light blue) bound to α1 of TrwE/VirB10$_{NTD}$ (green)) and that of the same region of the *Xanthomonas citri* I-layer (grey). The two structures superimpose with an RMSD in Cα of 0.9 Å. e, The interfaces between TrwF/VirB9$_{NTD}$ subunits (upper panel; the domain is in rainbow colour from N- to C-terminus) and between α1 of TrwE/VirB10$_{NTD}$ and two TrwF/VirB9$_{NTD}$ (lower panel). Secondary structures contributing residues to the interfaces are labelled. f, Insertion of two additional TrwF/VirB9$_{NTD}$-α1TrwE/VirB10$_{NTD}$ complexes diametrically opposite within the I-layer. Left: the OMCC C1 3.28 Å map shown in grey with TrwF/VirB9 linkers between the O- and I-layers shown in purple (0.17 σ level). Dashed line box indicates the section of density shown at right. Right: top view of the I-layer map section indicated by the dashed line box at left. The two inserted TrwF/VirB9$_{NTD}$ domains are recognizable because they do not have the linker connecting their NTDs (I-layer) to their CTDs (O-layer). g, Interactions between TrwI/VirB6 (the base of the Stalk) with TrwM/VirB3 and TrwK/VirB4 (in the IMC). TrwM/VirB3 and TrwI/VirB6 are shown in pale green and red ribbon, respectively, while two central TrwK/VirB4 subunits are shown in dark blue and sky blue ribbon and semi-transparent surface. Secondary structures contributing interacting residues are labelled. Inset: overview of the location of the zoomed-up structure shown in main panel. h, Additional densities (shown in semi-transparent grey) observed in the Arches and the IMC potentially corresponding to a fourth molecule of TrwG/VirB8. Additional density in the IMC-Arches-Stalk C1 6.18 Å map (Extended Data Fig. 3f) was observed forming a helix tube bound to the 3-helices bundle of the TrwG/VirB8$_{tails}$. Correspondingly, an additional density was observed near the TrwG/VirB8$_{peri}$ ring. This may indicate the presence of a fourth TrwG/VirB8 subunit (shown in magenta ribbon). Finally, there is density between the TrwG/VirB8$_{tails}$ and the TrwG/VirB8$_{peri}$ domains, which we hypothesize might be formed by the residues in between the two domains (residues 62-95; Extended Data Fig. 1b). However, the densities were too poor to be assigned and we remain unsure as to potential interpretations and assignments. i, Extra density also observed in the IMC-Arches-Stalk C1 6.18 Å map (Extended Data Fig. 3f). Left: overall structure of the T4SS with a dashed lined box showing the location of the extra density. Proteins are colour-coded as in Fig. 1d and are in ribbon representation, except for TrwE/VirB10 which is in surface representation. Right: zoom-in on the region of the structure shown in the dashed lined box shown at left. Two extra densities (in green) are seen at σ 0.04 which merge into one at σ 0.02. These indicate a structure that makes contact with TrwG/VirB8$_{peri}$ and that is flexibly (shown in dashed lines) connected to another structure that makes contact with the TrwI/VirB6 TM helices and with the two subunits of the TrwK/VirB4 dimer. The density was too poor to be assigned but could correspond to TrwE/VirB10$_{NTD}$, which is known to not only make a major part of the OMCC but also has an IM TM and a cytoplasmic tail that, in other T4SSs, has been known to interact with VirB8, VirB6 and VirB4[70–72]. However, it could be that this stretch of density may correspond to different proteins.

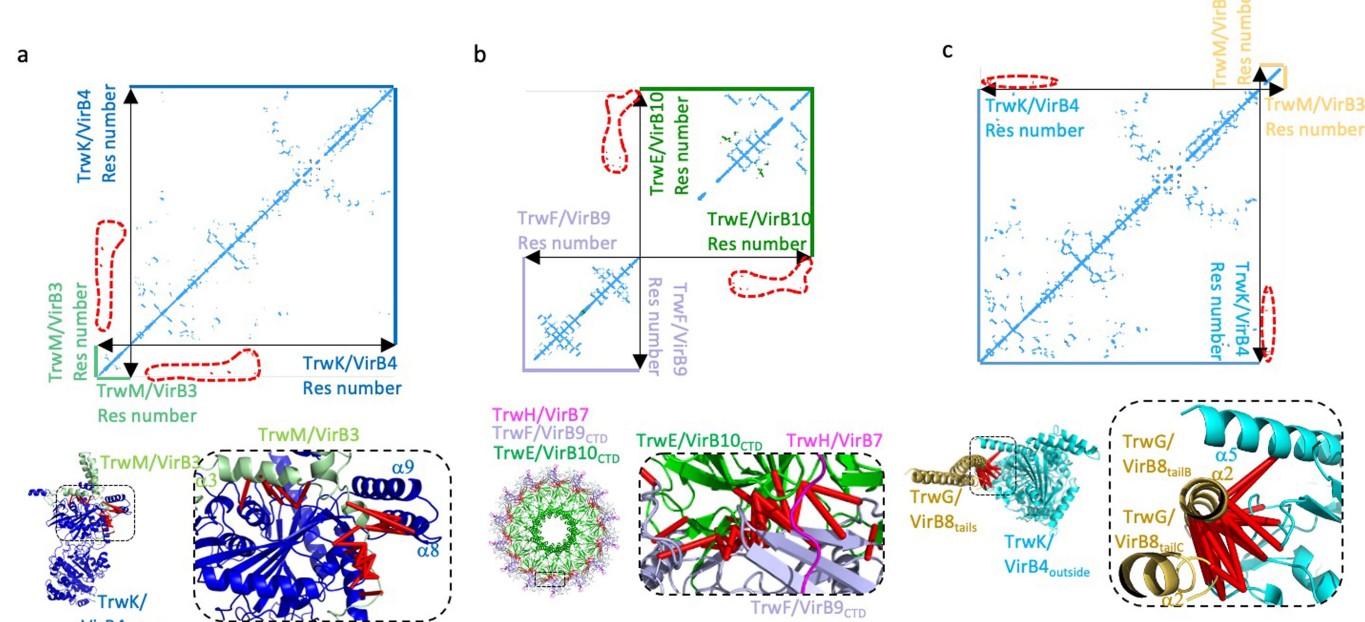

**Extended Data Fig. 9 | Validation of the T4SS heterologous interfaces using the co-evolution method as implemented by TrROSETTA.** For each panel, three sub-panels are shown. Upper panel: plots of pairwise TrROSETTA contact probability score. Each dot represents a pair of co-evolving residues with TrROSETTA score larger than a case-specific threshold (see Methods). Dots are coloured blue, green, or red (also surrounded by a red dashed line), for intra-protein co-evolution pairs, homo-oligomeric co-evolution pairs, and hetero-oligomeric co-evolution pairs, respectively. Lower panels: mapping of top co-evolution pairs onto the structure reported here (70% threshold except for the TrwK/VirB4$_{central}$ - TrwM/VirB3 interaction where the 30 top pairs are mapped; in green in Supplementary Table 4). Dashed box in lower left panel locates the structure showed right. Pairs of residues across the interface are linked by red bars. As can be seen from the list in Supplementary Table 4, going down the list, distances greater than the generally accepted 12 Å limit for Cα-Cα distances between interface residues[73,74] are found but rank very poorly (pairs in yellow in Supplementary Table 4). a, Plot and mapping of co-evolving residue pairs between TrwK/VirB4$_{central}$ and TrwM/VirB3. b, Plot and mapping of co-evolving residue pairs between TrwE/VirB10$_{CTD}$ and TrwF/VirB9$_{CTD}$ in the OMCC O-layer. c, Plot and mapping of co-evolving residue pairs between TrwK/VirB4$_{outside}$ and TrwG/VirB8$_{tails}$. See details in Supplementary Table 4 and Methods. Note that residues of TrwK/VirB4$_{outside}$ that interact with α1 residues in TrwG/VirB8$_{tailsA}$ (α1A in Fig. 2e) were not among the top 100 co-evolving residue pairs between these two proteins (Supplementary Table 4), therefore this very small part of our structural model remains unvalidated.

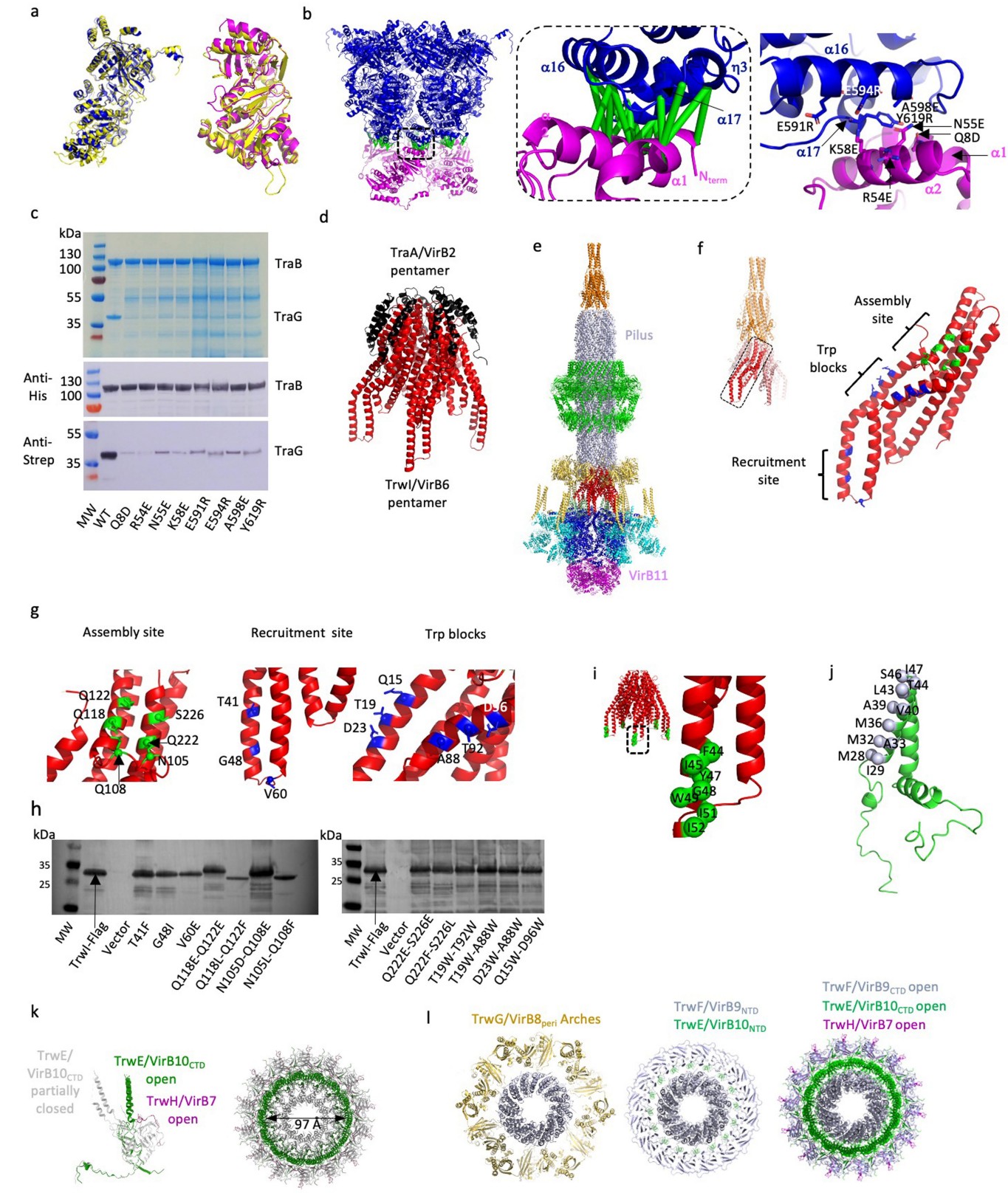

**Extended Data Fig. 10** | See next page for caption.

**Extended Data Fig. 10 | Validation of the VirB4-VirB11 interaction, mutational analysis of the VirB2-binding and -assembly sites on VirB6, and conformational changes required or not required for the pilus to pass through the Arches (not required), the I-layer (not required) and the O-layer (required).** a, Assessment of the TraB/VirB4 and TraG/VirB11 ALPHAFOLD models against known individual structural homologues. Left: superposition of the ALPHAFOLD model of TraB/VirB4 (in yellow) with the cryo-EM model of TrwK/VirB4$_{central}$ (in dark blue; this work). Right: superposition of the ALPHAFOLD model of TraG/VirB11 (in yellow) with the crystal structure of *B. suis* VirB11[38] (2GZA, in magenta). RMSD in Cα position are 1.26 Å and 1.23 Å, respectively. b, The VirB4-VirB11 structural model and the two independent methods used for its validation. Left: ALPHAFOLD structure of a complex of TraB/VirB4 (dark blue ribbon) bound to TraG/VirB11 (magenta ribbon). Middle: first validation of the complex model using the mapping of the TrROSETTA top-scoring (70% threshold) co-evolving pairs (listed in green in Supplementary Table 4; see Methods). Right: location of the interface residues mutated to provide a second independent validation of the ALPHAFOLD complex model. c, Pull-down of the TraB/VirB4-TraG/VirB11 wild-type and mutated complexes. Pull-downs were performed by taking advantage of a His-Tag at the C-terminus of TraB/VirB4. Samples were loaded so as to equalise as much as possible the amount of TraB/VirB4. The coomassie-stained gel is shown in upper panel, while the Western blots using anti-His antibodies to detect TraB/VirB4 (middle panel) or anti-Strep antibodies to detect TraG/VirB11 (lower panel) are shown below. Mutants are indicated as well as the positions of some molecular markers (MW), and those of TraB/VirB4 and TraG/VirB11. All mutated proteins expressed as well as wild-type and were equally soluble (not shown). Although similar amounts of TraB/VirB4 are loaded, the TraG/VirB11 band is less intense in the mutants, indicating weaker interactions; moreover, the inability of interface mutants to bind TraG/VirB11 also results in a slight degradation of TraB/VirB4 which is not observed in the wild-type interaction. For gel source data, see Supplementary Fig. 1c and corresponding legend in SI guide. n = 3 independent experiments. d, PATCHDOCK docking of the TraA/VirB2 and the TrwI/VirB6 pentamers. The docking is based on shape complementarity. The top-scoring model is shown in ribbon representation colour-coded black and red for TraA/VirB2 and TrwI/VirB6, respectively. The TrROSETTA analysis of VirB2-VirB6 did not detect any pairs involving VirB6 residues in this putative assembly site. This is not surprising: transient interactions are known to provide only weak co-evolutionary pressure[75] and one indeed expects VirB2 subunits to make only transient and weak interactions with the assembly site so as to not prevent incoming VirB2 subunits from displacing already assembled subunits at the base of the pilus. e, Model of the T4SS with bound F pilus and ALPHAFOLD TrwD/VirB11 model. Proteins are shown in ribbon colour-coded as in Fig. 1d, with the pilus in blue-white and TrwD/VirB11 in magenta. The O-layer is in the open conformation as shown in panel k. f, Overall view of the TrwI/VirB6 residues mutated in this study. Inset at left: overall stalk view in red (TrwI/VirB6) and orange (TrwJ/VirB5) ribbon representation. The dashed line box indicates the zoomed-in region at right. Residues mutated are shown in ball-and-stick representation. Mutations increasing conjugation are shown in green while those decreasing conjugation are shown in blue. The sites are labelled. For the VirB2- binding/recruitment site (labelled "recruitment site"), residues were mutated to bulky residues (T41F or G48I) or to acidic residue (V60E), all three anticipated to

interfere with VirB2-binding. For the VirB2-assembly site, 3 pairs of double mutations, one to acidic residues, the other to hydrophobic, were designed with the intention of potentially increasing the affinity of the site for VirB2 subunits, thereby potentially increasing the binding of the pilus to its base. The R388 and pKM01 pilus are known to only weakly attach to the cell surface and thus, mutations increasing its affinity to its VirB6 base might increase its residency time, thereby affecting conjugation. Note that none of the mutants in the assembly site overlaps with the TrwJ/VirB5 binding site. Finally, For the Trp-blocks, 3 double mutations to W were implemented, with the intention to create obstacles preventing VirB2 subunits from reaching the assembly site while translocating from their binding site. g, Details of residues mutated in the assembly site (left), the recruitment/binding site (middle) and to generate the Trp blocks (right). Residues and TrwI/VirB6 are as in panel e. Residues are labelled. h, Western blot analysis of wild-type and mutated TrwI/VirB6 in the membrane (see Methods). All mutated TrwI/VirB6 proteins express similarly and all locate to the IM to the same extent as wild-type. Note that Redzej et al. (2017)[9] have shown that deletion of TrwI/VirB6 does not affect the integrity of the T4SS except for TrwJ/VirB5 which is lost. However, none of the mutations described here are within the TrwJ/VirB5 binding site and thus, we can safely conclude that the T4SS assembly is not affected in any of the mutants. For gel source data, see Supplementary Fig. 1d and corresponding legend in SI guide. n=3 independent experiments. i, Position of the VirB6 residues involved in the 50 top-scoring co-evolution pairs listed in Supplementary Table 4 for the VirB2/VirB6 interaction. All locate in the α1 helix, making this helix a strong candidate for VirB2 binding and recruitment. Inset at left: overall TrwI/VirB6 pentamer structure in red ribbon representation. Dashed line box indicates the zoomed-in region at right. The Cα atom of each residue is shown in sphere representation coloured in green. j, Position of the VirB3 residues involved in the 50 top-scoring co-evolution pairs listed in Supplementary Table 4 for the VirB2/VirB3 interaction. All locate in the α1 helix, making this helix another strong candidate for VirB2 binding. The Cα atom of each residue is shown in sphere representation coloured in blue-white. k, Conformational change needed to open up the O-layer OM channel. As shown in Extended Data Fig. 8a, c, the heterotrimeric unit of the O-layer contains a 2-helix bundle that traverses the OM. 14 of these bundles form the OM channel. For the pilus to go across the OM through this OM channel, the 14 helical bundles need to open up, a conformational change that only requires a hinge motion along the two linkers that connect each of the helical bundles to the TrwE/VirB10$_{CTD}$ β-barrel. Left panel: superposition of the heterotrimeric unit of the open (in green) and partially closed (in grey) O-layer. The partially closed state is that seen in the T4SS structure solved here. The open state is the one modelled here as described in Methods. Right panel: superposition of the entire open and partially closed O-layer states. The dimension of the open channel is shown. l, Cut-away of the structure with pilus shown in panel e. Colour-coding for the various proteins are indicated by labels, except for the pilus which is as in panel e. Left: cut-away at the level of the Arches. Middle: cut-away at the level of the I-layer. Right: cut-away at the level of the O-layer in its open conformation. These panels illustrate the fact that no conformational change is needed in the Arches or the I-layer for the pilus to pass through during pilus biogenesis but is needed to pass through the O-layer. We hypothesize that the Arches and the I-layer provide scaffolding rings through which the pilus is directed.

# nature research

# Reporting Summary

Nature Research wishes to improve the reproducibility of the work that we publish. This form provides structure for consistency and transparency in reporting. For further information on Nature Research policies, see our Editorial Policies and the Editorial Policy Checklist.

## Statistics

For all statistical analyses, confirm that the following items are present in the figure legend, table legend, main text, or Methods section.

| n/a | Confirmed | |
|---|---|---|
| ☐ | ☒ | The exact sample size (*n*) for each experimental group/condition, given as a discrete number and unit of measurement |
| ☐ | ☒ | A statement on whether measurements were taken from distinct samples or whether the same sample was measured repeatedly |
| ☐ | ☒ | The statistical test(s) used AND whether they are one- or two-sided<br>*Only common tests should be described solely by name; describe more complex techniques in the Methods section.* |
| ☒ | ☐ | A description of all covariates tested |
| ☒ | ☐ | A description of any assumptions or corrections, such as tests of normality and adjustment for multiple comparisons |
| ☐ | ☒ | A full description of the statistical parameters including central tendency (e.g. means) or other basic estimates (e.g. regression coefficient) AND variation (e.g. standard deviation) or associated estimates of uncertainty (e.g. confidence intervals) |
| ☐ | ☒ | For null hypothesis testing, the test statistic (e.g. *F*, *t*, *r*) with confidence intervals, effect sizes, degrees of freedom and *P* value noted<br>*Give P values as exact values whenever suitable.* |
| ☒ | ☐ | For Bayesian analysis, information on the choice of priors and Markov chain Monte Carlo settings |
| ☒ | ☐ | For hierarchical and complex designs, identification of the appropriate level for tests and full reporting of outcomes |
| ☒ | ☐ | Estimates of effect sizes (e.g. Cohen's *d*, Pearson's *r*), indicating how they were calculated |

*Our web collection on statistics for biologists contains articles on many of the points above.*

## Software and code

Policy information about availability of computer code

| | |
|---|---|
| Data collection | EPU version 2.7 software (Thermo Fisher, USA), HMMER4 |
| Data analysis | RELION 3.1, CRYOSPARC v2.15, MOTIONCOR2 , CTFFIND v4.1 , GAUTOMATCH v0.563, PHENIX v1.18, PHENIX v1.18.2, MOLPROBITY v4.4, MOLPROBITY v4.5.1 , COOT v0.9.3, CHIMERA v1.13.1, CHIMERA v1.4, CHIMERAX v1.1, PYMOL v2.3.2, HDOCK webserver, PATCHDOCK webserver, YASARA webserver, HMMER hmmbuild webserver, ROSETTA webserver, ROSETTACM webserver, TrRosetta.v1, ROBETTA webserver, PSIPRED webserver, PHYRE webserver, PISA webserver, TMpred webserver, DALI webserver, ALPHAFOLD1, FIJI-1.53, IMAGIC-5. |

For manuscripts utilizing custom algorithms or software that are central to the research but not yet described in published literature, software must be made available to editors and reviewers. We strongly encourage code deposition in a community repository (e.g. GitHub). See the Nature Research guidelines for submitting code & software for further information.

## Data

Policy information about availability of data

All manuscripts must include a data availability statement. This statement should provide the following information, where applicable:

- Accession codes, unique identifiers, or web links for publicly available datasets
- A list of figures that have associated raw data
- A description of any restrictions on data availability

EM maps and atomic models were deposited to the EMDB and PDB data bases. Accession codes can be found in ED Table 1 of the manuscript. PDB codes for the various structures reported in this manuscript are 7O3J, 7O3T, 7O3V, 7O41, 7O42, 7O43, 7OIU, 7Q1V and the EMDB deposition numbers for the EM maps are EMD-12707, EMD-12708, EMD-12709, EMD-12715, EMD-12716, EMD-12717, EMD-13765, EMD-13766, EMD-13767, EMD-13768, EMD-12933. All constructs (wild-type and mutants) used in this study can be obtained on request to GW.

# Field-specific reporting

Please select the one below that is the best fit for your research. If you are not sure, read the appropriate sections before making your selection.

☒ Life sciences  ☐ Behavioural & social sciences  ☐ Ecological, evolutionary & environmental sciences

For a reference copy of the document with all sections, see nature.com/documents/nr-reporting-summary-flat.pdf

# Life sciences study design

All studies must disclose on these points even when the disclosure is negative.

| | |
|---|---|
| Sample size | 104,711 micrographs were collected. |
| Data exclusions | No data were excluded from the analysis |
| Replication | All conjugation measurements were performed in triplicates |
| Randomization | Not relevant to this study |
| Blinding | Not relevant to this study |

# Reporting for specific materials, systems and methods

We require information from authors about some types of materials, experimental systems and methods used in many studies. Here, indicate whether each material, system or method listed is relevant to your study. If you are not sure if a list item applies to your research, read the appropriate section before selecting a response.

## Materials & experimental systems

| n/a | Involved in the study |
|---|---|
| ☐ | ☒ Antibodies |
| ☒ | ☐ Eukaryotic cell lines |
| ☒ | ☐ Palaeontology and archaeology |
| ☒ | ☐ Animals and other organisms |
| ☒ | ☐ Human research participants |
| ☒ | ☐ Clinical data |
| ☒ | ☐ Dual use research of concern |

## Methods

| n/a | Involved in the study |
|---|---|
| ☒ | ☐ ChIP-seq |
| ☒ | ☐ Flow cytometry |
| ☒ | ☐ MRI-based neuroimaging |

## Antibodies

| | |
|---|---|
| Antibodies used | Anti-His antibodies, Anti-Strep antibodies, Anti-FLAG antibodies |
| Validation | Anti-His antibodies were from SIGMA, Anti-Strep antibodies were from MERCK, anti-FLAG and anti-rabbit antibodies were from ABCAM. These commercially available antibodies are validated by the companies. |

