## [Peer Review File · Nature]

Manuscript Title: Cryo-EM structure of a type IV secretion system

Reviewer Comments & Author Rebuttals

Reviewer Reports on the Initial Version:

Referees' comments:

Referee #1 (Remarks to the Author):

In this manuscript, Macé et al. describe a high-resolution structure of the Type IV secretion system from *E. coli*, including detailed structural information for several different constituent sub-complexes and the proteins that make them up.

Overall the work constitutes an impressive and important technical achievement, but the manuscript itself is quite difficult to follow and should be significantly revised prior to publication. Throughout the manuscript, the authors catalog a long and detailed list of structural features and describe them in detail. Discussion of biological background, prior work in the field, and implications for understanding T4SS function is almost entirely absent. For a journal with a broad readership, it would be extremely valuable to substantially rewrite the manuscript to emphasize biological context and minimize excessive structural descriptions. Ultimately, detailed structural information is best gleaned from the coordinates and maps, rather than verbal descriptions.

In addition, the authors claim not only to present the first near-atomic structure of a type IV secretion system, but also to shed light on the mechanisms by which bacteria with such systems develop pili for conjugation. Although manuscript clearly accomplishes the first of these goals, essentially no data are presented that support the second claim. While the computational methods used to interpret the structure are suggestive, they are presented as observational, rather than speculative, despite the lack of definitive evidence to support the model proposed.

Major points:

1. The proposed mechanism of pilus assembly is purely speculative and should be presented as such, unless the authors have further evidence to support their model. If so, they should include citations for the relevant work and discuss the most critical results in the main text. Otherwise, the authors should temper their language in describing the proposed mechanism, and perhaps use cartoons instead of structural models for illustration.
2. The authors use evolutionary coupling analysis as an orthogonal approach to validate the structure of the T4SS complex, but they show only a few regions in Fig 3 f-g. Since this analysis is used as validation, the complete analysis should be included as a table or figure, in which the scores (or percentile ranks) for each pairing are shown, including any couplings for each component pair that fall above a pre-defined threshold value.

3. Validation of the placement of TrwD/VirB11 using evolutionary coupling analysis is overstated. Although the data are consistent with the correct placement of the molecules, they are incompletely validated. Because the mutations used in the pull-down assays each contain at least one of the residues identified in the co-EV analysis, this doesn't serve as independent validation. If residues not identified by co-EV, but predicted by the docking, were used instead, this would provide a more compelling validation of the placement. In addition, an alternative method such as low-resolution cryo-EM or negative-stain EM could be used to support the model.

4. Clarify and rearrange the figures: each figure contains data from multiple structures, and this arrangement is fairly confusing (e.g., the breaking of symmetry in the OMCC layers is discussed in figures Fig.3, ED Fig. 2 and ED Fig. 5). The authors should consider grouping data that pertain to each substructure of the T4SS complex into a main figure and a number of extended figures attached to it, so as to make navigation more straightforward. Additionally, the figures that describe the symmetry relationships between various oligomers are not easy to understand. The authors could make use of additional arrows, cartoons and/or symmetry symbols to clarify the relationships and the flow between sub-figures.

5. Improve the quality of the models. Granted that the presented structure comprises numerous proteins and represents a major challenge to build, the authors may still be able to improve the quality of the models.

Minor points:

1. Is there a component missing in the Arches? There appears to be an additional density in the map that isn't present in the model in Figure 1.
2. How were the membrane densities defined in Figure 1?
3. The OM helices of VirB10 don't span the entirety of the OM density. Is this expected?
4. Observed 2D classes show that the OMCC position relative to the rest of the complex is dynamic. This is a key observation that should be shown as part of a main figure, and an explanation for this added to the main text.
5. Fig 2 in particular requires clarification to explain various relationships between oligomers (e.g., use symbols/arrows to explain the symmetry between the VirB4_{inside} and VirB4_{outside} protomers within the dimers of VirB4).
6. Why was the interface between adjacent TrwK/VirB4 central subunits in the hexamer termed "weak"? The authors speculate that a conformational change that tightens the contacts at those interfaces is required for activation of the ATPase. Is there evidence that a functional ATPase has a more compact structure? Additional information about the presence or absence of ATP and the function of this subunit would also help to clarify this point.
7. Why was the structure of the ATPase VirB4 solved separately and then docked into the complex structure?
8. What does "apo" mean in the context of the VirB4 ATPase? Outside of the T4SS complex, or not bound to ATP, or both? Please define this explicitly in the text.
9. The break in the symmetry between the O-layer and I-layer in the OMCC map shown in ED Fig 2a isn't clear in the text. How does this asymmetry arise? It would be helpful if the authors further elaborated. For example, are two components inserted in the I-layer, or conversely lost in the O-

layer? Are the inserted components truncated proteins or are the other domains simply flexible and therefore not visible in the density? The same comment applies to ED Fig 5k.

10. The authors point out structural differences between VirB5 pentamer in the T4SS structure and monomeric structures of its homologs determined previously and claim that those arise from conformational rearrangements (shown in ED Fig 5d). Is it possible that there are simple structural deviations between distantly related proteins? If it is motion, do the authors believe that it might represent the opening-closing transition of the conduction pore? Have similar conformational changes been observed in the past for other systems? As with the manuscript as a whole, more context would be useful.

11. In ED Fig 1g, it appears that the highest local resolution is in the noisy parts of the map where one would expect the resolution to be the lowest. Please comment.

12. In ED Fig 3a, the local resolution plot doesn't extend to the resolution of the map (3.5 Å).

13. Regarding the TrwK/VirB4 trimer of dimers which locates "sideways" and is reminiscent of negative-stain data: this point needs to be clarified. Perhaps a fit to the negative stain envelope could be used to help clarify.

14. Typo: TrwK/VirB5 on page 6, should be TrwJ/VirB5?

15. In Figure 3f, it isn't clear that the inter-protein couplings are shown between the intra-protein couplings. Please label.

Referee #2 (Remarks to the Author):

The authors present a work aimed at gaining insights into the structure and assembly of the Type IV secretion system (T4SS). The system is present in Gram-negative bacteria, is membrane embedded, and is central to bacterial conjugation. The authors applied cryo electron microscopy, molecular modeling and co-evolution analyses to arrive at a new-atomic model of the T4SS. More specifically, they show that the structurally fragile system can be 'separated' into 4 subcomplexes (OMCC, stalk, arches, IMC), which have been structurally investigated by focussed reconstructions to finally built a composite structure of the T4SS. Additionally, two components were added to the model by applying co-evolution studies.

Specific comments and questions:

Biochemistry and Function:

- The authors state that the T4SS structure presented here is not executing DNA transfer but is responsible for pilus biogenesis, and that the VirB4 ATPase is inactive (based on interface size between AAA+ motor subunits). Is the T4SS encoded by the R388 active or secretion-incompetent in *E. coli*? Was it capable of assembling a pilus before isolation or is it already not active in cellulose?

- In ED Fig. 1C, VirD4 appears to be co-purified with the visualized complex, yet there is no trace of it in the structure. At the same time, at a glance, the expression plasmid and purification protocol is the same as the one employed by the authors previously (Redzej et al., 2017). Can the authors comment on this? At the same time, the relatively large conformational differences observed for

VirB4 in the cytosol, as opposed to the ones observed in the past (Low et al., 2014, Redzej et al., 2017;) are not discussed (Low et al 2014 presented a negative stain structure but also provided evidence for its integrity using cryo-negative stain).

Have the authors observed 3D-classes that look similar to the ones published by the same group? What can induce this conformational change for VirB4 (which is not caused by VirD4). Is this in any way related to the functionality of the system? Or, is the previously published structure now in question?

- The main plasmid used in this study (pBADM11_trwN/virB1-trwE/virB10Strep_rbstrwD/virB11_rbsHistrwB/virD4) and used previously by the researchers in Redzej et al., 2017, is not included in the used constructs/plasmids.

- 'Apo' normally refers to ligand binding, so it appears misleading to call a sole VirD4 as 'apo state'.

- The context/reference to the type VII secretion system is unclear: Type VII is a hexameric assembly (Beckham et al, 2017 referenced in Famelis et al 2019) and in the cited paper (Famelis, Calzada et al., 2019) authors show a structure of a dimeric building block of this complex. As such, both protomers show the same level of occupancy, while the ATPase motor component (of both protomers) is flexible. The other 4 protomers of the type VII secretion system are not showing different levels of occupancy, but are missing completely, as the dimeric unit is considered to be a building block. This is also referenced in the Methods section again, but this reviewer does not understand the connection. Have the authors considered, if the imaged assembly is an intermediate structure towards a (potentially) final complex that could display the double barrel assembly that they have shown in the past? This would maybe explain why three VirB4 dimers are more stably bound.

- The authors show that VirB11 interacts with VirB4 (from pKM101 and not R388; ED Fig 6(b) . This interaction is disrupted upon mutating various residues in VirB4 or VirB11 and the authors presume this interaction to be disrupted by the high concentration of detergents used for membrane extraction of the entire complex. What happens to these complexes (VirB11:VirB4) upon exposure to the detergent-containing T4SS purification conditions? Since VirB11:VirB4 seem to copurify quite efficiently, have the authors tried to add VirB11 to their purified protein preparations after extracting/purifying the T4 complex from the membrane? According to the methods, these complexes (VirB11:VirB4) were purified by tandem-purification using a StrepTrap column first (VirB11-Strep) followed by IMAC (VirB4-his). Why does VirB4-his show up in lanes 2-4 after Strep purification if complexes do not form? (or is the labeling wrong?) SDS-PAGE: Proteins separated in Lane 1 should be adjusted to similar levels as in lane 2-4 based on the first affinity step (VirB11-Strep) (faster migrating bands are visible also in lane 2-4, but strongest in lane 3).

Structural analyses, model building and interpretation:

- The authors should include a Supplementary Figure showcasing the used data processing pipeline. This should include also images of the main maps/classes obtained along the way (e.g. critical 2D/3D classifications), that were obtained before the final maps. This will ease the following of the processing pipeline. In its current version, it is difficult to assess any of these steps as no intermediate results are shown.

- The authors are encouraged to deposit not only final sharpened maps, but also 3D-refinement maps, half-maps and masks. In particular, it would be important do also deposit the entire C1 EM map of the OMCC, shown in ED Fig 2a.

- Angular plots for reconstructions/particles should be provided for all maps (including for the C1 maps)

- It would be helpful, if the authors can make a table referring to their maps/models/depositions and clearly distinguish between what components (and corresponding residues) are built de novo in which map, which component/region is rigid body fitted and so on. The assembly of an IMC model should contain the name 'composite' in it, to make it clear that not all contact points between various components are clearly observed in the data.

- Are the OMCC complexes always attached to the IMC, or do the authors also observe particles in which these two sub-complexes are separated?

- Which experiments were performed in order to validate the high-symmetries observed in the OMCC (C14 and C16 symmetries)? Processing of the C1 map and validation tests for correct symmetries are not described and can therefore not be assessed. Why does the OMCC C1 map actually appear like a C2 map (based on density thickness of opposing densities (ED, Fig2/a, symmetry sections) and can the authors exclude wrong Euler angle assignment? (Note, the C1 maps shown in ED Fig2 were not part of this manuscript submission and details of the processing were not described but should be provided).

Have the authors checked a potential model/map bias during reconstruction (what is the outcome of the C1 reconstruction, when stronger low-pass filters are applied for the reference maps (such as 40Angstroems or 60Angstroems, or reference maps with different symmetries were used).

- Model building is reported to be done into 4 maps obtained at a resolution range from 2.5 to 3.7 Å. However, in the methods authors state that the model for the periplasmic VirB8 trimer (Arches) was built into a 6.2 Å C1 reconstruction (the best map available for this region) using rigid-body fitting of coordinates obtained from ROBETTA. Authors claim the resolution was "sufficient" to build a model including side chains for most parts of the structure (referencing Fig. 1b, which includes e.g. the periplasmic VirB8 trimer and the cytoplasmic face of inner VirB4 (res: 502-592) interacting with VirB11 to mention two example areas). The alpha helices (which are typically the features that are resolved best at low resolution) are barely visible in both of these areas, yet the models in these areas contain side chains (which are not supported by the experimental data). At the same time, authors do not model but only speculate about a fourth VirB8 copy, although the density corresponding to the TMH of this putative fourth VirB8 copy is considerably better resolved (in the IMC_Protomer_C1 map at 3.7 Angstroem map but also in the IMC-Arches_Stalk_C1 map at 6.2 Angstroem) than the entire periplasmic VirB8 area and the cytoplasmic face of inner VirB4. On which grounds do authors decide to build atomic models? How can the authors be confident that the periplasmic VirB8 and cytoplasmic VirB4 (res: res: 502-592) density has been interpreted correctly?

- The authors identify that the VirB9/10 NTDs are complexed in the I-layer of the T4SS, which has a

16-fold symmetry. In contrast, the O-layer does only contain 14 copies of VirB9/10 CTDs, revealing a symmetry mismatch in this part of the T4SS structure. Further, the authors show that the two VirB9 CTDs missing in the O-layer are the ones assuming diametrically opposite positions in the I-layer complex. The authors conclude that 2 heterodimeric VirB9/10 NTDs (alpha 1 for VirB10) insert into the I-layer, “while the CTDs of these proteins are not inserted in the O-layer”. This raises the very intriguing question where these CTDs end up. The authors speculate that the NTD of VirB10 may assume a scaffolding function (ED. Fig. 5n) by connecting the arches (below the I/O-layer) in the periplasm to the VirB4 cytoplasmic ATPase (thereby bridging the inner membrane). Why do authors suggest the VirB10 NTD rather than the VirB10 CTD (and why not VirB9?), which already contains a hydrophobic helix inserted into the OM that could be repurposed for this function? More importantly however, the reasoning that not only the two VirB9 but also the two VirB10 emerge from subunits located on opposite sides of the T4SS I-layer requires further validation. (Note: Could the authors provide a description of how the unaccounted density was visualized? In the map provided (6.2 Angstrom map) the unaccounted density does not show this long extension towards VirB8 and is about only half the length shown). Could it be that the unaccounted density corresponds to a yet to be identified protein component of the T4SS?

- The peripheral periplasmic area (Fig. 1a, IMC_Protomer_c1 map 3.7 Angstrom map but also in the IMC-Arches_Stalk_c1 map 6.2 Angstrom) surrounding the trimeric VirB8 proteins not only provides space for a fourth VirB8 copy but could eventually also host another soluble domain. Maybe the soluble domains of either VirB9/10 are forming a complex with the trimeric periplasmic VirB8 domains to recruit the fourth VirB8 copy at the very edge of the T4SS structure. Coevolution analysis between VirB9/VirB8 and VirB10/VirB8 could provide useful information about this.

- Coevolution analysis provides robust validation for the well-resolved VirB5/6 interface in the stalk, the VirB9/10 interface in the O/I-layer, the large VirB4central/VirB4central interface and the VirB3/VirB4 interface, however interactions that rely on structural plasticity are seemingly not captured well by this method (e.g. the N-termini of VirB8 forming either one continuous or two helices do show coevolving residues only for the continuous helix in the corresponding VirB4outside). How can this be explained? As a control and validation of the paucity of interactions between subcomplexes, authors should also provide coevolution analysis of proteins forming plastic intermolecular interactions such as the VirB8 periplasmic trimer and of those belonging to subcomplexes that are not immediately interacting in all areas of the presented structure (e.g. VirB6/VirB3 and VirB6/VirB4).

- The authors write that the VirB4 AAA+ motor has 6-fold symmetry and that the interface between subunits is small. Can the authors elaborate how the 6-fold symmetry is related to the function of this AAA+ motor? Understandably, better characterization is hampered by partial occupancy for which the reason is unclear. However, the overwhelming majority of AAA+ structures show that co-existing ATP/ADP nucleotides cause AAA+ motors to be asymmetric. And that the asymmetry is crucial for their function. In cellulose, nucleotides are always present (at different ATP/ADP ratios). It appears the 6-fold arrangement is a consequence of the configuration of the VirB4 ATPase in the absence of nucleotide-dependent interprotomer contacts, the rather low global resolution (6.2 Angstrom) achieved and the lack of VirB11. On a more general note, it probably would be helpful if the authors use the term ‘symmetry’ only in cases, where there is high-resolution structural

evidence provided, and use 'oligomericity' (such as hexamer) to claim a structural arrangement from several protomers.

- The fitting of the VirB11 hexamer model to the VirB4 structure is informed by coevolution analysis. The residues listed in Fig. 4a belong to pKM101-TraB/VirB4. It would be helpful if the authors would highlight interfacial/mutated residues identified by coevolution analysis/targeted by site-directed mutagenesis also in the sequence alignments (ED. Fig. 4). Moreover, showing the sequence alignments of the missing VirB2/11 would also be useful because VirB11 has been subject to mutational analysis (Fig. 4a) and VirB2 homologous sequences have been used to infer the putative structure of the R388_trwK T4SS pilus.

- The authors reason on the basis of superposition analysis (and the resulting clashes) that the interactions site for VirB3 and VirB8tail with VirB4central and VirB4outside, respectively, are unique and cannot be interchanged (ED Fig.3g+h). However, the three modelled VirB8tail structures show a high degree of structural plasticity, especially in their N-termini, which could as well be used to argue that VirB8tails adapt to their local environment. The unique positions of both proteins are much more supported by the EM maps than the superposition analysis.

- The authors make extensive use of coevolution analysis and use it as a replacement for the biochemical validation of the structural data (e.g. for the well-resolved VirB5/6 interface in the stalk and VirB9/10 interface in the O/I-layer). While some residues localize in very close proximity (indicating direct interactions), others appear more far away from each other (ED. Fig. 6a). It would be helpful to also provide the distances between the observed residue pairs to judge whether the interactions are direct or indirect. Moreover, how do authors interpret large distances between residue pairs and how do they contribute to infer structural models? The authors write: "Based on the constraints imposed by the distances between coevolving residue pairs,..." How big are these distances? Can this be explained in the methods?

- It would be helpful if the authors would elaborate a bit more on the implications of their structural findings for conjugation/substrate transport/assembly such as the role of the VirB8 arches anchoring in outerVirB4, the limited connections seen between IMC and stalk.

The model of pilus biogenesis is inferred from bioinformatics (using a combination of homology modeling, docking and coevolution analysis) and provides an exciting working hypothesis. However, the proposed model cannot replace structural data on the fully-assembled, DNA-translocating T4SS and the proposed mechanism of biogenesis must therefore be validated/substantiated by additional experimental data.

- Moreover, the proposed model raises important yet unanswered questions: Why does coevolution analysis provide the interfacial residues for the VirB2 recruitment site (in the inner membrane) but not the ones deemed to form the pilus base on top of the VirB6 pentamer (Fig. 4d+e)? What does coevolution look like for other proteins located in the area of interest such as VirB3. How can cytoplasmic VirB4 proteins extract VirB2 protomers from the membrane and promote their transport on the periplasmic side and eventually across the membrane? How can the recruitment of VirB2 subunits cause the displacement of the VirB2 subunits forming the pilus base?

- The concluding statement “The structure presented here thus provides the molecular framework for designing novel compounds capable of inhibiting the transfer of antibiotic-resistance genes among pathogens...” could probably be re-phrased as the presented structure is in fact not executing DNA transfer and the motors required to energize the assembly and transport are in an inactive, eventually even unnatural state (without nucleotides). The structure rather presents the framework to design assembly inhibitors.

Additional comments

- Figure 1: figure legend: The authors state the presence of ‘lipid densities’ but likely represent a detergent belt. Did the authors do any analyses (such as mass spec) to verify the presence of lipids?

- As previously mentioned, it would be very helpful to show a flowchart of all the different processing steps including intermediate results.

Processing of the T4SS IMC, Stalk, Arches:

- The authors performed a ‘strict selection’ of 60,722 particles out of 1,292,734 extracted particles. What were the selection criteria and which particles were excluded?

- The authors “confirm” a hexameric organization of the IMC. I am wondering, whether this is just a poss

- Hexameric IMC: The authors mention to confirm a hexameric arrangement indicating to a 6-fold symmetry. They obtain a 7.6Angstroem map from 126,975 particles (C6 imposed), representing approx 760,000 asymmetric units. One would expect that the resolution would be much higher with such a dataset provided the complex is symmetric. If, however, the complex is not symmetric, the application of symmetry operators later for model building is questionable.

- Apo-Trwk/VirB4: The authors report a 4.1 Angstroem map, however the fusion protein HCP1 in this construct shows a much lower resolution. Have the authors tested, whether a higher resolution could also be obtained by focused classification as a measure for image quality.

- ED Table 1: deposit also C1 maps

- ED Fig1: for all EM maps the authors should provide also the corresponding EMD-deposition number next to the panel figure

(B) increase font along bars

(C) SDS-PAGE analysis of purified complex: highlight which proteins are present in the cryoEM analyses

(G) the ‘highest-resolution’ parts based on local resolution tests are shown in red, yet, this particular part is not representing any meaningful structure.

(I) Provide numbers of how often those particles were found compared to others - this would define the word ‘rarely’ in the associated figure legend

- ED Fig2: Provide number of particles used in the C1 maps
(C) second row: how were particles selected to show the 'complete hexamer. The reconstruction actually suggests different occupancy at the hexameric position

Referee #3 (Remarks to the Author):

In this manuscript the authors present a complete structure of a T4SS involved in DNA conjugation. This is an exciting result that is of a broad interest to researchers in various fields of microbiology and structural biology. The presented data explain many of the previously poorly understood aspects of the T4SS assembly as well as pilus formation. The protein density maps generated by cryo-EM are of high quality and thus provide unprecedented level of details and novel insights. This dataset will be invaluable for future research of T4SS and DNA transfer. Structure determination and the search for coevolving residues was done by the state-of-the-art tools. There are many additional interesting and important details provided in the supplements.

In its current form, the manuscript mainly reports the obtained EM structure, which is supported by the analysis of coevolving residues to validate known protein-protein interactions and to propose new interactions to build the overall model. While this is a good approach, it is unclear if these predictions are valid. Therefore, it would be of a great value if the authors could confirm at least some of their most exciting predictions of protein-protein interactions experimentally. The performed analysis of coevolving residues should provide guidance for such experiments. A specific point mutation introduced into one of the two putative interaction partners should result in a loss-of-function phenotype, which should be possible to suppress by the introduction of another point mutation in the second protein. The appropriate pairs of point mutations should be possible to predict based on alignments of the available sequences and bioinformatic analyses of the coevolving residues.

Indeed, the authors show that in the case of VirB4 and VirB11 interaction model, certain mutations can abrogate the interaction of those two proteins but authors should also show that the interaction can be restored by another set of mutations. An analysis of phenotypes of point mutations (and their suppression) probing the interaction of VirB2 with VirB6 should be provided as this is the hallmark of the newly proposed mechanism of pilus biogenesis.

The authors should also expand their description of the proposed model of pilus assembly. From the current description, it is hard to understand how the individual VirB2 proteins move from the "biding site" to the "assembly site" during one cycle of pilus assembly. How exactly would VirB4 be involved? It would be interesting to read ideas about this or to learn if there is additional evidence supporting this model. Related to the model, authors should also elaborate on any available data supporting the proposed O-layer opening. In general, the manuscript should probably include an expanded introduction for the readers not familiar with T4SS and DNA conjugation.

Author Rebuttals to Initial Comments:

Dear Editor,

Please find enclosed a revised version of our manuscript entitled “Cryo-EM structure of a type IV secretion system” by Kévin Macé, Abhinav K. Vadakkepat, Natalya Lukoyanova, Nathalie Braun, Adam Redzej, Marta Ukleja, Fang Lu, Tiago R.D. Costa, Elena V. Orlova, David Baker, Qian Cong, and Gabriel Waksman.

We were delighted by the positive comments and suggestions that all three reviewers have made. We have implemented them all and, as a result, the paper is significantly improved. We would like to thank the reviewers for their remarkable and helpful comments.

Throughout this rebuttal, we make the case for not only an extraordinary structure (and a novel way to validate it), but also an entirely novel mechanism with likely implications for biological processes involving the formation of soluble fibres from insoluble membrane based components, and finally for the tremendous implications for the development of novel therapeutics against one of the most threatening public health crisis in the making (potentially worse than Covid-19), that of the spread of antibiotic resistance via conjugation, a process crucially dependent on conjugative pilus biogenesis.

In summary:

- 1- Reviewers requested a validation of our pilus biogenesis mechanism model. We now provide a mutagenesis study aiming to validate our model. We report on mutants in the putative binding/recruitment site and the putative assembly site of VirB6, and also describe mutants seeking to block the transit of VirB2 subunits between the two sites. All affect conjugation, consistent with the proposed model. Reviewers requested however that we tone down our claims and we have done so.
- 2- Reviewers requested additional validation of our VirB4-VirB11 complex. We significantly improved this section. We now present an ALPHAFOLD model validated by *a-* the comparison of its prediction with the known crystal structures of the individual proteins, *b-* by TrROSETTA co-evolution results, and *c-* the description of additional mutants (compared to the first submission) and their biochemical characterisation.
- 3- Reviewers had concerns related to data availability. We have now deposited 11 maps including all C1 maps, half maps, masks, as well as the *Ab Initio* model for the IMC-Arches-Stalk. For the TrROSETTA co-evolution results, we now present the original data in SI, top scoring pairs in Fig. 3 and ED Fig. 3 (with selection threshold now defined) and have added scores and distances.
- 4- We present a new analysis of the symmetry using IMAGIC, firmly establishing the symmetry of all parts of the complex. We also have considerably improved our models. Workflows are included for all EM work. Descriptions of how all maps were obtained are included.
- 5- We show that there is no contradiction between this work and the previous work (Low et al. (2014), Nature). We explain that we substantially changed the production and purification protocol between the two works, resulting in the enrichment of the hexamer of dimers form (this work) over the double-barrelled trimers of dimers (previous work). In fact, we see both in our cryo-EM data set but the double-barrelled form is a minority species. Our claim is consistent with the observation of two oligomerisation states for VirB4: hexamer of dimers in the T4SS and trimer of dimers for VirB4 alone (formerly termed Apo-TrwK/VirB4, now called TrwK/VirB4_{unbound}), the latter reminiscent of what we observe in the double-barrelled structure. We believe the two forms of the T4SS to be equally biologically relevant, one (hexamer of dimers) relevant to pilus biogenesis and the other (the double-barrelled architecture) relevant to DNA transfer (since we observe this form when VirD4 is present).

Referee #1 (Remarks to the Author):

General comment 1:

Reviewer: Overall the work constitutes an impressive and important technical achievement, but the manuscript itself is quite difficult to follow and should be significantly revised prior to publication. Throughout the manuscript, the authors catalog a long and detailed list of structural features and describe them in detail. Discussion of biological background, prior work in the field, and implications for understanding T4SS function is almost entirely absent. For a journal with a broad readership, it would be extremely valuable to substantially rewrite the manuscript to emphasize biological context and minimize excessive structural descriptions. Ultimately, detailed structural information is best gleaned from the coordinates and maps, rather than verbal descriptions.

Response

We thank the reviewer for praising our work. However, the reviewer asks that we substantially rewrite the paper to provide biological background and context, and leave structural detailed information out. We have now substantially rewritten the manuscript to comply with this reviewer's suggestions and request. The figure panels that showed structural details have been moved to ED and we have split the ED figures to follow the structure of the paper (Fig. 1 corresponding to ED Fig. 1, Fig. 2 to ED Fig. 2, etc...). We hope this layout makes it easier to read the paper for non-specialists. Finally, we have embraced the view of the reviewer that structural details can be obtained by the reader by directly downloading maps and models. Thus, we have either removed structural details, or moved them in the ED section, or, for the former ED Fig. 4, into the SI.

General comment 2:

Reviewer: In addition, the authors claim not only to present the first near-atomic structure of a type IV secretion system, but also to shed light on the mechanisms by which bacteria with such systems develop pili for conjugation. Although manuscript clearly accomplishes the first of these goals, essentially no data are presented that support the second claim. While the computational methods used to interpret the structure are suggestive, they are presented as observational, rather than speculative, despite the lack of definitive evidence to support the model proposed.

Response

The details of our response to this comment are described below. However, we agree with the reviewer that we should be more cautious in presenting our model for pilus biogenesis mechanism. We have therefore toned down our claims and, as recommended by the reviewer, replaced realistic figures with schematic ones. However, as detailed below, we also present additional data that aim to probe the surfaces that we hypothesize may contain the VirB2 binding/recruitment site and the assembly site. We also provide additional data aiming to test the idea that it might be possible to block VirB2 subunits from reaching their assembly site by engineering blocks between the binding/recruitment and assembly sites. We believe this constitutes the first body of evidence that our model is right.

Major comments:

Major Comment 1:

Reviewer: The proposed mechanism of pilus assembly is purely speculative and should be presented as such, unless the authors have further evidence to support their model. If so, they should include citations for the relevant work and discuss the most critical results in the main text. Otherwise, the authors should temper their language in describing the proposed mechanism, and perhaps use cartoons instead of structural models for illustration.

Response

As mentioned above, we now use a different language to describe our model. A typical new sentence is for example: "As reported in Fig. 4f, all mutants exhibit decreased conjugation,

consistent with the **hypothesis** that *VirB6* $\alpha 1$ **may** contain the *VirB2*-recruitment site". We have tempered our language throughout. Also, we have removed structural models shown in former Fig. 4g. We now have a schematic model instead focusing uniquely on the pilus biogenesis mechanism we propose.

More significantly, we provide further evidence to support our model. We have indeed started to probe our mechanistic model for pilus biogenesis by targeting the proposed *VirB2* binding/recruitment site and assembly site on *VirB6* for mutational analysis (see ED Fig. 4f figure and legend for list and location of all mutants, as well as rationale for their design; results

in Fig. 4e). We made a set of 3 single-residue mutations in the *VirB2* binding/recruitment site (a small site in terms of size) and a set of 3 pairs of double-residue mutations in the assembly site (a larger size site). Both affect conjugation, one (recruitment site) decreasing it and the other (assembly site) increasing it. The latter effect is, in our opinion, very interesting because mutations that increase biological readout are often of great biological significance. We next tested the idea of attempting to block *VirB2* subunits from reaching their assembly site by introducing bulky residues, Trp, in between the recruitment

and assembly sites. Four double mutants were made and tested: all decrease conjugation. All results are shown in Fig. 4e and reported above. It is also important to note that we ascertained that none of the mutations affect *VirB6* membrane localisation (ED Fig. 4h). Since we know that deletion of *VirB6* does not affect T4SS integrity except for the loss of *VirB5* (shown by Redzej et al. (2017) EMBO Journal) and since none of the *VirB6* mutations overlap with the *VirB5*-binding site (even those in the assembly site), we do not expect the mutations to affect T4SS assembly (now stated in ED Fig. 4h legend). We also present a new docking of the F TraA/*VirB2* pentamer onto *VirB6* using the shape-complementarity software PATCHDOCK: the 10 top scoring results map to the assembly site (ED Fig. 4d). This programme is superior to HDock (initially used) as it does not require prior input/knowledge of potential interaction surfaces.

Major Comment 2:

Reviewer: The authors use evolutionary coupling analysis as an orthogonal approach to validate the structure of the T4SS complex, but they show only a few regions in Fig 3 f-g. Since this analysis is used as validation, the complete analysis should be included as a table or figure, in which the scores (or percentile ranks) for each pairing are shown, including any couplings for each component pair that fall above a pre-defined threshold value.

Response

We brought a number of modifications to this section. Previously, we used GREMLIN for co-evolution analysis of most interfaces except for *VirB5*-*VirB6* where we used TrROSETTA due to the fact that GREMLIN was not sensitive or accurate enough for the task of simultaneously validating the hetero-oligomeric and homo-oligomeric interface. Thus, TrROSETTA being a more sophisticated software, we decided to switch to TrROSETTA entirely. TrROSETTA is a Deep Learning network trained on tens of thousands of proteins in the PDB to convert the coevolution signals detected in the MSA of a protein and its homologs to residue-residue distances. In SI, we now list all the coevolving residue pairs with TrROSETTA score ≥ 0.05 for all heterologous interactions (*VirB3*-*VirB4*, *VirB5*-*VirB6*, *VirB4*-*VirB8* and *VirB9*-*VirB10*): this report includes not only the residue pairs but also, as requested, the TrROSETTA score and the corresponding distances in our structural models. As you will see, in the first 100 pairs

listed, most are within the 12 Å distance cutoff for generally accepted for C α -C α distances across interfaces. Any interaction in the first 100 pairs above 12 Å is shown in yellow in SI. However, for Fig. 3g and ED Fig. 3q-s where we show the pairs mapped onto our structure, we have limited the number of pairs shown to those with scores above 70% of the score of the top scoring pair. This is arbitrary so as to not overcrowd the figures. The only exception to this rule is for the VirB3-VirB4 interaction where using the 70% threshold would have resulted in 102 pairs being shown; therefore, for this interaction, we show the 30 top pairs. All residues shown in Fig. 3g and ED Fig. 3q-s are listed in green in SI. Note that for VirB4-VirB8, the situation is more complicated as there are 3 N-terminal tails (TrwG/VirB8_{tailsA-C}), only the interaction of two of them (B and C) being validated by TrROSETTA. Thus, only one very small interaction region of our structure (that involving α 1 of TrwG/VirB8_{tailsA}) remains unvalidated.

Major Comment 3:

Reviewer: Validation of the placement of TrwD/VirB11 using evolutionary coupling analysis is overstated. Although the data are consistent with the correct placement of the molecules, they are incompletely validated. Because the mutations used in the pull-down assays each contain at least one of the residues identified in the co-EV analysis, this doesn't serve as independent validation. If residues not identified by co-EV, but predicted by the docking, were used instead, this would provide a more compelling validation of the placement. In addition, an alternative method such as low-resolution cryo-EM or negative-stain EM could be used to support the model.

Response

We have completely redesigned this part of the project with the co-evolution analysis being no longer used to generate the VirB4-VirB11 model, but instead used (alongside site-directed mutagenesis) to validate a model obtained in a different way.

We now present a model of the VirB4-VirB11 complex structure generated by the programme ALPHAFOLD from DeepMind. This programme is shown to produce structural models to atomic accuracy in community-wide competition for structure prediction. Reassuringly, the structures predicted for each individual protein were very similar to those experimentally derived (ED Fig. 4a). However, the observed interface between the two proteins needed to be verified experimentally. Thus, we proceeded to validate it using 2 independent methods: 1- co-evolution using TrROSETTA and 2- site-directed mutagenesis. In this respect, the co-evolution exercise is entirely comparable to an experimental validation since TrROSETTA analyses the mutational landscape acquired throughout evolution. Moreover, the second validation method we used, site-directed mutagenesis, no longer seeks to validate co-evolution results but to validate the ALPHAFOLD model. To do so, we inspected this structural model and selected residues at the interface that, when mutated, would likely disrupt the interaction. The results are shown in Fig. 4a (reproduced at left): all mutants show decrease in the VirB4-VirB11 interaction, suggesting that they are indeed at the interface of the two proteins. The complete results including Western blots to identify the proteins are shown in ED Fig. 4c.

Another important change compared to the original submission is that we doubled the number of mutants analysed and all are single mutations (as opposed to double mutations previously).

Major Comment 4:

Reviewer: Clarify and rearrange the figures: each figure contains data from multiple structures, and this arrangement is fairly confusing (e.g., the breaking of symmetry in the OMCC layers is discussed in figures Fig.3, ED Fig. 2 and ED Fig. 5). The authors should consider grouping data that pertain to each substructure of the T4SS complex into a main figure and a number of extended figures attached to it, so as to make navigation more straightforward. Additionally,

the figures that describe the symmetry relationships between various oligomers are not easy to understand. The authors could make use of additional arrows, cartoons and/or symmetry symbols to clarify the relationships and the flow between sub-figures.

Response

We apologise for this lack of clarity. We have now re-organised the Figures and ED Figures so that there is correspondence between them i.e. Fig. 1 related to ED Fig. 1, etc... The old ED Figure 4, which is in effect a large data set of residue interaction is now in SI. We hope this will make the navigation a lot more fluid. Concerning the symmetry relationships between various oligomers, we now make use of colours and lines partitioning the asymmetric units in the various C1 map sections to analyse the symmetry of the various sub-complexes. We also provide a symmetry analysis using IMAGIC. We hope this will help clarify what is illustrated in each subsequent panel and how they relate with each other.

Major Comment 5:

Reviewer: Improve the quality of the models. Granted that the presented structure comprises numerous proteins and represents a major challenge to build, the authors may still be able to improve the quality of the models.

Response

We have greatly improved the quality of our model with MolProbity scores now all below 2.05 and clash score below 11.15. For some of the models, these scores were much higher before.

Minor comments:

Minor Comment 6:

Reviewer: Is there a component missing in the Arches? There appears to be an additional density in the map that isn't present in the model in Figure 1.

Response

We indeed observe additional density in the Arches which we could not interpret. In ED Fig. 3o, we show this density is located between the TrwG/VirB8_{tails} and TrwG/VirB8_{peri} domains. We postulate that this density corresponds to the intervening sequence between the two. We have added in the legend to ED Fig. 3o: "*Finally, there is density between the TrwG/VirB8_{tails} and the TrwG/VirB8_{peri} domains, which we hypothesize might be formed by the residues in between the two domains (residues 62-95; ED Fig. 1b). However, the density was too poor to be assigned and we remain unsure as to potential interpretations and assignments*".

Comment 7:

Reviewer: How were the membrane densities defined in Figure 1?

Response

The density shown in Figure 1 is the same as shown in the IMC-Arches-Stalk C1 6.18 Å map (ED Fig. 1n) and in the OMCC C1 3.61 Å (ED Fig. 1j). However, we slightly increased the contour level of the map in the membrane region and smoothed it using a gaussian filter in Chimera. Note that the density most likely consists of detergents. We have added in Fig. 1b legend: "*For the detergents/lipids densities (in transparent light blue) at the IM and OM, the maps are shown at increased contour levels of 0.03 and 0.15, respectively, and smoothed using a gaussian filter*". We have also modified Fig. 1 compared to the previous submission, which we believe now provides a better account of our study.

Comment 8:

Reviewer: The OM helices of VirB10 don't span the entirety of the OM density. Is this expected?

Response

This is expected. It has been indeed observed in our previous crystal structure of the O-layer (Chandran et al. Nature 2009) and in the present structure that the loop between the 2 OM TM helices is disordered. Yet, when, in Chandran et al. (2009), a FLAG tag was added in the

loop between these two helices, the FLAG tag was shown to be accessible at the surface of the bacterial cell, demonstrating that these loops complete the TM region and emerge out to the bacterial cell surface. In the legend of ED Fig 3i, we now say: “*Note that loop between $\alpha 3$ and $\alpha 4$ of TrwE/VirB10_{CTD} is disordered and, as a result, these helices appear to insert half-way through the membrane. However, in Chandran et al. (2009)¹¹, we showed that the loop connecting the two helices is accessible from the surface of the bacterium and therefore completes the TM region and emerges out to the bacterial cell surface.*”

Comment 9:

Reviewer: Observed 2D classes show that the OMCC position relative to the rest of the complex is dynamic. This is a key observation that should be shown as part of a main figure, and an explanation for this added to the main text.

Response

We have now added in main text as Fig. 1a and the corresponding legend has been provided.

Comment 10:

Reviewer: Fig 2 in particular requires clarification to explain various relationships between oligomers (e.g., use symbols/arrows to explain the symmetry between the VirB4_{inside} and VirB4_{outside} protomers within the dimers of VirB4).

Response

We apologise for the lack of clarity. We have now added a cartoon explaining the symmetry relationships within the IMC between TrwK/VirB4 dimers. The symmetry relationship is between VirB4 dimers within the hexamer of dimers, not between subunits within the dimer.

Comment 11:

Reviewer: Why was the interface between adjacent TrwK/VirB4 central subunits in the hexamer termed “weak”? The authors speculate that a conformational change that tightens the contacts at those interfaces is required for activation of the ATPase. Is there evidence that a functional ATPase has a more compact structure? Additional information about the presence or absence of ATP and the function of this subunit would also help to clarify this point.

Response

As now mentioned in the introduction, the VirB4 ATPase belongs to the superfamily of AAA+ ATPase that are known to function as hexamers. In this class of ATPases, binding of nucleotides often stabilises the hexamer. Since the structure of the VirB4 ATPase with bound nucleotides is not known, the structure of its active form cannot be inferred. We have therefore removed the sentence. We thank the reviewer for pointing this out. We have added in ED Fig. 2e legend: “*Without the structure of ATP-bound TrwK/VirB4, it is unclear whether the TrwK/VirB4_{central} hexamer is in an active form or conformational changes are required to transition to one.*”

Comment 12:

Reviewer: Why was the structure of the ATPase VirB4 solved separately and then docked into the complex structure?

Response

In the history of this project, the TrwK/VirB4_{unbound} (formerly named Apo-TrwK/VirB4) was solved first, before we obtained high resolution maps for the entire T4SS. This is why we used the word “first” in main text: “*The structure of TrwK/VirB4 was first solved in its unbound form (referred to thereafter as “TrwK/VirB4_{unbound}”).*” In retrospect, it is good that the TrwK/VirB4_{unbound} structure was determined because it shows an entirely different oligomerisation mode, head-to-tail trimers of dimers, reminiscent of the double-barrelled architecture we reported previously.

Comment 13:

Reviewer: What does “apo” mean in the context of the VirB4 ATPase? Outside of the T4SS complex, or not bound to ATP, or both? Please define this explicitly in the text.

Response

We have now renamed the apo-TrwK/VirB4 structure to TrwK/VirB4_{unbound}. See comment above.

Comment 14:

Reviewer: The break in the symmetry between the O-layer and I-layer in the OMCC map shown in ED Fig 2a isn't clear in the text. How does this asymmetry arise? It would be helpful if the authors further elaborated. For example, are two components inserted in the I-layer, or conversely lost in the O-layer? Are the inserted components truncated proteins or are the other domains simply flexible and therefore not visible in the density? The same comment applies to ED Fig 5k.

Response

We apologise for the lack of clarity. From the new figure we now include (ED Fig. 1r and ED Fig. 3m), we believe we can state unambiguously that there are 16 TrwF/VirB9_{NTDs}-TrwE/VirB10_{NTD} complexes in the I-layer while there are only 14 of the CTDs of the same proteins in the O-layer. We do not see truncations in the gels, so we must conclude that the 2 missing CTDs in the O-layer must be flexibly “floating” in the periplasm. However, Reviewer 2 makes an interesting suggestion that these flexible domains may account for some of the additional density we see but could not assign. Moreover, symmetry mismatches between the O- and I-layer have been described for other T4SS OMCCs. We have corrected in main text to: “*Thus, 2 heterodimeric TrwE/VirB10_{NTD}-TrwF/VirB9_{NTD} complexes insert into the I-layer (diametrically opposite as shown in ED Fig. 3m) while the CTDs of these 2 complexes are not inserted in the O-layer. Similar symmetry mismatches have been reported between layers of the OMCC of other T4SSs^{25,26}.*” Some additional texts pertaining to the same issue have been added in response to Reviewer 2 comment 14. We hope this makes it clearer.

Comment 15:

Reviewer: The authors point out structural differences between VirB5 pentamer in the T4SS structure and monomeric structures of its homologs determined previously and claim that those arise from conformational rearrangements (shown in ED Fig 5d). Is it possible that there are simple structural deviations between distantly related proteins? If it is motion, do the authors believe that it might represent the opening-closing transition of the conduction pore? Have similar conformational changes been observed in the past for other systems? As with the manuscript as a whole, more context would be useful.

Response

This is a very interesting point. For hemolysin, for example, two conformations are known, one where the N-terminal helix is folded against the hemolysin core, and another where this helix projects through the membrane. In a previous VirB5 structure (that of TraC), we observed that this N-terminal helix folds against the protein core while in the VirB5 structure in the assembled T4SS, the N-terminal helix projects out and thus would possibly be poised for insertion into the recipient membrane. Although very interesting, it is also very speculative and thus, we would like to keep the discussion very succinct. We have thus added in main text: “*However, in its pentameric form, VirB5 appears to have undergone a conformational change compared to the protein alone, with its N-terminal part projected out in a manner reminiscent of pore-forming proteins (ED Fig. 3e), a function that may be required to interact with the recipient's membrane.*”

Comment 16:

Reviewer: - In ED Fig 1q, it appears that the highest local resolution is in the noisy parts of the map where one would expect the resolution to be the lowest. Please comment.

Response

The region in question corresponds to the disordered detergent/lipid region around the TM helices of TrwI/VirB6. We observe stronger density in those regions from the start of the processing, even in 2D classes. It may be that detergent/lipids have a strong density signal

than proteins which, in turn, impact the local resolution calculation for that region. We optimised the mask to minimise this artifact.

Comment 17:

Reviewer: In ED Fig 3a, the local resolution plot doesn't extend to the resolution of the map (3.5 Å).

Response

We apologise for the mistake. It has now been rectified.

Comment 18:

Reviewer: Regarding the TrwK/VirB4 trimer of dimers which locates "sideways" and is reminiscent of negative-stain data: this point needs to be clarified. Perhaps a fit to the negative stain envelope could be used to help clarify.

Response

We have added as ED Fig. 2i.

Comment 19:

Reviewer: Typo: TrwK/VirB5 on page 6, should be TrwJ/VirB5?

Response

We have corrected.

Comment 20:

Reviewer: In Figure 3f, it isn't clear that the inter-protein couplings are shown between the intra-protein couplings. Please label.

Response

The inter-protein coupling is shown in the red circle. We have increased the thickness of the circle to make it more apparent. We also added an arrow next to the circle to draw attention to it.

Referee #2 (Remarks to the Author):

Reviewer: The authors present a work aimed at gaining insights into the structure and assembly of the Type IV secretion system (T4SS). The system is present in Gram-negative bacteria, is membrane embedded, and is central to bacterial conjugation. The authors applied cryo electron microscopy, molecular modeling and co-evolution analyses to arrive at a new-atomic model of the T4SS. More specifically, they show that the structurally fragile system can be 'separated' into 4 subcomplexes (OMCC, stalk, arches, IMC), which have been structurally investigated by focussed reconstructions to finally built a composite structure of the T4SS. Additionally, two components were added to the model by applying co-evolution studies.

Comment 1:

Reviewer: - The authors state that the T4SS structure presented here is not executing DNA transfer but is responsible for pilus biogenesis, and that the VirB4 ATPase is inactive (based on interface size between AAA+ motor subunits). Is the T4SS encoded by the R388 active or secretion-incompetent in *E. coli*? Was it capable of assembling a pilus before isolation or is it already not active in cellulo?

Response

The construct we used for purification of the complex was shown by Redzej et al (2017, EMBO Journal) to mediate conjugation in *E. coli*. We have added this remark in the Methods section.

Comment 2:

Reviewer: - In ED Fig. 1C, VirD4 appears to be co-purified with the visualized complex, yet there is no trace of it in the structure. At the same time, at a glance, the expression plasmid and purification protocol is the same as the one employed by the authors previously (Redzej et al., 2017). Can the authors comment on this? At the same time, the relatively large conformational differences observed for VirB4 in the cytosol, as opposed to the ones observed in the past (Low et al., 2014, Redzej et al., 2017;) are not discussed (Low et al 2014 presented a negative stain structure but also provided evidence for it's integrity using cryo-negative stain).

Have the authors observed 3D-classes that look similar to the ones published by the same group? What can induce this conformational change for VirB4 (which is not caused by VirD4). Is this in any way related to the functionality of the system? Or, is the previously published structure now in question?

Response

These are very important questions which we will answer one by one:

Reviewer: In ED Fig. 1C, VirD4 appears to be co-purified with the visualized complex, yet there is no trace of it in the structure.

In Redzej et al. (2017, EMBO Journal), we reported that VirD4 does not remain associated with the complex without being crosslinked. We apologise for not having mentioned this in the original version as it is important prior observation that explains why VirD4 is not part of the complex. We could speculate as to why it is weakly associated with the complex but it is likely that VirD4, known to be involved in DNA transfer and not in pilus biogenesis, needs to be stabilised by the DNA substrate, the relaxosome. It could also be that cryogenic conditions lead to dissociation of VirD4. We have added in the legend to ED Fig. 1c: "*Association of VirD4 with the T4SS may need stabilising through its interaction with the DNA substrate or its dissociation might be triggered by the cryogenic conditions.*"

Reviewer: At the same time, at a glance, the expression plasmid and purification protocol is the same as the one employed by the authors previously (Redzej et al., 2017). Can the authors comment on this? At the same time, the relatively large conformational differences observed for VirB4 in the cytosol, as opposed to the ones observed in the past (Low et al., 2014, Redzej et al., 2017;) are not discussed

We thank the reviewer for the opportunity to clarify the link between our previous structures and the one we present here. Previously, we described a double-barrelled architecture for the T4SS (Low et al, 2014, Nature), each IMC barrel made of a TrwK/VirB4 trimer of dimers. Here, we describe a hexamer of dimers architecture for the IMC. We have added in the legend to ED Fig. 1c a paragraph explaining why we obtain here a different structure. Crucial to our case is the fact that extraction and purification conditions used here are significantly different (and significantly improved) from those used by Low et al. (2014), resulting in much better yields and improved stability, making the cryo-EM analysis possible. Since we observe a small proportion of double-barrelled particles in the cryo-EM data set presented here, we conclude that the hexamer of dimers form of the T4SS must have been unstable in the conditions used by Low et al. (2014) since Low et al. did not observe it. We write the following in ED Fig. 1c legend: "*Improved yields and higher quality sample combined to make the determination of this complex structure by cryo-EM possible. As explained in main text, a minority of particles in the cryo-EM data set presented here display the typical side views of the double-barrelled structure, indicating that the majority hexamer of dimers architecture we observe here must have been unstable in the buffer and NSEM conditions used by Low et al. (2014)¹ since they did not observe it. Conversely, yields and stability of the double-barrelled complex obtained by Low et al. (2014)¹ were low, making it impossible to solve its cryo-EM structure*".

As a matter of fact, we are inspired by comment 5 of this reviewer (see below) which suggests the fascinating possibility that the two forms (double-barrelled and hexamer of dimers) might

indeed interconvert, i.e. be in a dynamic equilibrium using the IMC protomer as a building block. However, following Reviewer 1's recommendation we feel we must be careful in not excessively speculating. Nevertheless, it is our working hypothesis that the double-barrelled architecture with VirD4 in the middle as observed by Redzej et al. (2017) is representative of the DNA transfer apparatus. Indeed, VirD4 is essential for DNA transfer to occur. In contrast, we suggest here that the hexamer of dimers architecture is representative of a T4SS in its pilus biogenesis mode.

Reviewer: Have the authors observed 3D-classes that look similar to the ones published by the same group?

We observe side views typical of the double-barrelled architecture in our cryo data set. We have added in the main text: "*Interestingly, a minority of the 2D classes in our cryo-EM data set displayed the typical side-views of the double-barrelled architecture observed in the NSEM structure (ED Fig. 1w), indicating a small number of double-barrelled particles being present*". In the legend to ED Fig. 1w, the way these 2D classes were obtained is provided. However, the views of the double-barrelled form are too rare to obtain its 3D reconstruction. A challenge for the future will be to find conditions that shift the equilibrium between the two T4SS forms towards the double-barrelled form so that enough particles can be collected to solve its structure.

Reviewer: What can induce this conformational change for VirB4 (which is not caused by VirD4). Is this in any way related to the functionality of the system? Or, is the previously published structure now in question?

Not at all. We believe the Low et al. (2014, Nature) double-barrelled architecture is relevant to T4SS function. Not only do we observe double-barrelled particles in our data set, but we also report here two hexamerisation modes for VirB4: trimer of dimers in TrwK/VirB4_{unbound} as in the double-barrelled structure and hexamers of dimers in the T4SS. Moreover, as the reviewer pointed out, our double-barrelled structure was confirmed by cryo negative stain. We believe that the IMC protomer acts as a modular building block assembling to form different T4SS architectures depending on function. We have added in main text: "*Thus, the IMC protomer appears to provide a building block to form various T4SS IMC assemblies*".

Comment 3:

Reviewer: - The main plasmid used in this study (pBADM11 trwN/virB1-trwE/virB10Strep rbstrwD/virB11 rbsHistrwB/virD4) and used previously by the researchers in Redzej et al., 2017, is not included in the used constructs/plasmids.

Response

The construct is now added in ED Table 3.

Comment 4:

Reviewer: - 'Apo' normally refers to ligand binding, so it appears misleading to call a sole VirB4 as 'apo state'.

Response

We now call it "TrwK/VirB4_{unbound}". We hope this new name reflects the fact this is TrwK/VirB4 on its own, i.e. unbound to any other T4SS components or any form of nucleotides.

Comment 5:

Reviewer: - The context/reference to the type VII secretion system is unclear: Type VII is a hexameric assembly (Beckham et al, 2017 referenced in Famelis et al 2019) and in the cited paper (Famelis, Calzada et al., 2019) authors show a structure of a dimeric building block of this complex. As such, both protomers show the same level of occupancy, while the ATPase motor component (of both protomers) is flexible. The other 4 protomers of the type VII secretion system are not showing different levels of occupancy, but are missing completely.

as the dimeric unit is considered to be a building block. This is also referenced in the Methods section again, but this reviewer does not understand the connection. Have the authors considered, if the imaged assembly is an intermediate structure towards a (potentially) final complex that could display the double barrel assembly that they have shown in the past? This would maybe explain why three VirB4 dimers are more stably bound.

Response

We were inspired by the concept of building blocks that the new research on T7SS seems to illustrate. However, since this part is confusing, we have removed it. As mentioned above, we now more specifically introduce the “building blocks” concept to T4SS architecture and link it to a potential interconversion of the two T4SS forms, double-barrelled trimer of dimers and hexamer of dimers.

Comment 6:

Reviewer: - The authors show that VirB11 interacts with VirB4 (from pKM101 and not R388; ED Fig 6(b) . This interaction is disrupted upon mutating various residues in VirB4 or VirB11 and the authors presume this interaction to be disrupted by the high concentration of detergents used for membrane extraction of the entire complex. What happens to these complexes (VirB11:VirB4) upon exposure to the detergent-containing T4SS purification conditions? Since VirB11:VirB4 seem to copurify quite efficiently, have the authors tried to add VirB11 to their purified protein preparations after extracting/purifying the T4 complex from the membrane? According to the methods, these complexes (VirB11:VirB4) were purified by tandem-purification using a StrepTrap column first (VirB11-Strep) followed by IMAC (VirB4-his). Why does VirB4-his show up in lanes 2-4 after Strep purification if complexes do not form? (or is the labeling wrong?) SDS-PAGE: Proteins separated in Lane 1 should be adjusted to similar levels as in lane 2-4 based on the first affinity step (VirB11-Strep) (faster migrating bands are visible also in lane 2-4, but strongest in lane 3).

Response

These are also very important questions which we will answer one by one:

Reviewer: What happens to these complexes (VirB11:VirB4) upon exposure to the detergent-containing T4SS purification conditions?

As recommended by the reviewer, we have tested the effect of the detergents we used to purify the T4SS on the TraB/VirB4-TraG/VirB11 complex stability. This result has been incorporated in ED Fig. 1d and reproduced at left. As can be seen, detergents disrupt the interaction, consistent with our interpretation that VirB11 might dissociate from the T4SS complex in the presence of detergents.

Reviewer: Since VirB11:VirB4 seem to copurify quite efficiently, have the authors tried to add VirB11 to their purified protein preparations after extracting/purifying the T4 complex from the membrane?

The VirB11:VirB4 complex copurifies readily only because the complex is purified in the absence of detergents (neither VirB4 or VirB11 are membrane proteins). Since the T4SS is purified in detergents and we now know that the VirB11/VirB4 interaction is disrupted by detergents, we can safely conclude that adding VirB11 to the purified T4SS will not result in VirB11 binding to the system.

Reviewer: According to the methods, these complexes (VirB11:VirB4) were purified by tandem-purification using a StrepTrap column first (VirB11-Strep) followed by IMAC (VirB4-his). Why does VirB4-his show up in lanes 2-4 after Strep purification if complexes do not form?

The reviewer is absolutely right and we apologise for missing to describe this part of our Methods. It has now been rectified. The purification of the VirB11-VirB4 complex is still provided since it is used to assess the effect of detergents on complex stability. However, we have now incorporated the section describing the pull-down with His-VirB4, which we used to assess the effect of mutations on complex formation. Note also that many more mutants are now shown.

Reviewer: SDS-PAGE: Proteins separated in Lane 1 should be adjusted to similar levels as in lane 2-4

We have now adjusted VirB4 to similar levels (ED Fig. 4c). The anti-His and anti-Strep Westerns are included in ED Fig. 4c.

Comment 7:

Reviewer: - The authors should include a Supplementary Figure showcasing the used data processing pipeline. This should include also images of the main maps/classes obtained along the way (e.g. critical 2D/3D classifications), that were obtained before the final maps. This will ease the following of the processing pipeline. In it's current version, it is difficult to assess any of these steps as no intermediate results are shown.

Response

We apologise for the lack of information. We now describe all intermediate steps and have included a workflow in ED Fig. 1f-i.

Comment 8:

Reviewer: - The authors are encouraged to deposit not only final sharpened maps, but also 3D-refinement maps, half-maps and masks. In particular, it would be important do also deposit the entire C1 EM map of the OMCC, shown in ED Fig 2a.

Response

We have deposited all maps, half maps, and masks and have summarised all EMDB and PDB depositions in ED Table 1a and 1b. The maps are also now rigorously and consistently named by naming first the region of the structure, the symmetry applied (or lack thereof), and the resolution. For example, the C1 map for the IMC-Arches-Stalk is called "IMC-Arches-Stalk C1 6.18 Å map". Overall, we have deposited 11 maps including the one mentioned above by the reviewer and also the *Ab Initio* model for the IMC-Arches-Stalk. For each map, we also state their use in ED Table 1a.

Comment 9:

Reviewer: - Angular plots for reconstructions/particles should be provided for all maps (including for the C1 maps)

Response

We have included all angular plots including for the intermediate maps and the C1 maps.

Comment 10:

Reviewer: - It would be helpful, if the authors can make a table referring to their maps/models/depositions and clearly distinguish between what components (and corresponding residues) are built de novo in which map, which component/region is rigid body fitted and so on. The assembly of an IMC model should contain the name 'composite' in it, to make it clear that not all contact points between various components are clearly observed in the data.

Response

We have created such a table (ED Table 1a). We now refer to the hexameric IMC model as a “composite IMC model”. We have also changed “T4SS model” to “composite T4SS model”.

Comment 11:

Reviewer: - Are the OMCC complexes always attached to the IMC, or do the authors also observe particles in which these two sub-complexes are separated?

Response

We have not observed any OMCC alone.

Comment 12:

Reviewer: - Which experiments were performed in order to validate the high-symmetries observed in the OMCC (C14 and C16 symmetries)? Processing of the C1 map and validation tests for correct symmetries are not described and can therefore not be assessed. Why does the OMCC C1 map actually appear like a C2 map (based on density thickness of opposing densities (ED, Fig2/a, symmetry sections) and can the authors exclude wrong Euler angle assignment? (Note, the C1 maps shown in ED Fig2 were not part of this manuscript submission and details of the processing were not described but should be provided). Have the authors checked a potential model/map bias during reconstruction (what is the outcome of the C1 reconstruction, when stronger low-pass filters are applied for the reference maps (such as 40Angstroms or 60Angstroms, or reference maps with different symmetries were used).

Response

These are very important questions which we will answer one by one:

Reviewer: Which experiments were performed in order to validate the high-symmetries observed in the OMCC (C14 and C16 symmetries)?

We now report a bias-free assessment of the symmetry of the OMCC map obtained without imposing any symmetry and mask using the rotational autocorrelation function option in IMAGIC. The two results are shown below and reproduced in ED Fig. 1r for two sections of the OMCC C1 3.61 Å map, one for the O-layer shown at left here and one for the I-layer shown at right:

This analysis clearly shows 14 peaks for the O-layer and 16 peaks for the I-layer. As the reviewer rightly noticed, there is an anisotropy clearly visible particularly in the I-layer. As suggested by the reviewer, to further validate the OMCC symmetry, we generated maps for the I-layer where C14, C15, or C16 symmetry was applied and we assessed which map has most interpretable density. The results are shown below and in ED Fig. 1s. As can be seen, C16 yields a much better map.

In our opinion, the symmetry of the O- and I-layer is now firmly established.

Concerning the symmetry of IMC, the Arches, and the Stalk, the same IMAGIC analysis was performed but on the IMC-Arches-Stalk C1 6.18 Å map for IMC and Arches, and on that same map and the Stalk C1 3.71 Å map for the Stalk, IMAGIC clearly reporting on the angles between protomers in the IMC and the Arches (60 degrees), and between monomers in the Stalk (72 degrees) (ED Fig. 1t-v)

Reviewer: Processing of the C1 map and validation tests for correct symmetries are not described and can therefore not be assessed.

We apologise for this omission. It is now fully described in the Methods section. It has also been deposited.

Reviewer: Why does the OMCC C1 map actually appear like a C2 map (based on density thickness of opposing densities (ED, Fig2/a, symmetry sections) and can the authors exclude wrong Euler angle assignment?

As pointed out by the reviewer, there is indeed a C2 aspect to the OMCC C1 3.61 Å map. That's because the angular distribution is not uniform as is now reported in ED Fig. 1j. Note that this map was only used for symmetry analysis. The high-resolution maps for the O-layer and the I-layer were obtained using a different protocol (see ED Fig. 1g)

Reviewer: Have the authors checked a potential model/map bias during reconstruction (what is the outcome of the C1 reconstruction, when stronger low-pass filters are applied for the reference maps (such as 40Angstroms or 60Angstroms, or reference maps with different symmetries were used).

The OMCC C1 3.61 Å is an entirely unbiased map that has been generated by 3D classification with no symmetry applied. We apologise for not having described the map in our original version. As pointed out above, we applied C14, C15, or C16 symmetry to generate 3 different maps of the I-layer and the C16 map is clearly better resolved.

Comment 13:

Reviewer: - Model building is reported to be done into 4 maps obtained at a resolution range from 2.5 to 3.7 Å. However, in the methods authors state that the model for the periplasmic VirB8 trimer (Arches) was built into a 6.2 Å C1 reconstruction (the best map available for this region) using rigid-body fitting of coordinates obtained from ROBETTA. Authors claim the resolution was "sufficient" to build a model including side chains for most parts of the structure (referencing Fig. 1b, which includes e.g. the periplasmic VirB8 trimer and the cytoplasmic face of inner VirB4 (res: 502-592) interacting with VirB11 to mention two example areas). The alpha helices (which are typically the features that are resolved best at low resolution) are barely

visible in both of these areas, yet the models in these areas contain side chains (which are not supported by the experimental data). At the same time, authors do not model but only speculate about a fourth VirB8 copy, although the density corresponding to the TMH of this putative fourth VirB8 copy is considerably better resolved (in the IMC Protomer C1 map at 3.7 Angstrom map but also in the IMC-Arches Stalk C1 map at 6.2 Angstrom) than the entire periplasmic VirB8 area and the cytoplasmic face of inner VirB4. On which grounds do authors decide to build atomic models? How can the authors be confident that the periplasmic VirB8 and cytoplasmic VirB4 (res: res: 502-592) density has been interpreted correctly?

Response

These are important questions which we will answer one by one:

Reviewer: Model building is reported to be done into 4 maps obtained at a resolution range from 2.5 to 3.7 Å. However, in the methods authors state that the model for the periplasmic VirB8 trimer (Arches) was built into a 6.2 Å C1 reconstruction (the best map available for this region) using rigid-body fitting of coordinates obtained from ROSETTA. Authors claim the resolution was “sufficient” to build a model including side chains for most parts of the structure (referencing Fig. 1b, which includes e.g. the periplasmic VirB8 trimer and the cytoplasmic face of inner VirB4 (res: 502-592) interacting with VirB11 to mention two example areas). The alpha helices (which are typically the features that are resolved best at low resolution) are barely visible in both of these areas, yet the models in these areas contain side chains (which are not supported by the experimental data).

We have corrected the statement and now say: “*We present here a single particle cryo-Electron Microscopy (cryo-EM) structure of a T4SS complex from the R388 plasmid that comprises all 4 sub-complexes: OMCC, Stalk, Arches, and IMC (Fig. 1a-c; ED Fig. 1b-i). Near-atomic resolution was achieved for all except for the Arches sub-complex (ED Fig. 1j-q; ED Table 1).*”

Side chains have been removed from the TrwG/VirB8_{peri} domains as they are not resolved in the density. We have also removed the side chains for all residues of TrwK/VirB4 for which side chain densities is not observed in the 502-592 region and elsewhere. Also, more generally, we have removed residues for which there is no density. A summary is provided in ED Fig. 1b. We thank the reviewer for his/her demand for more rigour and precision.

It is also important to note that the PDB requires the deposition of coordinates with residues named corresponding to the sequence of the protein. It used to be that we would build residues without side chains as ALA. However, this is no longer allowed which means that the name of the residues will appear, but where the side chains are missing the corresponding atoms are removed.

Reviewer: At the same time, authors do not model but only speculate about a fourth VirB8 copy, although the density corresponding to the TMH of this putative fourth VirB8 copy is considerably better resolved (in the IMC Protomer C1 map at 3.7 Angstrom map but also in the IMC-Arches Stalk C1 map at 6.2 Angstrom) than the entire periplasmic VirB8 area and the cytoplasmic face of inner VirB4. On which grounds do authors decide to build atomic models? How can the authors be confident that the periplasmic VirB8 and cytoplasmic VirB4 (res: res: 502-592) density has been interpreted correctly?

We are a little puzzled by this statement. We didn't build the fourth TrwG/VirB8_{tail} because the density does not show side chains; so we cannot be sure it is actually a fourth TrwG/VirB8_{tails}. It could be VirB2 or the TM of VirB10. We simply don't know and therefore it is legitimate not to build it. This is already stated in our text.

The case for TrwG/VirB8_{peri} is completely different but our decision to build is just as valid: In the IMC-Arches-Stalk C1 6.18 Å map (our new name for the 6.2 Å map of the IMC, Arches, and Stalk), we observe density in which the known structures of the TrwG/VirB8_{peri} domains fit; Moreover, the interfaces between TrwG/VirB8_{peri} domains are strikingly similar to interfaces observed in independently determined VirB8_{peri} domain structures (mentioned in main text). This provides a very high degree of confidence we are interpreting the density correctly. This was explained in our previous version of the manuscript. But, clearly, it needed to be more clearly stated, which we have now done in the Methods by saying: “For the Arches, the IMC-Arches-Stalk C1 6.18 Å map (ED Fig. 1n) was used, the best map for this region. The resolution was however high enough to clearly show secondary structural features into which 9 homology models of the TrwG/VirB8 periplasmic domain (TrwG/VirB8_{peri}; residues 95-231; obtained using ROSETTA) were docked as rigid bodies using CHIMERA. The high cross-correlation (0.83) indicated the good correspondence between the map and the model. Confidence in the correctness of this model was increased when it was realised that the TrwG/VirB8_{peri} domains come together in a manner that has been observed before (see main text). Note that the side chains are removed from our final model of the Arches as they are not defined in the density”.

For VirB4, we are very confident the structure is correct for the following reasons: 1- We report here two independently-determined structures of VirB4: one which we named “Apo-form” (now termed TrwK/VirB4_{unbound}) and the other bound to the T4SS; 2- The TrwK/VirB4 C-terminal domain is very similar to that of the VirB4 from *T. pseudethanolicus* which we solved some years ago by X-ray crystallography (PDB entry code: 4ag5; Wallden et al. PNAS 2012). Regions that cannot be built because of poor density for main chain atoms have not been built and in regions with poor side chain densities but interpretable main chain densities, only main chain atoms are reported.

Comment 14:

Reviewer: - The authors identify that the VirB9/10 NTDs are complexed in the I-layer of the T4SS, which has a 16-fold symmetry. In contrast, the O-layer does only contain 14 copies of VirB9/10 CTDs, revealing a symmetry mismatch in this part of the T4SS structure. Further, the authors show that the two VirB9 CTDs missing in the O-layer are the ones assuming diametrically opposite positions in the I-layer complex. The authors conclude that 2 heterodimeric VirB9/10 NTDs (alpha 1 for VirB10) insert into the I-layer, “while the CTDs of these proteins are not inserted in the O-layer”. This raises the very intriguing question where these CTDs end up. The authors speculate that the NTD of VirB10 may assume a scaffolding function(ED. Fig. 5n) by connecting the arches (below the I/O-layer) in the periplasm to the VirB4 cytoplasmic ATPase (thereby bridging the inner membrane). Why do authors suggest the VirB10 NTD rather than the VirB10 CTD (and why not VirB9?), which already contains a hydrophobic helix inserted into the OM that could be repurposed for this function? More importantly however, the reasoning that not only the two VirB9 but also the two VirB10 emerge from subunits located on opposite sides of the T4SS I-layer requires further validation. (Note: Could the authors provide a description of how the unaccounted density was visualized? In the map provided (6.2Angstrom map) the unaccounted density does not show this long extension towards VirB8 and is about only half the length shown). Could it be that the unaccounted density corresponds to a yet to be identified protein component of the T4SS?

Response

This is a fascinating comment to which we respond below.

However, before responding fully, we would like here to emphasize that we mention non-assigned densities only because we believe it might be of interest to the field to know that there are still ill-defined regions that could potentially require their input for identification. However, we wish to keep further speculations to the more suitable setting of a review. We simply don't know what these densities consists of.

Reviewer: More importantly however, the reasoning that not only the two VirB9 but also the two VirB10 emerge from subunits located on opposite sides of the T4SS I-layer requires further validation.

We believe that what is shown in ED Fig. 3m clearly shows where the additional two VirB9 NTDs are inserted. The OMCC is the best defined part of the structure where the resolution is highest. Therefore, we do not feel there is any ambiguity in the positioning of these two subunits. As we mention in the text, very similar observations were reported in the *Legionella* OMCC where there are three additional subunits regularly inserted in what is the equivalent of the I-layer.

Reviewer: Could the authors provide a description of how the unaccounted density was visualized? In the map provided (6.2Angstrom map) the unaccounted density does not show this long extension towards VirB8 and is about only half the length shown

We apologise for the lack of clarity. There are indeed two stretches of density as the reviewer indicates, but they do connect when the map is contoured to a lower level. We attach at left a figure showing that ($\sigma = 0.04$ and $\sigma = 0.02$; TrwG/VirB_{peri} and what we propose to potentially be a

sequence of TrwE/VirB10_{NTD} are shown here in ribbon color-coded in yellow and green, respectively. The arrow points to the part of the density that links the two parts of TrwE/VirB10_{NTD} when the map is contoured at a lower level). We have now modified our legend to ED Fig. 3p to provide a more extensive account of what we observe and also the reasons why we think this density may correspond to VirB10_{NTD}. We however also acknowledge that this density could be ascribed to other proteins as the reviewer suggests. It now reads: “Two extra densities are seen at $\sigma 0.04$ which merge into one at $\sigma 0.02$. These indicate a structure that makes contact with TrwG/VirB_{peri} and that is flexibly (shown in dashed lines) connected to another structure that makes contact with the TrwI/VirB6 TM helices and with the two subunits of the TrwK/VirB4 dimer. The density was too poor to be assigned but could correspond to TrwE/VirB10_{NTD}, which is known to not only make a major part of the OMCC but also has an IM TM and a cytoplasmic tail that, in other T4SSs, has been known to interact with VirB8, VirB6 and VirB4¹²⁻¹⁴. However, it could be that this stretch of density may correspond to different proteins”. Such a cautious statement is justified also by the fact that, as mentioned below, TrROSETTA does not detect a co-evolution signal between VirB10_{NTD} and VirB8, VirB6, or VirB4.

Comment 15:

Reviewer: - The peripheral periplasmic area (Fig. 1a, IMC Protomer c1 map 3.7 Angstrom map but also in the IMC-Arches Stalk c1 map 6.2 Angstrom) surrounding the trimeric VirB8 proteins not only provides space for a fourth VirB8 copy but could eventually also host another soluble domain. Maybe the soluble domains of either VirB9/10 are forming a complex with the trimeric periplasmic VirB8 domains to recruit the fourth VirB8 copy at the very edge of the T4SS structure. Coevolution analysis between VirB9/VirB8 and VirB10/VirB8 could provide useful information about this.

Response

Indeed, there is un-assigned density between the trimeric VirB8 periplasmic domains and the 4th one. As shown in the picture at left (a reproduction of EF Fig. 1o), this density connects also to the N-terminal tails of VirB8. So, we believe that this unassigned density would be best interpreted as the rather long sequence linker between the VirB8 periplasmic domain and its N-terminal tail (33 residues). Three or four linkers might account for this density. However, in the absence of side chains, we cannot ascertain what it is. And co-evolution analysis between VirB9/VirB8 or VirB10/VirB8 does not provide confident prediction. So we have not included this region in our model.

In ED Fig. 3o, we have toned down the potential interpretation for this un-assigned density. We state: “*This may indicate the presence of a fourth TrwG/VirB8 subunit. However, the densities were too poor to be assigned and we remain unsure as to potential interpretations and assignments.*”

Comment 16:

Reviewer: - Coevolution analysis provides robust validation for the well-resolved VirB5/6 interface in the stalk, the VirB9/10 interface in the O/I-layer, the large VirB4central/VirB4central interface and the VirB3/VirB4 interface, however interactions that rely on structural plasticity are seemingly not captured well by this method (e.g. the N-termini of VirB8 forming either one continuous or two helices do show coevolving residues only for the continuous helix in the corresponding VirB4outside). How can this be explained? As a control and validation of the paucity of interactions between subcomplexes, authors should also provide coevolution analysis of proteins forming plastic intermolecular interactions such as the VirB8 periplasmic trimer and of those belonging to subcomplexes that are not immediately interacting in all areas of the presented structure (e.g. VirB6/VirB3 and VirB6/VirB4).

Response:

As mentioned above, we switch from GREMLIN to TrROSETTA for the analysis of co-evolution across MSAs. For VirB4/VirB8, while GREMLIN returned residue pairs only for VirB8_{tails} MolB (VirB8_{tailsB}), the much more accurate and sophisticated TrROSETTA analysis returns pairs not only with VirB8_{tailsB} but also with VirB8_{tailsC} (ED Fig. 3s).

However, the reviewer makes a very valid point that flexible/plastic interactions appear to have a weak co-evolutionary signal. This has been documented in an excellent paper by Mintseris and Weng (2005, PNAS) which shows that surfaces involved in transient interactions have weak co-evolution signals. We believe that this is why the VirB2 assembly site of VirB6 does not show up in the co-evolution analysis. We now state in ED Fig. 4d legend: “*The TrROSETTA analysis of VirB2-VirB6 did not detect any pairs involving VirB6 residues in this putative assembly site. This is not surprising: transient interactions are known to provide only weak co-evolutionary pressure¹⁸ and one indeed expects VirB2 subunits to make only transient and weak interactions with the assembly site so as to not prevent incoming VirB2 subunits from displacing already assembled subunits at the base of the pilus.*” It is important to note at this stage that we did mutate the assembly site residues of VirB6 and found mutations to affect conjugation (see Reviewer 1 comment 3, Reviewer 2 comment 21, and Reviewer 3 comment 1).

Regarding controls, it is known that TrROSETTA scores cannot be compared from one interaction to another. Thus, a much better control is to be found internally i.e. within interactions by comparing scores of residue pairs shown to interact in the structure with residue pairs that do not. Let’s take, for example, the TrROSETTA results obtained for the VirB4 and VirB8 interaction again. In VirB8, only the N-terminus (the VirB4_{tail}) is seen interacting with VirB4 in our structure, the periplasmic domain (VirB4_{peri}) not interacting. As

can be seen from the full result data set presented in the SI, while the top-scoring pair has a score of 0.76, the first pair that involves a residue of VirB8_{peri} has a score of 0.17 and is ranked 231, the first 230 pairs all involving residues in VirB8_{tail}. The next pair listed that involves a residue in VirB8_{peri} is ranked 291 with a score of 0.14. Clearly, the overwhelming number of top-ranking pairs involve pairs when a VirB8_{tail} residue is involved. It indicates that a score below 0.17 (or possibly 0.14) for this pair of proteins indicates no co-evolution detected. This threshold would be different for each pair because TrROSETTA scores depend on the protein pairs under examination and is sensitive to parameters such as the sampling of protein sequences in the MSA. So, we therefore provide here self-consistent controls for all interactions. We have added in the main text: “*Finally, regions of structures that do not interact score very poorly compared to regions that do, providing internal validation of the co-evolution results (see SI).*”

As suggested by the reviewer, we ran TrROSETTA on VirB8/VirB10, VirB6/VirB3 or VirB6/VirB4, and the scores returned are much lower than for any of the interactions we report here, suggesting that there is no co-evolution signal between these proteins.

Comment 17:

Reviewer: - The authors write that the VirB4 AAA+ motor has 6-fold symmetry and that the interface between subunits is small. Can the authors elaborate how the 6-fold symmetry is related to the function of this AAA+ motor? Understandably, better characterization is hampered by partial occupancy for which the reason is unclear. However, the overwhelming majority of AAA+ structures show that co-existing ATP/ADP nucleotides cause AAA+ motors to be asymmetric. And that the asymmetry is crucial for their function. In cellulose, nucleotides are always present (at different ATP/ADP ratios). It appears the 6-fold arrangement is a consequence of the configuration of the VirB4 ATPase in the absence of nucleotide-dependent interprotomer contacts, the rather low global resolution (6.2 Angstrom) achieved and the lack of VirB11. On a more general note, it probably would be helpful if the authors use the term ‘symmetry’ only in cases, where there is high-resolution structural evidence provided, and use ‘oligomericity’ (such as hexamer) to claim a structural arrangement from several protomers.

Response

We thank the reviewer for his/her very helpful insights on AAA+ ATPase. Indeed, AAA+ proteins cycle through conformational states that breaks down the symmetry of the hexamer.

However, we are unsure how to respond to this comment. We certainly no longer mention six-fold symmetry. In fact, we embrace wholeheartedly this reviewer’s suggestion that we should use “oligomericity” to describe the various oligomerisation states with observe. The new main text reads: “*The IMC is made of 6 protomers, each including 1 TrwM/VirB3, 2 TrwK/VirB4, and the 3 N-terminal tails of TrwG/VirB8 (TrwG/VirB8_{tails}). Three of these protomers are significantly occupied while the occupancy of the 3 others is weaker (ED Fig. 1t). All protomers were related by an angle of 60° (ED Fig. 1t,u). The IMC is thus a hexameric structure with compositional heterogeneity (i.e. variable occupancy of its constituent protomers) as defined by Huiskonen¹⁷. The Arches, comprising the TrwG/VirB8 periplasmic domains (TrwG/VirB8_{peri}), also form an hexameric assembly with variable occupancy (ED Fig. 1t,u). The Stalk is composed of pentamers of each TrwJ/VirB5 and TrwI/VirB6 (ED Fig. 1v). The O-layer, made of full-length TrwH/VirB7 and the C-terminal domains of TrwE/VirB10 (TrwE/VirB10_{CTD}) and TrwF/VirB9 (TrwF/VirB9_{CTD}), and the I-layer, composed of the N-terminal domains of these proteins (termed TrwE/VirB10_{NTD} and TrwF/VirB9_{NTD}, respectively) form tetradecameric and hexadecameric assemblies, respectively (ED Fig. 1r,s).*”

Comment 18:

Reviewer: - The fitting of the VirB11 hexamer model to the VirB4 structure is informed by coevolution analysis. The residues listed in Fig. 4a belong to pKM101-TraB/VirB4. It would be helpful if the authors would highlight interfacial/mutated residues identified by coevolution analysis/targeted by site-directed mutagenesis also in the sequence alignments (ED. Fig. 4).

Moreover, showing the sequence alignments of the missing VirB2/11 would also be useful because VirB11 has been subject to mutational analysis (Fig. 4a) and VirB2 homologous sequences have been used to infer the putative structure of the R388 trwK T4SS pilus.

Response

It is important to note that the fitting is no longer informed by co-evolution analysis. Indeed, as explained in detail in the response to Reviewer 1's comment 3, we completely changed our approach for examining the VirB4/VirB11 interaction. Co-evolution analysis as implemented by TrROSETTA is now used to validate a VirB4/VirB11 model generated by ALPHAFOLD, a software that is shown to provide structure models to atomic accuracy. We also use a second validation method, site-directed mutagenesis of the ALPHAFOLD model interface between the two proteins.

To answer this reviewer's comment 18: we now provide the alignments for VirB11 (in SI with all the other alignments). We have also highlighted residues interacting with VirB4. Also, on recommendation from the other reviewers, we have expanded our mutational analysis of the VirB4/VirB11 interface and a zoom-in figure of the interaction and residues mutated is now provided in Fig. 4a. However, in contrast to VirB4/VirB11 where ALPHAFOLD provided models for the single proteins which were accurate (as assessed against known structures), for VirB2, we have observed a discrepancy between the ALPHAFOLD model for TrwL/VirB2 and the experimentally-derived structure of VirB2 of the F and pED208 system (published by Costa et al. (2016) Cell). It could be that ALPHAFOLD does not yet produce accurate models for all membrane proteins. Therefore, we no longer present a model of TrwL/VirB2 and therefore there is no need to present a VirB2 sequence alignment. This does not in any case weaken our conclusions: indeed, in the co-evolution analysis by TrROSETTA between VirB2 and VirB6, the 50 top scoring pairs involves residues in VirB6 that are located in VirB6 α 1, one of the two VirB6 TM helices, making it a strong candidate for binding/recruiting VirB2 (itself a membrane protein; see main text), an hypothesis we subsequently test using site directed mutagenesis of VirB6 residues forming this putative VirB2 binding/recruitment site.

Comment 19:

Reviewer: - The authors reason on the basis of superposition analysis (and the resulting clashes) that the interactions site for VirB3 and VirB8tail with VirB4central and VirB4outside, respectively, are unique and cannot be interchanged (ED Fig.3g+h). However, the three modelled VirB8tail structures show a high degree of structural plasticity, especially in their N-termini, which could as well be used to argue that VirB8tails adapt to their local environment. The unique positions of both proteins are much more supported by the EM maps than the superposition analysis.

Response

The EM analysis is indeed the best supporting evidence for the position of VirB3 and VirB8_{tails}. Our superposition exercise was meant to provide some additional rationale as to why VirB3 does not bind to VirB4_{outside} and why the VirB8_{tails} do not bind VirB4_{central}. However, we can see the reviewer's point and therefore we have removed these panels and associated texts.

Comment 20:

Reviewer: - The authors make extensive use of coevolution analysis and use it as a replacement for the biochemical validation of the structural data (e.g. for the well-resolved VirB5/6 interface in the stalk and VirB9/10 interface in the O/I-layer). While some residues localize in very close proximity (indicating direct interactions), others appear more far away from each other (ED. Fig. 6a). It would be helpful to also provide the distances between the observed residue pairs to judge whether the interactions are direct or indirect. Moreover, how do authors interpret large distances between residue pairs and how do they contribute to infer structural models?The authors write: "Based on the constraints imposed by the distances between coevolving residue pairs,...". How big are these distances? Can this be explained in the methods?

Response

As stated in our response to Reviewer 1's comment 3, we brought a number of modifications to this section. Previously, for the co-evolution analysis, we used TrROSETTA for VirB5/VirB6 and GREMLIN for the other interactions. However, TrROSETTA is a much more sensitive and accurate way to look at co-evolution across interfaces, and we thus decided to switch to TrROSETTA entirely. TrROSETTA is a Deep Learning network trained on tens of thousands of proteins in the PDB to convert the coevolution signals detected in the MSA of a protein and its homologs to residue-residue distances. In SI, we now list all residue pairs with TrROSETTA contact probability ≥ 0.05 for all heterologous interactions (VirB3-VirB4, VirB5-VirB6, VirB4-VirB8 and VirB9-VirB10) in our structure: this report includes not only the residue pairs but also, as requested, the TrROSETTA score and the corresponding all atoms and $C\alpha$ - $C\alpha$ distances in our structural models. Top residue pairs shown in green in the SI lists are the residues we map onto our structural models shown in Fig. 3g (VirB5-VirB6), ED Fig. 3q-s (VirB3/VirB4, VirB4/VirB8 and VirB9/VirB10) and Fig. 4a (VirB4/VirB11). We could have mapped many more but it would have made the figures very messy. As can be seen from the complete lists in SI, most $C\alpha$ - $C\alpha$ distances in the top-scoring pairs are below 12 Å which corresponds to the average $C\alpha$ - $C\alpha$ distance seen in interfaces between proteins. For example, for VirB3/VirB4, among the first 100 top-scoring pairs, we observe only 9 pairs with distances > 12 Å (in yellow in corresponding sheet in SI) and most of those are < 12.5 Å. It might be that these pairs might reflect a different conformation of the concerned interface. But given their very low ranking, we are reluctant to endorse such an interpretation. Predicting the couple of hundred interacting residue pairs from hundreds of thousands non-interacting residue pairs (signal to noise ratio $\sim 1:1000$) is indeed a challenging task, and we indeed expect some rare false predictions arising from noise. We have now added in the ED Fig. 3q-s legend: "*As can be seen from the list in SI, going down the list, distances greater than the generally accepted 12 Å limit for $C\alpha$ - $C\alpha$ distances between interface residues^{15,16} are found but rank very poorly (pairs in yellow in SI).*"

Comment 21:

Reviewer: - It would be helpful if the authors would elaborate a bit more on the implications of their structural findings for conjugation/substrate transport/assembly such as the role of the VirB8 arches anchoring in outerVirB4, the limited connections seen between IMC and stalk. The model of pilus biogenesis is inferred from bioinformatics (using a combination of homology modeling, docking and coevolution analysis) and provides an exciting working hypothesis. However, the proposed model cannot replace structural data on the fully-assembled, DNA-translocating T4SS and the proposed mechanism of biogenesis must therefore be validated/substantiated by additional experimental data.

Response

The mechanism of pilus biogenesis that we propose is indeed novel and exciting as the reviewer points out. We agree more validation is required.

To that effect, we have started to probe our mechanistic model for pilus biogenesis by targeting the proposed/putative VirB2 binding/recruitment site and assembly site on VirB6 for mutational analysis. Three single mutations were made in the recruitment site (a small site consisting of TrwI/VirB6 $\alpha 1$) and six double-mutations were made in the assembly site (a much larger site). Both affect conjugation, one (recruitment site) decreasing it and the other (assembly site) increasing it. The latter effect is, in our opinion, very interesting because mutations that increase biological readout are often of great biological significance. We next tested the idea of attempting to block VirB2 subunits from reaching their assembly site by introducing bulky residues, Trp, in between the recruitment and assembly sites. Four double mutants were made and tested: all decrease conjugations. All results are shown in Fig. 4e and reported in Reviewer 1's comment 1. These results are all consistent with our model.

This is only the beginning. There is no doubt that deciphering the multiple steps required for the machinery to add layers upon layers of insoluble (membrane-bound) subunits to make a

soluble pilus assembly will require a multitude of additional experiments and therefore time. However, as all reviewers agree, our structure is exceptional and, maybe more importantly, provides for the first time a rationale for a plausible model for the biogenesis of these important filaments that play fundamental roles in the spread of antibiotic resistance genes.

Reviewer: It would be helpful if the authors would elaborate a bit more on the implications of their structural findings for conjugation/substrate transport/assembly such as the role of the VirB8 arches anchoring in outerVirB4, the limited connections seen between IMC and stalk.

This is an interesting issue. We believe the Arches serve as a scaffold between the OMCC and the IMC/stalk. Perhaps more importantly, we believe that the Arches serve as a scaffolding ring through which the pilus is directed (now stated in the text in corresponding ED Fig. 4I). Given space constraints and our reluctance to speculate, we feel this is all we can say. Further discussions might be kept for a review.

Comment 22:

Reviewer: - Moreover, the proposed model raises important yet unanswered questions: Why does coevolution analysis provide the interfacial residues for the VirB2 recruitment site (in the inner membrane) but not the ones deemed to form the pilus base on top of the VirB6 pentamer (Fig. 4d+e)? What does coevolution look like for other proteins located in the area of interest such as VirB3. How can cytoplasmic VirB4 proteins extract VirB2 protomers from the membrane and promote their transport on the periplasmic side and eventually across the membrane? How can the recruitment of VirB2 subunits cause the displacement of the VirB2 subunits forming the pilus base?

Response

These are very important questions which we will answer one by one:

Reviewer: Why does coevolution analysis provide the interfacial residues for the VirB2 recruitment site (in the inner membrane) but not the ones deemed to form the pilus base on top of the VirB6 pentamer.

It has been shown that strong interactions display strong co-evolutionary constraints while transient interactions display weak co-evolutionary constraints (Mintseris and Weng (2005) PNAS, 102: 10930-5). Binding of VirB2 to VirB6 will likely impose strong co-evolutionary constraints in the binding/recruitment interface. And indeed, we find that the 50 top-ranking VirB2/VirB6 residue pairs involve VirB6 residues located in its $\alpha 1$ TM helix, making this site a strong candidate for VirB2 recruitment. Site-directed mutants of $\alpha 1$ residues reduce conjugation, consistent with this hypothesis.

In contrast, the surfaces of VirB6 potentially involved in VirB2 assembly require to make only transient interactions with VirB2. This is required because pilus biogenesis is a process in which layers upon layers of VirB2 subunits are added, one layer inserting under the other until a fully-assembled pilus is formed. For an incoming layer of VirB2 subunits to insert under the previously-assembled layer, this previously-assembled layer cannot bind strongly to the assembly surface: if it were, pilus biogenesis would not occur, incoming VirB2 subunits being unable to insert under the previously-assembled layers. In other words, there is a requirement for the assembly site to make transient interactions with VirB2s. Since transient interactions are known to display very weak co-evolutionary constraints (see Mintseris and Weng (2005)), we do not expect the assembly site to show up in the co-evolution analysis. We have added in the legend to ED Fig. 4d: *“The TrROSETTA analysis of VirB2-VirB6 did not detect any pairs involving VirB6 residues in this putative assembly site. This is not surprising: transient interactions are known to provide only weak co-evolutionary pressure¹⁸ and one indeed expects VirB2 subunits to make only transient and weak interactions with the assembly site so as to not prevent incoming VirB2 subunits from displacing already assembled subunits at the base of the pilus.”*

In that context, it is remarkable that this assembly site responds to site-directed mutagenesis. Additional experiments are indeed now reported. We quote here our revised main text: “*To validate this site, we made 3 pairs of double mutants (ED Fig. 4f-h), each to either acidic or hydrophobic residues. Since pilus biogenesis is essential for conjugation between bacteria to occur, these mutated T4SSs were tested for their ability to mediate conjugation (Fig. 4e). Strikingly, all display increasing conjugation rates (Fig. 4e), substitution to hydrophobic residues having a more pronounced effect.*” And later on, we describe another set of mutations aiming to block the transit of VirB2 subunit from their binding/recruitment site to their assembly site: “*To further test this model, we introduced Trp residues in-between the binding/recruitment site and the assembly site, which may form steric obstacles (referred to as “Trp-blocks”) affecting the translocation path of VirB2 subunits from their binding/recruitment site to their assembly site (Fig. 4e and ED Fig. 4f-h): all mutants exhibited decreased conjugation, consistent with expectation.*”

Reviewer: What does coevolution look like for other proteins located in the area of interest such as VirB3.

This is a wonderful comment that prompted us to run TrROSETTA on VirB2/VirB3. We indeed find a strong co-evolutionary link between VirB3 and VirB2 (the full list is now included in SI and a new ED Fig 4j reports on the location of the VirB3 residues involved in the 50 top-scoring pairs). We now state in our revised main text: “*Based on TrROSETTA analysis, another strong candidate to form a VirB2-binding site is VirB3 α 1 (ED Fig. 4j; SI). However, VirB3 is hexameric, a symmetry that does not match the symmetry of the pilus. We hypothesize that VirB2 binding to VirB3 α 1 might potentially represent an intermediate binding station for VirB2 subunits.*”

Reviewer: How can cytoplasmic VirB4 proteins extract VirB2 protomers from the membrane and promote their transport on the periplasmic side and eventually across the membrane?

In our opinion, this is a question that can only be answered by investigating the structural changes brought in on VirB4 by either VirB11 and/or by ATP binding and hydrolysis. For the moment, we do not have an answer to this question. It is one of the exciting avenues of research that the work presented here opens up. We now state in main text: “*Which regions of VirB4 act as lever remains unclear. However, potential triggers may include binding of VirB11³⁵ as well as ATP binding and/or hydrolysis.*”

Reviewer: How can the recruitment of VirB2 subunits cause the displacement of the VirB2 subunits forming the pilus base?

We do not believe the recruitment itself triggers a displacement of the VirB2 subunits layer forming the pilus base, but instead, it is the VirB4-mediated translocation of VirB2 subunits out of the IM that starts the process of VirB2 subunits being pushed towards the assembly site. At the assembly site, as stated above, the pilus base layer may make only transient interactions with the site. This means that the newly-translocating subunits would have no difficulty inserting under the base layer. We envisage a constant train of VirB6-bound VirB2 subunits being extracted from the IM by VirB4 and pushing incoming subunits towards the assembly site. Also, it is possible, as had been described previously for another secretion system (Geibel et al. (2013) Nature 496:243–246), that a favourable energy path is formed between the binding/recruitment site and the assembly site on VirB6 that drives VirB2 subunits from one to the other.

Comment 23:

Reviewer: - The concluding statement “The structure presented here thus provides the molecular framework for designing novel compounds capable of inhibiting the transfer of

antibiotic-resistance genes among pathogens...” could probable be re-phrased as the presented structure is in fact not executing DNA transfer and the motors required to energize the assembly and transport are in an inactive, eventually even unnatural state (without nucleotides). The structure rather presents the framework to design assembly inhibitors.

Response

We have amended our concluding statement, making it more general than it was. We now state: “*Thus, the near-atomic structure of a conjugative T4SSs presented here provides a structural basis for a plausible model for conjugative pilus biogenesis by T4SSs. Conjugative pili are crucial appendages, without which DNA transfer among bacterial populations would not occur and, thus, the structure also provides the means to develop anti-conjugation strategies (including assembly inhibitor design) that may result in limiting the spread of antibiotic-resistance genes among pathogens.*” Implicitly, we suggest here that one of the strategies is to design assembly inhibitors.

Additional comments

Comment 24:

Reviewer: - Figure 1: figure legend: The authors state the presence of ‘lipid densities’ but likely represent a detergent belt. Did the authors do any analyses (such as mass spec) to verify the presence of lipids?

Response

The reviewer is absolutely right. We apologise for the mistake. We have changed to “detergents”.

Comment 25:

Reviewer: - As previously mentioned, it would be very helpful to show a flowchart of all the different processing steps including intermediate results.

Response

We have now incorporated a flowchart and have submitted all maps, including intermediate ones (see ED Fig. 1f-i and ED Table 1a).

Comment 26:

Reviewer: Processing of the T4SS IMC, Stalk, Arches: - The authors performed a ‘strict selection’ of 60,722 particles out of 1,292,734 extracted particles. What were the selection criteria and which particles were excluded?

Response

In the workflow shown in EF Fig. 1h, we show the types of particles we rejected and selected.

Comment 27:

Reviewer: - The authors “confirm” a hexameric organization of the IMC. I am wondering, whether this is just a poss

Response

We have substantially rewritten this section and removed the word “confirm”. However, we state: “*The IMC is made of 6 protomers, each including 1 TrwM/VirB3, 2 TrwK/VirB4, and the 3 N-terminal tails of TrwG/VirB8 (TrwG/VirB8_{tails}). Three of these protomers are significantly occupied while the occupancy of the 3 others is weaker (ED Fig. 1t). All protomers were related by an angle of 60° (ED Fig. 1t,u). The IMC is thus a hexameric structure with compositional heterogeneity (i.e. variable occupancy of its constituent protomers) as defined by Huiskonen¹⁷.*”

Moreover, not only are the higher occupancy IMC protomers related by 60° angles, but when a composite model which we generated applying C6 symmetry is docked into the IMC-Arches-Stalk C1 6.18 Å map, three protomer models fit remarkably well into the most occupied

protomer densities (ED Fig. 1u). Moreover, when fitted into the cryo-ET map of the F system IMC, a remarkable fit is also obtained (ED Fig. 1u). It is therefore very reasonable to propose a hexameric model for the IMC. Note that we no longer use the word “C6 symmetrical” or such, we only refer to “oligomericity” states as the reviewer recommends.

Comment 28:

Reviewer: - Hexameric IMC: The authors mention to confirm a hexameric arrangement indicating to a 6-fold symmetry. They obtain a 7.6Angstroem map from 126,975 particles (C6 imposed), representing appr 760,000 asymmetric units. One would expect that the resolution would be much higher with such a dataset provided the complex is symmetric. If, however, the complex is not symmetric, the application of symmetry operators later for model building is questionable.

Response

This is a very important question and therefore we revisited our work on this C6 map. What

we found is that if we further 3D-classify the 126,975 particles with no symmetry applied, all 3D classes so obtained show only incomplete hexamers. This is showed at left where bottom views are provided. Therefore, the reason why applying C6 symmetry did result in an only marginal gain in resolution was because most particles we selected for this exercise are not complete hexamers. Therefore, we no longer present this map as it is no longer used for symmetry assessment. Instead, we used

IMAGIC to show that indeed, all IMC protomers are related by a $\sim 60^\circ$ angle and are therefore positioned along a hexagon. It is also important to note that we do describe a new C6 map generated as for the initial *Ab Initio* model for the IMC, Arches, and Stalk (starting from 60,722 particles; see ED Fig. 1h) except C6 symmetry was applied (see ED Fig. 1x): this new map was generated to derive the C6 symmetry operators (using PHENIX) and position the OMCC. These operators were used to generate the composite hexameric IMC model.

Comment 29:

Reviewer: - Apo-Trwk/VirB4: The authors report a 4.1 Angstroem map, however the fusion protein HCP1 in this construct shows a much lower resolution. Have the authors tested, whether a higher resolution could also be obtained by focused classification as a measure for image quality.

Response

We were not entirely sure that we understood what the reviewer had in mind here. So, we did both and found that including or excluding HCP from the refinement did not improve neither resolution nor image quality for Trwk/VirB4_{unbound}. We apologise in advance if we misunderstood the comment.

Comment 30:

Reviewer: - ED Table 1: deposit also C1 maps

Response

We have deposited all the C1 maps. See Reviewer 2's comment 8.

Comment 31:

Reviewer: - ED Fig1: for all EM maps the authors should provide also the corresponding EMD-deposition number next to the panel figure

(B) increase font along bars

(C) SDS-PAGE analysis of purified complex: highlight which proteins are present in the cryoEM analyses

(G) the 'highest-resolution' parts based on local resolution tests are shown in red, yet, this particular part is not representing any meaningful structure.

(I) Provide numbers of how often those particles were found compared to others - this would define the word 'rarely' in the associated figure legend

Response

We have modified the figure accordingly. For (g), we have already responded to this comment (see response to reviewer 1, minor comment 16). For (I), we now provide an estimate.

Comment 32:

Reviewer: - ED Fig2: Provide number of particles used in the C1 maps

(C) second row: how were particles selected to show the 'complete hexamer. The reconstruction actually suggests different occupancy at the hexameric position

Response

All C1 maps are now described in detail and have been deposited. The symmetry analysis that was in ED Fig. 2 is now part of ED Fig. 1 and, with the addition of IMAGIC symmetry analysis, is now described much better.

Reviewer: How were particles selected to show the 'complete hexamer'?

See response to Reviewer 2's comment 28. Prompted by this reviewer, we revisited our results and further 3D classification revealed that most particles we selected were in fact incomplete hexamers. This explains the limited resolution improvement we observe when applying C6 symmetry. The reviewer suggested that there might also be deviations from C6 which may impact negatively on averaging. We endorse this interpretation. We have withdrawn this C6 map (see comment 28). However, as explain above, our symmetry analysis using IMAGIC clearly show 6 protomers related by an angle of 60°, i.e. positioned along an hexagon, with the 6th position very weakly occupied.

Referee #3 (Remarks to the Author):

Reviewer: In this manuscript the authors present a complete structure of a T4SS involved in DNA conjugation. This is an exciting result that is of a broad interest to researchers in various fields of microbiology and structural biology. The presented data explain many of the previously poorly understood aspects of the T4SS assembly as well as pilus formation. The protein density maps generated by cryo-EM are of high quality and thus provide unprecedented level of details and novel insights. This dataset will be invaluable for future research of T4SS and DNA transfer. Structure determination and the search for coevolving residues was done by the state-of-the-art tools. There are many additional interesting and important details provided in the supplements.

Comment 1:

Reviewer: In its current form, the manuscript mainly reports the obtained EM structure, which is supported by the analysis of coevolving residues to validate known protein-protein interactions and to propose new interactions to build the overall model. While this is a good approach, it is unclear if these predictions are valid. Therefore, it would be of a great value if the authors could confirm at least some of their most exciting predictions of protein-protein interactions experimentally. The performed analysis of coevolving residues should provide guidance for such experiments. A specific point mutation introduced into one of the two putative interaction partners should result in a loss-of-function phenotype, which should be possible to suppress by the introduction of another point mutation in the second protein. The appropriate pairs of point mutations should be possible to predict based on alignments of the available sequences and bioinformatic analyses of the coevolving residues. Indeed, the authors show that in the case of VirB4 and VirB11 interaction model, certain mutations can

abrogate the interaction of those two proteins but authors should also show that the interaction can be restored by another set of mutations.

Response

For all interactions found between heterologous interfaces in our structure, we use co-evolution to validate them. However, compared to the first submission, we switched to a much more sophisticated software, TrROSETTA. TrROSETTA is a Deep Learning machine network trained on tens of thousands of proteins in the PDB to convert the co-evolution signals detected in the MSA of a protein and its homologs to residue-residue distances. This method extracts information present in the mutational landscape from sequences and therefore, as the reviewer we believe points out, there is no point testing these interactions by site-directed mutagenesis.

However, as the reviewer rightly points out, the situation is different for the VirB4/VirB11 interactions for which we have no complex structure.

In a completely different approach from what we described in the first submission where we relied on co-evolution to generate a model of the VirB4/VirB11 complex, we now instead use co-evolution methods as implemented by TrROSETTA as a validation tool to assess the quality of a VirB4/VirB11 model generated by the programme ALPHAFOLD, which has been shown to be able to produce structural models to atomic accuracy. In fact, we use three validation tools: 1- we make sure that the ALPHAFOLD model for each protein is actually correct by comparing it to their known structure (ED Fig. 4a), 2- we make sure the ALPHAFOLD VirB4-VirB11 complex model is correct by mapping the co-evolution pairs from TrROSETTA onto the ALPHAFOLD model of the complex (Fig. 4a and ED Fig. 4b), and 3- we mutate 8 residues at the interface and show that all affect the stability of the interaction (Fig. 4a).

For VirB2/VirB6, the situation is more complicated because the model that ALPHAFOLD generates for VirB2 is not the same as the structural models known for the VirB2 encoded by the F and pED208 plasmids. Instead, we turned to TrROSETTA to generate a list of co-evolving residues between the two proteins and focused our attention to the co-evolving VirB6 residues on the list. Remarkably all 50 top-scoring residues pairs involve VirB6 residues that locate in its $\alpha 1$ TM helix. The fact that all top residue pairs involve VirB6 $\alpha 1$ residues suggest that this region of VirB6 is likely a good candidate for forming a VirB2-binding site. We subsequently test this hypothesis by making mutation in $\alpha 1$ which, we show, reduce conjugation between bacteria (Fig. 4e).

Comment 2:

Reviewer: An analysis of phenotypes of point mutations (and their suppression) probing the interaction of VirB2 with VirB6 should be provided as this is the hallmark of the newly proposed mechanism of pilus biogenesis.

Response

The reviewer is absolutely right, and this is what we have done. As explained in Reviewer 1 comment 3, we have mutated the VirB2-binding site, the VirB2-assembly site and we have even sought to create obstacles, Trp blocks, between the two sites to prevent VirB2 subunits from reaching their destination site (the assembly site).

Comment 3:

Reviewer: The authors should also expand their description of the proposed model of pilus assembly. From the current description, it is hard to understand how the individual VirB2 proteins move from the “biding site” to the “assembly site” during one cycle of pilus assembly. How exactly would VirB4 be involved? It would be interesting to read ideas about this or to learn if there is additional evidence supporting this model. Related to the model, authors should also elaborate on any available data supporting the proposed O-layer opening. In

general, the manuscript should probably include an expanded introduction for the readers not familiar with T4SS and DNA conjugation.

Response

The issue of making the manuscript more understandable to a broader audience is also raised by reviewer 1. We have therefore added more background about the field of T4SS, conjugation, and the prior structural knowledge in both the summary and the introduction. We have added more details concerning the mechanism. Concerning VirB4, please refer to the response to Reviewer 2 comment 22. In our opinion, this is a question that can only be answered by investigating the structural changes brought on VirB4 by VirB11 binding and by ATP binding and hydrolysis to possibly both ATPases, certainly VirB4. For the moment, we do not have an answer to this question. It is one of the exciting avenues of research that the work presented here opens up. We have added the following sentence in main text: "*Which regions of VirB4 act as lever remains unclear. However, potential triggers may include binding of VirB11³⁵ as well as ATP binding and/or hydrolysis.*"

Concerning the opening of the O-layer, we have already documented two conformations of its channel: in the first low-resolution cryo-EM structure of the OMCC (Fronzes et al. (2009) Science), the channel was observed closed, while in the crystal structure of the O-layer (Chandran et al. (2009) Nature), the channel was semi-open. Thus, the flexibility of the linker region at the base of the helical hairpins that form the OM channel is well documented. We apologise for not having mentioned this. We now state in main text: "*As layers of pentameric VirB2 are added, the pilus grows from the bottom, pushing the VirB5 pentamer out, passing through the Arches, the I-layer (no conformational changes needed (ED Fig. 4k,l)), and finally through the O-layer channel, which is known to be flexible enough to open up^{7,23} (ED Fig. 4k,l).*"

At this stage, we wish to keep speculations to the minimum. Working out the details of this mechanism will take years. For this submission, we provide the first high resolution structure of a T4SS, which, in turn, provides for the first time a rationale for a plausible mechanism for conjugative pilus biogenesis. Moreover, in this revised version, we provide a validation of our model through site-directed mutagenesis. We feel that, with the wonderful help of the reviewers comments, we have lifted the manuscript to a level of excellence, novelty and general interest deserving publication in Nature.

Reviewer Reports on the First Revision:

Referees' comments:

Referee #1 (Remarks to the Author):

The authors have been highly responsive to reviewer comments. In my view the revised manuscript is suitable for publication.

Referee #2 (Remarks to the Author):

The authors have addressed the comments raised, provided new data (especially co-evolution analyses), re-structured and re-wrote the manuscript. In particular, this reviewer acknowledges that the interpretation of the results has now been carefully implemented. The manuscript reads well (despite the overall complexity of the system).

Re-addressing comment 12 (together with Image processing of the T4SS OMCC (see Methods section page 28) and figures ED Fig 1 (f, g):

The outcome of the rotational autocorrelation will be dependent on the processing strategy employed. It was recommended to initiate the reconstructions with very low filtered reference maps (40 or 60 Angstroms). However, from the methods section or the processing workflow (extended figure) it is unclear, to which resolution the reference map (obtained from a C14-symmetrized model PDB 3JQO) has been filtered. If a 20 Angstrom map has been used (as for picking particles (Methods section)), a potential bias for a 14-symmetric structure should be excluded. Ideally the authors should use a rotationally averaged structure obtained from PDB 3JQO (filtered to at least 40 Angstrom) as an initial reference map and reconstruct as C1 or try to do an a priori reconstruction without reference maps. Moreover, it is still puzzling, why the OMCC (C1) reconstruction shows a non-uniform Euler angle distribution (azimuth +/- 90 degrees), despite the fact that there is no obvious indication from the structure or the sample preparation that would lead to a preferred particle orientation.

Re-addressing comment 14:

In an attempt to visualise the unaccounted densities (shown now in Fig ED Fig 3p, green) (a) several densities could be identified and (b) the single stretched density in the upper part could not be seen. At this point in time, the basis for this discrepancy is still unclear. Maybe the authors could describe again, how they prepared the content of fig ED Fig 3p (or provide a ChimeraX-session file to follow the individual steps that could also go along with a future publication?) (see figure uploaded separately)

Referee #3 (Remarks to the Author):

I would like to congratulate the authors on the revised manuscript! Thank you for addressing all my concerns. The manuscript is, in my opinion, acceptable for publication.

Author Rebuttals to First Revision:

Dear Editor,

Please find enclosed a second revised version of our manuscript entitled “Cryo-EM structure of a type IV secretion system” by Kévin Macé, Abhinav K. Vadakkepat, Natalya Lukoyanova, Nathalie Braun, Adam Redzej, Marta Ukleja, Fang Lu, Tiago R.D. Costa, Elena V. Orlova, David Baker, Qian Cong, and Gabriel Waksman.

We were delighted to see that Reviewers 1 and 3 have accepted the paper and that Reviewer 2 has only 2 suggestions/comments left to address.

You also informed us that, in confidential comments, one of the referees indicated that it would be important to clarify that the work does not present the structure as a full, intact complex, which limits the ability to observe interactions and transitions *in situ*. To address this issue, we have added on page 3 and 4 of main text (part added in red): “*All protomers were related by an angle of 60° (ED Fig. 1t,u). The IMC is thus a hexameric structure with compositional heterogeneity (i.e. variable occupancy of its constituent protomers) as defined by Huiskonen¹⁷, thereby limiting the ability to observe interactions and transitions in situ.*” It seems to us a good place to add such a comment. Please let us know what you think.

We have addressed Reviewer 2’s two comments/suggestions in the following way.

Re-addressing comment 12:

Re-addressing comment 12 (together with Image processing of the T4SS OMCC (see Methods section page 28) and figures ED Fig 1 (f, g):The outcome of the rotational autocorrelation will be dependent on the processing strategy employed. It was recommended to initiate the reconstructions with very low filtered reference maps (40 or 60 Angstroms). However, from the methods section or the processing workflow (extended figure) it is unclear, to which resolution the reference map (obtained from a C14-symmetrized model PDB 3JQO) has been filtered. If a 20 Angstrom map has been used (as for picking particles (Methods section)), a potential bias for a 14-symmetric structure should be excluded. Ideally the authors should use a rotationally averaged structure obtained from PDB 3JQO (filtered to at least 40 Angstrom) as an initial reference map and reconstruct as C1 or try to do an a priori reconstruction without reference maps. Moreover, it is still puzzling, why the OMCC (C1) reconstruction shows a non-uniform Euler angle distribution (azimuth +/- 90 degrees), despite the fact that there is no obvious indication from the structure or the sample preparation that would lead to a preferred particle orientation.

Response:

We followed the reviewer’s suggestion and computed a new OMCC C1 map *Ab Initio*. This *Ab Initio* reconstruction is based on 100,000 particles automatically chosen by CRYOSPARC from the stack of 1,729,311 particles (see workflow in ED Fig. 1f reproduced below for the reviewer’s convenience). The new *Ab Initio* OMCC C1 map is presented in ED Fig. 1j, upper panels (reproduced below) and has been deposited in the EMDB. A section of this *Ab Initio* map is shown in ED Fig. 1r, upper left panel (reproduced below): it displays clear C14 symmetry. Further refinement using this map as reference and the entire stack of particles led to an improved high resolution OMCC C1 map at 3.28 Å resolution (ED Fig. 1k, reproduced below). Repeat of the symmetry analysis using IMAGIC yielded the same results found previously, i.e. C14 symmetry for the O-layer and C16 symmetry for the I-layer (ED Fig. 1r, reproduced below). This new analysis is bias-free and establishes beyond doubt the symmetry of the OMCC. The corresponding text in the Methods section has been updated and is reproduced here for the reviewer’s convenience:

“Symmetry analysis (ED Fig. 1f). 100,000 of these particles (chosen automatically by CRYOSPARC) were submitted to *ab initio* reconstruction with no symmetry applied and the resulting map was used as initial model to a 3D refinement with the same particles, yielding a map at 3.52 Å (referred in ED Table 1a as “*ab initio* model for OMCC C1” map; ED Fig. 1j). Symmetry of the O-layer was assessed visually by displaying sections of the corresponding

region of the *Ab Initio* map (ED Fig. 1r, top left panel), clearly indicating C14 symmetry for this region. Then, a 3D homogeneous refinement was carried out using this new map as initial model and using all 1,729,311 particles, yielding a C1 map of the OMCC with a resolution of 3.28 Å (referred in ED Table 1a as “OMCC C1 3.28 Å” map; ED Fig. 1k). To assess the symmetry in various regions of this map (O- and I-layers), sets of map sections were selected and extracted as separate images using the FIJI⁵⁰ software. Images were imported into IMAGIC⁵¹ where the function “rotational-auto-correlation” was used and results plotted (ED Fig. 1r)”.

The reviewer wonders why, in the previous OMCC C1 work, there appears to be a non-uniform angle distribution. The reviewer is right and this puzzled us as well. However, with the new *Ab Initio* approach, we no longer observe this phenomenon. Non-uniform distribution artifacts generated by the programme, notably at the 3D classification stage, have been reported (<https://discuss.cryosparc.com/t/heterogeneous-refinement-separates-orientations-rather-than-conformations/1544>). However, we no longer use 3D classification for the OMCC C1 reconstruction and that may be why the angle distribution is now uniform. In any case, the reviewer's suggestion has led us to produce a better C1 reconstruction for the OMCC with a uniform angle distribution and improved resolution.

Re-addressing comment 14:

In an attempt to visualise the unaccounted densities (shown now in Fig ED Fig 3p, green) (a) several densities could be identified and (b) the single stretched density in the upper part could not be seen. At this point in time, the basis for this discrepancy is still unclear. Maybe the authors could describe again, how they prepared the content of fig ED Fig 3p (or provide a ChimeraX-session file to follow the individual steps that could also go along with a future publication?) (see figure uploaded separately)

Response:

We hope it will be acceptable to the Reviewer and to the Editor that we remove this panel. The region being discussed here is indeed poorly defined and we have acknowledged this fact from the start. It represents a tiny part of the density map and its interpretation is unclear and subject to interpretation. Therefore, since the Reviewer remains unconvinced, it might just be better to remove the panel altogether. Removing this panel will have no consequence overall on the scope of the paper. Therefore, we have removed this panel and all discussion of the panel has been removed.

We thank the reviewers again for their formidable and enlightened input. Implementing their constructive and helpful suggestions have results in a much better manuscript which, we hope is now acceptable for publication in Nature.

Yours, sincerely

Gabriel Waksman

Reviewer Reports on the Second Revision:

Referees' comments:

Referee #2 (Remarks to the Author):

Response to 're-addressing comment 12'

It's great to see that the authors have tried to follow some of the recommendations to generate the structure for the OMCC in C1 and finally ended up with an improved map, additional validation for symmetry but also a more even distribution of Euler angles.

Response to 're-addressing comment 14'

The authors found density stretches within the structure that couldn't be convincingly explained – partly due to their low resolution and the authors now claim to remove the entire part from the manuscript (figure panel and description). Because the densities are in fact part of the structure this should not be neglected and the authors should mention and even show a figure of the unaccounted densities (within the extended data). This might also be of help addressing the structure and functioning of the systems in follow-up projects by other scientists in the future. Furthermore, it would emphasize that the authors carefully analyzed the entire structure and didn't leave out elements that – at least at this point in time – could not be fully explained.

The recommendation to the authors is to mention and show the densities in the extended data AND provide a ChimeraX session (or script/a few command lines in ChimeraX) file that goes along with the publication containing the model(s) and the EM map(s) as well as a separate map of only unaccounted densities (subtraction: IMC-Arches-Stalk_C1sym_6_2A-resolution_map_Sharpemd.mrc minus densities next to models (Stalk.cif, IMC_arches_hexamer.cif)).